# Epigenetic memory of radiotherapy in dermal fibroblasts impairs wound repair capacity in cancer survivors

Xiaowei Bian [1,13], Minna Piipponen [1,13], Zhuang Liu[1], Lihua Luo [1], Jennifer Geara[1], Yongjian Chen[1], Traimate Sangsuwan[2], Monica Maselli[1], Candice Diaz[3,4], Connor A. Bain[5], Evelien Eenjes[6], Maria Genander [6], Michael Crichton[5], Jenna L. Cash[7], Louis Archambault[8], Siamak Haghdoost[2,9], Julie Fradette[3,4,10], Pehr Sommar[11,12], Martin Halle [11,12,14] ✉ & Ning Xu Landén [1,14] ✉

Radiotherapy (RT), a common cancer treatment, unintentionally harms surrounding tissues, including the skin, and hinders wound healing years after treatment. This study aims to understand the mechanisms behind these late-onset adverse effects. We compare skin biopsies from previously irradiated (RT⁺) and non-irradiated (RT⁻) sites in breast cancer survivors who underwent RT years ago. Here we show that the RT⁺ skin has compromised healing capacity and fibroblast functions. Using ATAC-seq, we discover altered chromatin landscapes in RT⁺ fibroblasts, with *THBS1* identified as a crucial epigenetically primed wound repair-related gene. This is further confirmed by single-cell RNA-sequencing and spatial transcriptomic analysis of human wounds. Notably, fibroblasts in both murine and human post-radiation wound models show heightened and sustained *THBS1* expression, impairing fibroblast motility and contractility. Treatment with anti-THBS1 antibodies promotes ex vivo wound closure in RT⁺ skin from breast cancer survivors. Our findings suggest that fibroblasts retain a long-term radiation memory in the form of epigenetic changes. Targeting this maladaptive epigenetic memory could mitigate RT's late-onset adverse effects, improving the quality of life for cancer survivors.

With increased survival rates due to advancements in cancer diagnostics and treatments, long-term toxicities from therapies have emerged as a significant problem among cancer survivors. Radiotherapy (RT) is a widely used cancer treatment that inevitably leads to significant early or late side effects in normal tissues[1,2]. Among these, the skin is particularly susceptible due to its high cell turnover[1,2]. Alongside acute radiodermatitis, which typically resolves within weeks, late-onset adverse effects (LAEs), such as chronic ulceration and fibrosis, can manifest months to years after RT. In particular, surgery in irradiated tissues due to either cancer recurrences or breast reconstruction can be challenging and is often associated with

impaired wound healing[3]. Current approaches to managing radiation injury, including conventional wound care and surgical interventions, have proven limited success due to the compromised nature of previously irradiated tissues[4]. To address these challenges and improve outcomes for cancer survivors, it is crucial to develop targeted treatments that address the underlying mechanisms driving LAEs.

RT has been shown to induce DNA damage and release of reactive oxygen species through radiolysis of cellular water and induction of mitochondrial dysfunction[5,6], which cause cellular depletion, extracellular matrix (ECM) changes, microvascular damage, premature senescence, and altered pro-inflammatory mediator expression in

---

irradiated skin[7,8]. While these cellular changes contribute to acute radiodermatitis, the underlying molecular mechanisms of frequent LAEs are not well understood. Recent studies have revealed that ionizing radiation alters the epigenome, which can have long-lasting effects on gene expression, and therefore, may constitute a significant process mediating the development of LAEs[9]. For example, radiation-induced alterations to DNA methylation have been implicated in genomic instability and proposed as one of the pathological under-pinnings of radiation-induced carcinogenesis[9,10] and fibrosis[11]. More-over, chromatin dynamics have been described as mediating innate immune memory[12]. Intriguingly, not only immune cells but epidermal stem cells also harbor a memory of previous inflammation, which is carried out by maintaining the accessibility of the chromatin regions of several essential stress response genes to transcription factors, enabling the skin to respond to a secondary assault (e.g., injuries) more rapidly or intensely[13–15]. Building on these intriguing findings, we hypothesize that certain skin cell types surviving acute radiation injury may retain a 'radiation memory,' compromising their normal functions long after RT exposure.

In this study, we compare paired skin biopsies from previously irradiated (RT$^+$) and non-irradiated (RT$^-$) sites of the same breast cancer patients who underwent RT several years ago. We observe compromised ex vivo wound healing capacity and impaired fibroblast functions in the RT$^+$ skin. Using the assay of transposase-accessible chromatin sequencing (ATAC-seq), we identify an altered chromatin landscape in the RT$^+$ fibroblasts. Notably, we discover that thrombos-pondin 1 (*THBS1*), a gene crucial for wound repair, is epigenetically primed in the RT$^+$ fibroblasts, which led to elevated THBS1 production upon tissue injury. This elevation hinders fibroblast motility and con-tractibility, thereby delaying wound repair. Encouragingly, our study demonstrates that inhibition of THBS1 using its antibody improved ex vivo wound healing in RT$^+$ skin, offering a promising therapeutic avenue for addressing radiation ulcers and impaired wound healing following surgery in irradiated skin.

## Results

### The impaired healing capacity of late irradiated human skin

To evaluate the long-term effects of RT, we conducted a study invol-ving breast cancer survivors who underwent autologous-tissue breast reconstruction. These individuals had previously received external beam RT with a total dose of 40-60 Gy at least one year prior to the reconstruction surgery (Supplementary Data 1). Skin biopsies were collected from both the irradiated site (RT$^+$) on the chest and non-irradiated areas (RT$^-$) such as the abdomen or the opposite side of the chest during the reconstruction surgery (Fig. 1A).

Clinical examination did not reveal any noticeable differences in the macroscopic appearance of the RT$^+$ and RT$^-$ skin, suggesting that skin homeostasis was not severely compromised by the prior irradia-tion. Moreover, we conducted histological evaluations by following Masson's trichrome staining of these skin samples, and measured the collagen fiber orientation using the FIBRAL application. Our findings indicate that while there was no significant difference in collagen fiber alignment between the RT$^+$ and RT$^-$ skin, the contrast in alignment between the two groups decreased over time post-radiotherapy (Supplementary Fig. 1A, B).

To further investigate the healing capacity, we utilized an ex vivo wound healing model. Wounds were created on surgically discarded human skin, and the healing process was monitored in real-time using a fluorescent cell tracer dye (Fig. 1B)[16–18]. Surprisingly, we discovered a significant delay in wound closure in RT$^+$ skin compared to donor-matched RT$^-$ skin and the skin from healthy donors (HS, Fig. 1C–E, Supplementary Fig. 1C). Unlike the effective wound contraction observed in the RT$^-$ and HS skin, the RT$^+$ wounds exhibited minimal contraction during the repair process (Fig. 1D, E), a finding supported by histological analysis (Fig. 1F).

As wound contraction is primarily driven by dermal fibroblasts[19], we isolated fibroblasts from the RT$^-$ and RT$^+$ skin using an explant outgrowth approach (see **Methods**)[20] and assessed their functionality. During the cell isolation, we observed a significantly slower outgrowth of fibroblasts from the RT$^+$ skin explants compared to the RT$^-$ skin (Fig. 1G). The proliferation of RT$^+$ fibroblasts did not show a significant change when compared to RT$^-$ fibroblasts (Supplementary Fig. 1D, E). However, through scratch wound assays, we discovered that the migration of both RT$^+$ fibroblasts and keratinocytes was significantly slower in comparison to RT$^-$ cells (Fig. 1H, Supplementary Fig. 1F). We further investigated the effect of TGF-β, a known inducer of fibroblast differentiation into myofibroblasts[19], which express alpha-smooth muscle actin (ACTA2) mediating wound contraction[19] and produce increased amounts of ECM, e.g., fibronectin 1 (FN1), elastin (ELN), and collagens (COL1A1 and COL3A1). Our findings revealed that RT$^+$ fibro-blasts, upon TGF-β treatment, exhibited reduced expression of *ACTA2* and ECM when compared to their RT$^-$ counterparts (Fig. 1I, J, Supple-mentary Fig. 1G, H). The impaired expression of these critical genes in RT$^+$ fibroblasts may explain their reduced migration and wound heal-ing capabilities.

Collectively, our findings indicate that while skin previously exposed to RT can maintain homeostasis years later, inherent defects in dermal fibroblasts result in functional alterations that impair their wound healing abilities. Noteworthy, we observed no significant cor-relation between these functional defects in fibroblasts (such as migration and ex vivo wound healing) and the time elapsed since RT (Supplementary Fig. 1I, J).

### Altered chromatin landscape in late irradiated dermal fibroblasts

Epigenetic modifications can imprint a cell's past experiences onto its chromatin landscape, creating an epigenetic memory[14,15]. Recent stu-dies have identified various cell types, including epithelial cells in the skin, lung, intestine, and pancreas, as well as neurons, capable of retaining an inflammatory memory[13]. However, it remains unclear whether dermal fibroblasts, a long-lived tissue cell type, also possess the ability to harbor an epigenetic memory. In light of this, we propose the existence of a "maladaptive" radiation memory in skin fibroblasts that have withstood the effects of acute radiation injuries.

To investigate this hypothesis, we performed ATAC-seq to iden-tify genomic regions with open chromatin states in paired RT$^-$ and RT$^+$ fibroblasts that were isolated by explant outgrowth[21]. As expected, we observed an enrichment of ATAC-seq signals near transcription start sites (TSSs) (Fig. 2A). While the overall distribution patterns of open chromatin regions across various genomic and intergenic regions were similar between the two groups (Supplementary Fig. 2A), we identified 74 peaks showing increased accessibility in 59 genes [log$_2$(fold change, FC) > 0, false discovery rate (FDR) < 0.05, denoted as RT$^+$ up domains] and 97 less accessible peaks that were annotated in 84 genes (log$_2$FC < 0, FDR < 0.05, denoted as RT$^-$ up domains) in RT$^+$ fibroblasts compared to RT$^-$ fibroblasts (Fig. 2B, Supplementary Fig. 2B-C, Sup-plementary Data 2). We did not find any significant correlation between these epigenetic changes with the time elapsed since RT (Supplementary Fig. 2D, E). Moreover, we performed ATAC-seq on dermal fibroblasts from skin areas with surgery 9-25 months prior but no radiotherapy (S+) and from areas without surgery and radiotherapy (S-) in three breast cancer survivors (Supplementary Data 1). We found that previous surgery induced increased chromatin accessibility in 16 genes, with only one gene, *HOXA3*, among the 59 genes with the RT$^+$ up domains (Supplementary Fig. 2F, Supplementary Data 3). Notably, *THBS1* locus accessibility did not differ between S+ and S- fibroblasts, suggesting that the lasting epigenetic changes in RT$^+$ fibroblasts are unlikely due to previous surgery.

To determine if patient fibroblasts' epigenetic changes stem from RT, we exposed normal human fibroblasts to 8 Gy of ionizing radiation

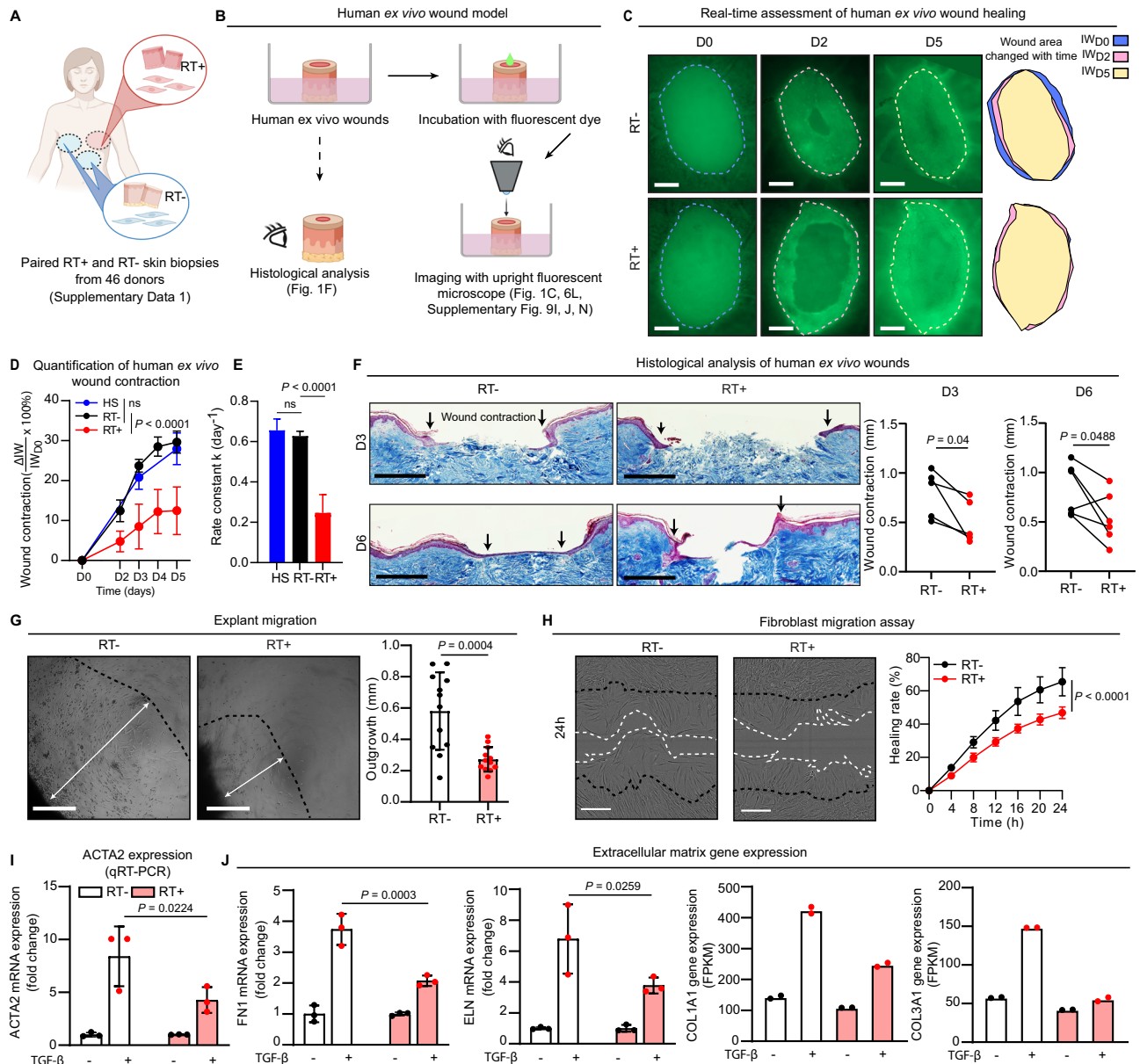

**Fig. 1 | The reduced healing capacity of late irradiated human skin. A** Collection of skin biopsies from previously irradiated (RT⁺) and non-irradiated (RT⁻) sites of 46 donors. **B** Illustration of human ex vivo wound model. **C** Representative images of ex vivo wounds on RT⁻ and RT⁺ skin on days 0-5 post-injury (RT⁻, $n = 6$; RT + , $n = 7$). The initial wound edges were demarcated with dashed lines. The areas within the initial wound edge (IW$_{time point}$) are illustrated (right panel). Scale bars: 500 μm. **D** Wound contraction (%) was quantified as ΔIW$_{time point}$/IW$_{D0}$ × 100% (RT⁻, $n = 6$; RT⁺, $n = 7$; HS, $n = 5$). **E** Healing kinetics calculated from wound contraction area from day 0 to day 5 using one-phase decay model. **F** Masson's trichrome staining of human ex vivo wounds on days 3 and 6 post-injury. Wound contraction was evaluated by the distance between the initial wound edges indicated with arrows (D3,

$n = 5$; D6, $n = 6$). Scale bars: 500 μm. **G** Fibroblast outgrowth from human skin explants after ten days. Outgrowth distance (indicated by white arrows) was quantified ($n = 12$). Scale bars: 400 μm. **H** Scratch wound assay of RT⁻ and RT⁺ fibroblasts ($n = 5$). Scale bars: 300 μm. Expression of *ACTA2* (**I**) and extracellular matrix genes (**J**) in human fibroblasts treated with or not with TGF-β. *ACTA2, FN1*, and *ELN* were analyzed by qRT-PCR, $n = 3$; *COL1A1* and *COL3A1* were analyzed by RNA-seq, $n = 2$. Data are presented as means ± SEM (**D**) or means ± SD (**E, G**). Two-way ANOVA (**D, H**), paired (**F**), or unpaired two-tailed student's t-test (**E, G**). Figures 1A and 1B were created with BioRender.com released under a Creative Commons Attribution-NonCommercial-NoDerivs 4.0 International license. Source data are provided as a Source Data file.

and collected them one day (D1), seven days (D7), and 14 days (D14) post-treatment. ATAC-seq analysis revealed increased chromatin accessibility in 2000 genes at D1, 2765 genes at D7, and 1441 genes at D14 compared to non-irradiated cells (log₂FC > 0, *p*-value < 0.05) (Fig. 2C, Supplementary Data 4). Notably, 9, 17, and 9 of the 59 genes with RT⁺ up domains showed greater accessibility post-irradiation in D1, D7, and D14 samples, respectively (Fig. 2C, Supplementary Data 4). Interestingly, apart from two differentially accessible chromatin domains in *SIN3B* and *NPSR1* that overlapped in both in vitro and in vivo settings, the majority of changes in chromatin accessibility

following in vitro irradiation and post-radiotherapy were not identical. Instead, these changes were reflected across different chromatin domains but converged on the same genes (Supplementary Data 4). Additionally, we conducted focal irradiation (IR) on the skin of C57BL/6 mice (Fig. 5A, Supplementary Fig. 7I). ATAC-seq was performed on dermal cells isolated from irradiated (IR⁺) and non-irradiated (IR⁻) skin one (D1) and seven days (D7) after IR. We identified 3675 genes and 9913 genes with increased accessibility in IR⁺ dermal cells compared to IR⁻ cells (log2FC > 0, *p*-value < 0.05) on D1 and D7, respectively (Supplementary Fig. 7I, Supplementary Data 9). Notably, 34 of the 59 genes

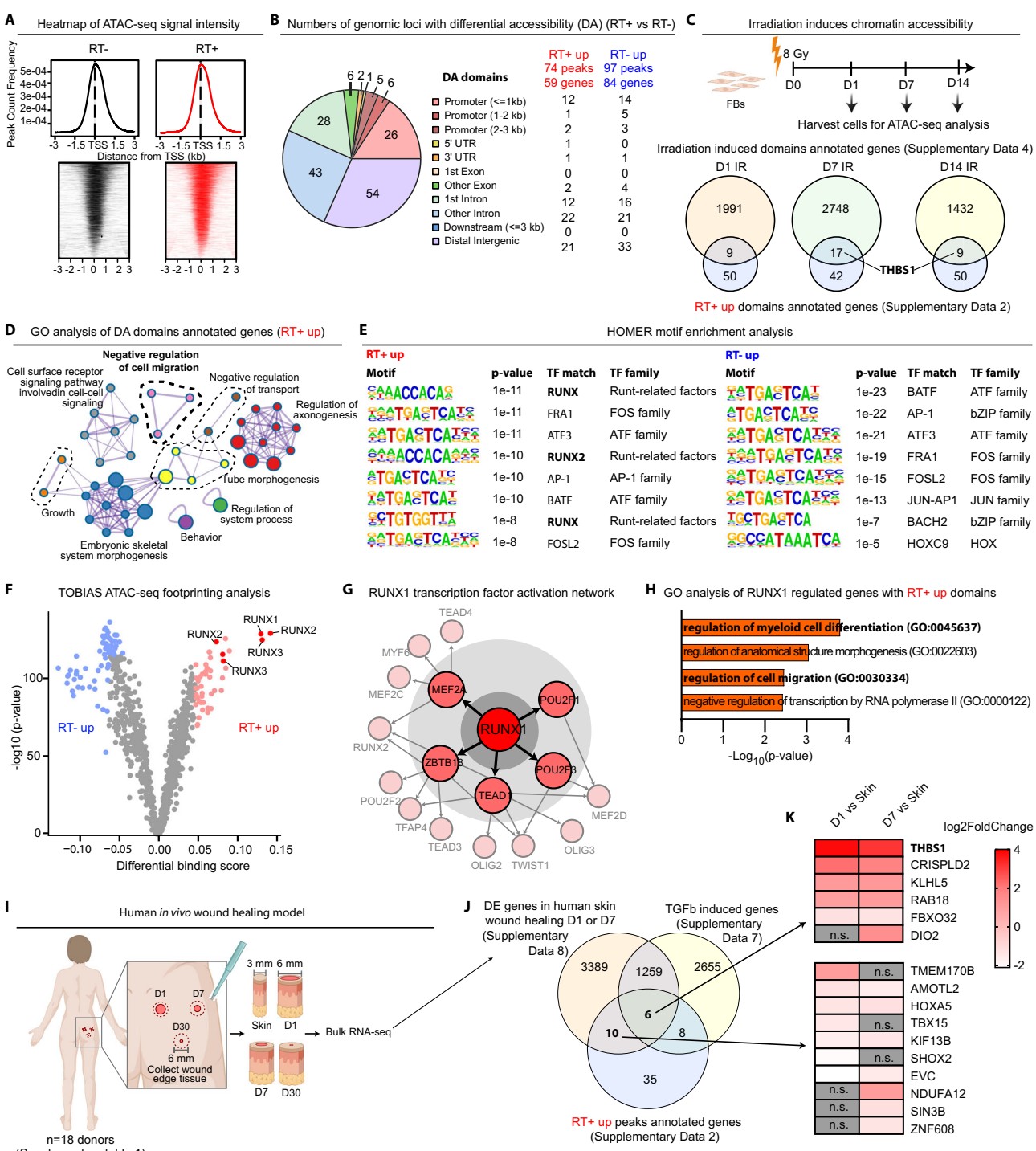

**Fig. 2 | Altered chromatin landscape in late irradiated dermal fibroblasts.**
**A** Heatmap of ATAC-seq signal intensity over chromatin peaks of RT⁻ and RT⁺ fibroblasts (n = 5 donors). TSS, transcription starting site. Kb, kilobase pairs. **B** Numbers of genomic loci with differential accessibility between RT⁻ and RT⁺ fibroblasts. **C** Comparison of genes with RT⁺ up domains against genes exhibiting chromatin accessibility induced by irradiation. The overlapping genes are detailed in Supplementary Data 4. **D** GO analysis of genes with RT⁺ up domains: each node represents a biological process, node size indicates the number of gene enriched in that process. **E** HOMER motif analysis of the DA domains. **F** Comparison of transcription factor (TF) activities between RT⁺ and RT⁻ fibroblasts. Volcano plot showing TOBIAS differential binding score and -log10 (p-value); each dot represents one TF, and the top 5% TF are highlighted in blue or pink. RUNX family members are indicated in red. **G** A TF-TF network is built of RUNX1 and the other

top 5% TF genes in RT⁺ fibroblasts. Sizes of nodes represent the level of the network starting with RUNX1. Directed edges indicate TF binding sites in the respective gene. **H** GO analysis of the RT⁺ up domain annotated genes that RUNX1 targets. **I** Illustration of human in vivo wound healing model. **J** Comparison of differentially expressed (DE) genes in day 1 (D1) and day 7 (D7) human acute wounds versus skin (n = 5 donors, |log2FC | > 1, FDR < 0.05) with RT⁺ up domain genes in RT⁺ fibroblasts and genes induced by TGF-β treatment in fibroblasts. The overlapping genes are detailed in Supplementary Data 7. **K** A heatmap displays the expression changes of overlapping genes in human acute wounds versus skin. Hypergeometric test (**D, E, H**). Two-tailed student's t-test (**F**). Wald test (**J**). n.s.: no significant change. Figure 2I were created with BioRender.com released under a Creative Commons Attribution-NonCommercial-NoDerivs 4.0 International license.

with the RT$^+$ up domains in human fibroblasts, including *THBS1*, showed greater accessibility in murine cells post-irradiation. Despite challenges such as differences between fibroblasts' in vivo and in vitro states, human-mouse variations, and differences in clinical and experimental radiation exposures, our ATAC-seq data suggest that some lasting epigenetic changes in cancer survivors' fibroblasts are directly associated with radiation.

## Impacts of radiotherapy-induced epigenetic changes on fibroblast functions

We delved deeper into the biological effects of these epigenetic changes in fibroblasts from cancer survivors post-radiation. Gene Ontology (GO) analysis unraveled that genes associated with the RT$^+$ up domains and RT$^-$ up domains were enriched in terms related to negative regulation of cell migration and organ development, respectively (Fig. 2D, Supplementary Fig. 2G, Supplementary Data 5). Utilizing HOMER (hypergeometric optimization of motif enrichment) motif analysis[22] and TOBIAS transcription factor (TF) footprinting[23] on the ATAC-seq dataset of patient fibroblasts, we observed an increased presence of RUNX motifs and a strong enrichment in the differential binding score for RUNX footprints in the RT$^+$ up domains (Fig. 2E, F, Supplementary table 2). Notably, the RUNX gene was not differentially expressed between the RT$^+$ and RT$^-$ fibroblasts (Supplementary Fig. 2H). To explore the relationship between RUNX and the other top TFs in the RT$^+$ fibroblasts [identified using TOBIAS TF footprinting: differential binding scores > 95% quantiles or -log$_{10}$($p$-value) > 95% quantile], we constructed a TF-activation network. Our model revealed that RUNX1 activated five primary TFs, which subsequently activated the expression of 11 additional TFs, highlighting the central role of RUNX1 in the epigenetic changes associated with late irradiation (Fig. 2G). RUNX1 is a known TGF-β regulated TF required for the proliferation and differentiation of mesenchymal progenitor cells[24,25]. Our GO analysis further suggested that the genes associated with RT$^+$ up domains and regulated by RUNX were involved in cell migration (Fig. 2H, Supplementary table 3).

Moreover, we conducted RNA-seq on fibroblasts from five patients, revealing minor gene expression differences between the RT$^+$ and RT$^-$ cells (Supplementary Fig. 3A, Supplementary Data 6). Notably, the majority (93%) of genes with chromatin accessibility changes in the RT$^+$ cells were not expressed or not differentially expressed between the RT$^+$ and RT$^-$ cells, suggesting that the epigenetic memory of radiotherapy was not unfolded at the homeostatic cell state (Supplementary Fig. 3B). This corresponds with the comparable clinical and histological profiles of RT$^+$ and RT$^-$ skin (Supplementary Fig. 1A, B).

We have shown that RT$^+$ and RT$^-$ cells respond differently to TGF-β, a critical regulator of fibroblast function in wound healing (Fig. 1I, J, Supplementary Fig. 1G, H)[26]. TGF-β is known to increase RUNX1 expression and activity[25,27]. Accordingly, we conducted RNA-seq analysis on both RT$^+$ and RT$^-$ cells with TGF-β treatment (Supplementary Fig. 3C, Supplementary Data 7). GO analysis revealed that TGF-β more strongly induced ECM-related genes in RT$^-$ cells than in RT$^+$ cells, while genes linked to ribosome biogenesis were more upregulated by TGF-β in RT$^+$ cells (Supplementary Fig. 3C). Additionally, we found that TGF-β activated 14 of the 59 genes with RT$^+$ up domains, suggesting it can trigger part of the epigenetic memory in RT$^+$ fibroblasts (Fig. 2J, Supplementary Data 7).

To explore how epigenetic changes in late irradiated fibroblasts affect wound healing, we concentrated on genes showing altered (upor down-regulated) expression during this process. To this end, we developed a unique human in vivo wound healing model. In this model, excisional wounds were created on healthy volunteers' skin, and wound-edge tissues were collected at different healing stages, including day 1, day 7, and day 30, from the same donors (Fig. 2I, Supplementary table 1). Through RNA-seq of these samples, we unraveled in vivo gene expression dynamics during human skin wound

repair (GSE174661)[28]. By comparing the 4664 differentially expressed genes in human wounds [Day1 or Day7 wounds vs. the donor-matched skin: |log$_2$FC | > 1, FDR < 0.05] with genes showing increased chromatin accessibility in the RT$^+$ fibroblasts, we identified 16 overlapping genes, which potentially link the epigenetic alterations to the reduced healing capacity of RT$^+$ fibroblasts (Fig. 2J, K, Supplementary Data 8).

## *THBS1* is epigenetically primed in late irradiated dermal fibroblasts

Among the 16 genes identified as epigenetically altered post-RT and active in wound repair (Fig. 2J, K), we specifically focused on *THBS1*. This gene showed the most significant changes at Day1 and Day7 acute wounds compared to the skin (Fig. 2K). Despite RNA-seq (Supplementary Data 6) and qRT-PCR (Fig. 3A) revealing no significant difference in *THBS1* expression between RT$^+$ and RT$^-$ fibroblasts, the disparity in *THBS1* mRNA levels between these groups lessened over time post-RT, independent of donor age or radiation dose (Supplementary Fig. 4A–C). ATAC-seq analysis identified six *THBS1* genomic regions, five in the promoter, with significantly higher accessibility in RT$^+$ fibroblasts (Fig. 3B, Supplementary Data 2). HOMER motif enrichment analysis indicated two RUNX1 binding sites (5'-TGTGGT-3') in these regions, and TOBIAS TF footprinting showed higher binding scores for these sites in RT$^+$ fibroblasts (Fig. 3B). Additionally, GO analysis highlighted *THBS1* as a RUNX-regulated gene involved in cell migration (Fig. 2H, Supplementary table 3).

To validate the binding of RUNX1 to these regions, we performed RUNX1 chromatin immunoprecipitation followed by quantitative polymerase chain reaction (ChIP-qPCR) analysis. The results confirmed the binding of RUNX1 to these two regions specifically in RT$^+$ fibroblasts, while no binding was observed in paired RT$^-$ fibroblasts (Fig. 3C, D), consistent with their differential chromatin accessibility states (Fig. 3B). Intriguingly, we observed that TGF-β induced the recruitment of RUNX1 to these motifs in both RT$^+$ and RT$^-$ fibroblasts. However, RT$^+$ fibroblasts exhibited a significantly higher recruitment of RUNX1 to the *THBS1* promoter compared to RT$^-$ fibroblasts (Fig. 3D). Furthermore, ChIP-qPCR analysis revealed more robust H3K4me1 histone modification near these RUNX1 binding sites in RT$^+$ fibroblasts than RT$^-$ fibroblasts, which was further enhanced after TGF-β treatment (Fig. 3E). This histone modification is indicative of an open and primed chromatin state that persists in epigenetic memory domains long after the initial stimulus has been withdrawn[29]. Aligned with the chromatin state and histone modification of the *THBS1* gene, both RNA-seq (Supplementary Fig. 3C) and qRT-PCR (Fig. 3F) analyses showed a notably higher induction of *THBS1* mRNA in RT$^+$ fibroblasts than in RT$^-$ fibroblasts following TGF-β treatment. Noteworthy, we determined that a 5 ng/ml concentration of TGFβ, close to physiological levels in human wounds[30], is the minimum required to induce THBS1 expression in RT$^+$ fibroblasts without affecting RT$^-$ cells (Supplementary Fig. 4D). These findings indicate that RT$^+$ fibroblasts might produce more *THBS1* during wound healing due to a more open and primed chromatin state at this gene compared to RT$^-$ fibroblasts.

To investigate whether the differential *THBS1* expression could be attributed to previous radiotherapy, we irradiated RT$^-$ fibroblasts and fibroblasts isolated from healthy donors by explant outgrowth (HDFa) with 8 Gy and collected samples from six hours to two weeks post-treatment (Fig. 3G). We observed a transient upregulation of cyclin-dependent kinase inhibitor 1 A (*CDKN1A*) and proliferating cell nuclear antigen (*PCNA*) expression post-irradiation, with their levels increasing within six hours and returning to baseline at approximately one week, indicating DNA damage repair activation (Fig. 3H, I, Supplementary Fig. 4E, F)[31,32]. However, *THBS1* transcript levels remained unchanged over two weeks (Fig. 3J, Supplementary Fig. 4G). In ATAC-seq of fibroblasts D1 and D7 post-irradiation, increased accessibility of the *THBS1* gene was observed only in D7 samples, suggesting the epigenetic changes in *THBS1* gene occur during the DNA damage repair

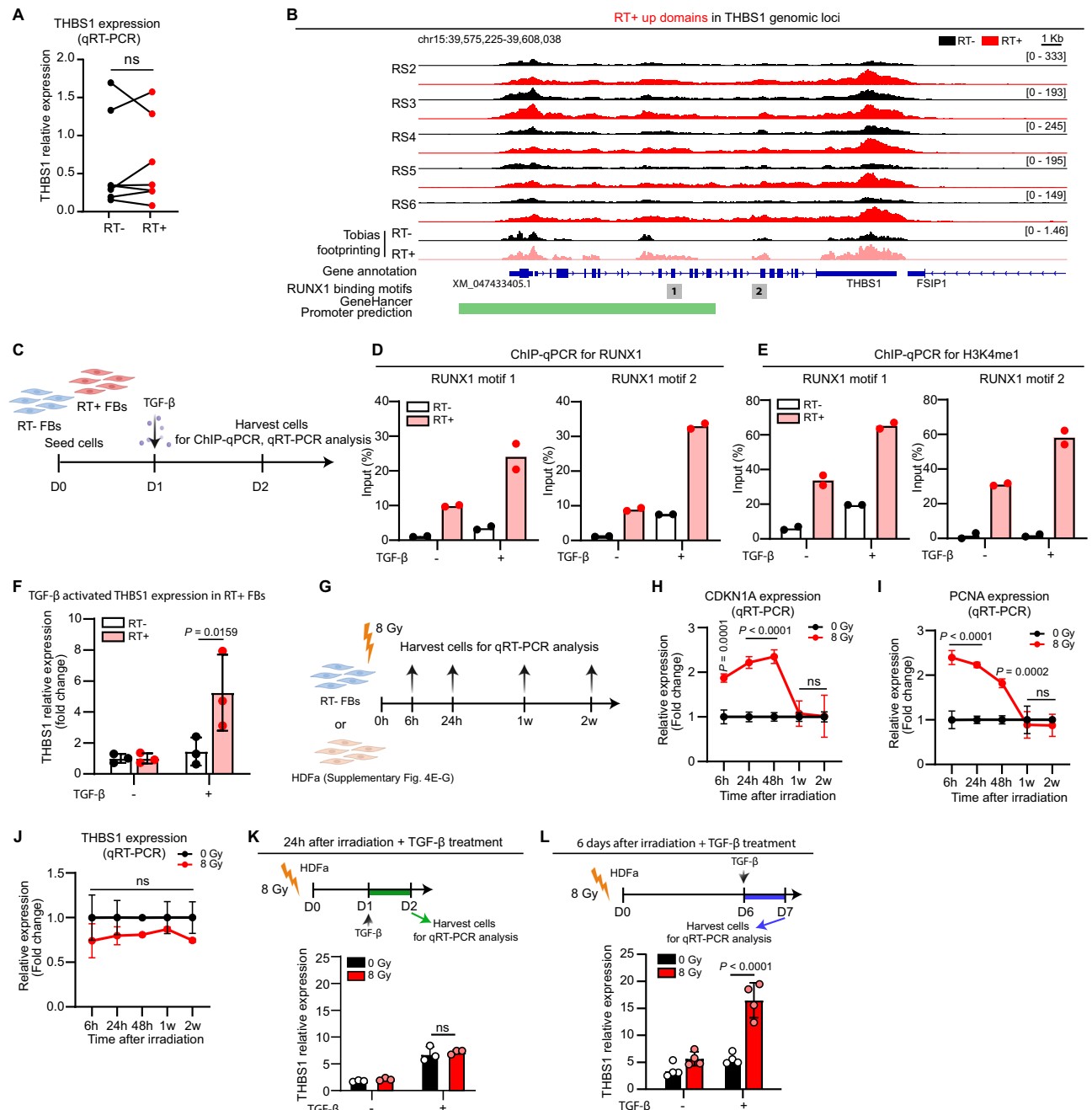

**Fig. 3 | _THBS1_ is epigenetically primed in late irradiated dermal fibroblasts.**
**A** qRT-PCR of _THBS1_ in paired RT⁻ and RT⁺ fibroblasts ($n = 7$ donors). **B** Integrative genomics viewer image of _THBS1_ genomic loci with ATAC signals in paired RT⁻ vs. RT⁺ fibroblasts ($n = 5$ donors). The analysis of TOBIAS footprinting in the paired RT⁻ and RT⁺ fibroblasts is shown in gray and pink tracks. The green bar indicates a predicted promoter region of _THBS1_ and two RUNX1 binding motifs are identified. **C** Paired RT⁻ and RT⁺ fibroblasts were treated with or not with TGF-β for 24 hours and the cells were harvested for ChIP-qPCR detecting RUNX1 binding (**D**) and H3K4me1 histone modification (**E**) at the _THBS1_ gene ($n = 2$), and qRT-PCR of _THBS1_ expression ($n = 3$) (**F**). **G** Fibroblasts from RT⁻ skin or skin of healthy donors were irradiated and harvested 6 hours – 2 weeks post-irradiation. qRT-PCR analysis of _CDKN1A_ **H**, _PCNA_ **I**, and _THBS1_ (**J**) in non-irradiated vs. irradiated fibroblasts ($n = 3$). Fibroblasts were irradiated and then treated with or not with TGF-β 1 day ($n = 3$) **K**, or 6 days post-irradiation ($n = 4$) **L**. _THBS1_ expression was analyzed by qRT-PCR. Data are presented as means ± SD **D**, **F**, **H–J**, **K**, **L**. Paired two-tailed student's t-test (**A**) or two-way ANOVA **F**, **H–J**, **K**, **L**. Source data are provided as a Source Data file.

process (Fig. 2C, Supplementary Data 4). Correspondingly, treating cells with TGF-β six days after radiation resulted in more _THBS1_ expression compared to non-irradiated cells, a response not seen in fibroblasts treated with TGF-β three hours or one day post-irradiation (Fig. 3K, L, Supplementary Fig. 4H).

Furthermore, we utilized the 10x Genomics Chromium Single Cell Multiome ATAC+Gene Expression platform to analyze paired RT⁺ and RT⁻ skin from a cancer survivor (Supplementary Data 1). This technique

provided in vivo gene expression and epigenomic data from single cells in the patient's skin (Supplementary Fig. 5A). Among 14 skin cell types, the _THBS1_ promoter region showed greater accessibility in RT⁺ fibroblasts, although mRNA levels were similar between RT⁺ and RT⁻ groups (Supplementary Fig. 5B, C). Moreover, despite comparable RUNX expression in both RT⁺ and RT⁻ fibroblasts, RUNX family motif activity was higher in RT⁺ cells, supporting RUNX's role in the long-term impact of radiotherapy (Supplementary Fig. 5D, E).

Together, these findings confirmed that radiation primed fibroblasts and enabled them to express higher levels of *THBS1* in response to TGF-β signaling during wound healing.

### Dynamic expression of *THBS1* during human skin wound healing

Thbs1 has been implicated in the regulation of tissue repair in various mice models[33]. Interestingly, both deletion and overexpression of *Thbs1* have been associated with delayed wound repair, suggesting that the quantity and duration of its expression are crucial for tissue healing[34–36]. Thbs1 expression post-wounding in mice has been previously characterized through in situ hybridization, immunostaining[34,35,37,38], and single-cell RNA-seq[39,40] (Supplementary Fig. 6A, B). Our own studies in a murine wound healing model, using both qRT-PCR (Supplementary Fig. 6C) and single-cell RNA sequencing (scRNA-seq, Supplementary Fig. 6D), have confirmed these *Thbs1* expression dynamics. We observed its minimal expression in unwounded skin, a rapid increase following injury, and a return to baseline levels during the remodeling phase.

To probe the in vivo expression pattern of *THBS1* during human skin wound healing, we analyzed the samples from our human wound healing model (Fig. 2I, Supplementary table 1 using scRNA-seq and spatial transcriptomics (ST) techniques. The scRNA-seq analysis of 16,098 cells from Day-1 acute wounds of three healthy donors identified 26 cell types (Fig. 4A). We noted that *THBS1* expression was primarily in fibroblasts and occurred at lower levels in macrophages and angiogenic cells, including vascular endothelial cells (VE), pericytes, and vascular smooth muscle cells (PC-vSMC) (Fig. 4A, Supplementary Fig. 6E). Within the fibroblast sub-clusters, FB-III (papillary fibroblasts) showed the highest expression of *THBS1*, while FB-IV (proliferating fibroblasts) had the least expression (Supplementary Fig. 6F)[41,42]. Moreover, bulk RNA-seq analysis revealed a rapid and transient upregulation of *THBS1* expression in human Day-1 and Day-7 acute wounds compared to the skin[28] (Fig. 4B). This was also observed in CD90⁺ dermal cells, rich in fibroblasts, from these wound tissues (Fig. 4C).

Additionally, we employed the cell2location method[43] to deconvolute our ST data using the scRNA-seq results of human wounds, elucidating the spatial distribution of fibroblasts with single-cell precision (Fig. 4D). Our analysis revealed a spike in *THBS1* expression in fibroblasts at Day 1 post-wounding, which then returns to baseline by Day 30 (Fig. 4E, F). Moreover, we observed a positive correlation between *THBS1* expression and the number of fibroblasts in human skin and wounds (Fig. 4G). This finding was further confirmed by fluorescence in situ hybridization (FISH) analysis of *THBS1* mRNA in additional donor-matched skin and wound tissues (Supplementary Fig. 6G). Notably, the upregulated *THBS1* expression was localized to the wound bed, with minimal *THBS1* signal detected in the surrounding skin away from the wound-edge (Supplementary Fig. 6G).

Remarkably, CellChat analysis of the scRNA-seq dataset highlighted THBS1 signaling as one of the prominent cell-cell crosstalks during human skin wound healing[44]. THBS1, primarily produced by fibroblasts, acts as a ligand and interacts with its receptors, such as CD47, CD36, SDCs, and integrins, present on keratinocytes, angiogenic cells, and immune cells (Fig. 4H, I, Supplementary Fig. 6E). Together, our results demonstrated a rapid and transient elevation of THBS1 signaling derived from fibroblasts in human acute wounds, emphasizing its crucial role in the wound repair process.

### Aberrant *THBS1* expression in wounds of late irradiated skin

Our study reveals the lasting impact of RT on wound healing in breast cancer survivors, supporting our recent findings on the extended effects of radiation on wound repair observed in a murine post-radiation wound model built on CD-1 mice[45]. Briefly, murine skin was exposed to increasing unique doses of 6MV photons, namely 45, 60, or 80 Gy. Four weeks after irradiation, excisional wounds were created on

both previously irradiated (IR⁺) and non-irradiated (IR⁻) dorsal skin regions, and the wound healing process was monitored for 33 days (Supplementary Fig. 7A). This model demonstrated significantly delayed wound healing in the IR⁺ skin compared to the IR⁻ skin[45] (Supplementary Fig. 7B, C). The granulation tissue of the IR⁺ wounds appeared less organized with a lower proportion of the neodermis occupied by collagen fibers[45]. We performed co-staining of Thbs1 (using FISH and immunofluorescence staining, IF) with the fibroblast marker Pdgfra (using IF), confirming increased Thbs1 mRNA and protein expression in fibroblasts of IR⁺ versus IR⁻ murine wounds (Supplementary Fig. 7D, E, Supplementary Fig. 8A). Additionally, we carried out IF co-staining for pSMAD2, a downstream effector of TGF-β signaling, and Pdgfra (Supplementary Fig. 8B, C). Our results show a more pronounced and persistent TGF-β signal in IR⁺ D33 wound fibroblasts compared to their IR⁻ counterparts. Interestingly, this disparity in TGF-β signaling was also present in IR⁺ compared to IR⁻ mouse skin prior to wounding (Supplementary Fig. 8C).

In addition to the CD-1 murine model, we established another post-radiation wound model on C57BL/6 mice (Fig. 5A). We performed focal irradiation (20 Gy) on the murine skin, which induced acute radiation effects (erythema, desquamation, ulceration, evaluated with RTOG scores[46]) appearing around 5 days and peaking at 14 days post-IR, with macroscopic recovery by 37 days (Supplementary Fig. 7F, H). qRT-PCR analysis revealed that dermal expression of Cdkn1a, a marker for DNA damage repair, increased within one day and returned to baseline at approximately one-week post-IR (Supplementary Fig. 7G). We created wounds at both IR⁺ and IR⁻ sites 45 days post-IR, once acute effects had subsided, and also on non-irradiated control mice (Ctr). Consistent with the CD-1 model results (Supplementary Fig. 7B, C), we observed significantly delayed wound healing in the C57BL/6 murine model (Fig. 5B, C). Notably, dermal *Thbs1* expression was rapidly induced upon injury in both IR⁺ and IR⁻ murine skin (Fig. 5D). However, *Thbs1* levels remain elevated in IR⁺ wounds even at the late healing stage (seven and ten days post-wounding), whereas in IR⁻ wounds, *Thbs1* returns to basal levels (Fig. 5D).

Furthermore, we examined the expression dynamics of *THBS1* in human ex vivo wounds at Day-3 and Day-6, which were created on paired RT⁻ and RT⁺ skin samples. Through FISH analysis, we observed transient upregulation of *THBS1* expression in RT⁻ dermis during wound healing, peaking at Day 3 in ex vivo wounds. Additionally, *THBS1* mRNA expression was elevated in the wound edge dermis of RT⁺ skin compared to RT⁻ skin (Fig. 5E). This finding was further confirmed through qRT-PCR quantification (Supplementary Fig. 8D). Moreover, IF staining revealed a more pronounced THBS1 protein expression in the RT⁺ human ex vivo wounds compared to the RT⁻ wounds (Supplementary Fig. 8E).

Based on the murine and human data, we concluded that fibroblasts primed by previous RT exhibited a more potent and persistent *THBS1* expression during wound repair. Intriguingly, we observed intense *THBS1* mRNA and protein expression in human chronic radiation ulcers but not in the donor-matched normal skin, as shown by FISH and IF analysis, clearly endorsing the clinical relevance of our findings (Fig. 5F, G).

### Targeting THBS1 improves wound healing of late irradiated human skin

To elucidate the impact of aberrant *THBS1* expression, we employed the CRISPR/dCas9 SAM system to activate endogenous *THBS1* transcription in human dermal fibroblasts[47]. Among six single-guide RNAs (sgRNAs) targeting different regions within 300 bp upstream of the *THBS1* transcription starting site (TSS), we found that sgRNA 6 enhanced *THBS1* expression approximately three-fold (Fig. 6A, Supplementary Fig. 9A), akin to the level of *THBS1* overexpression observed in both human and murine post-radiation wound models. Notably, the elevated *THBS1* expression significantly decreased

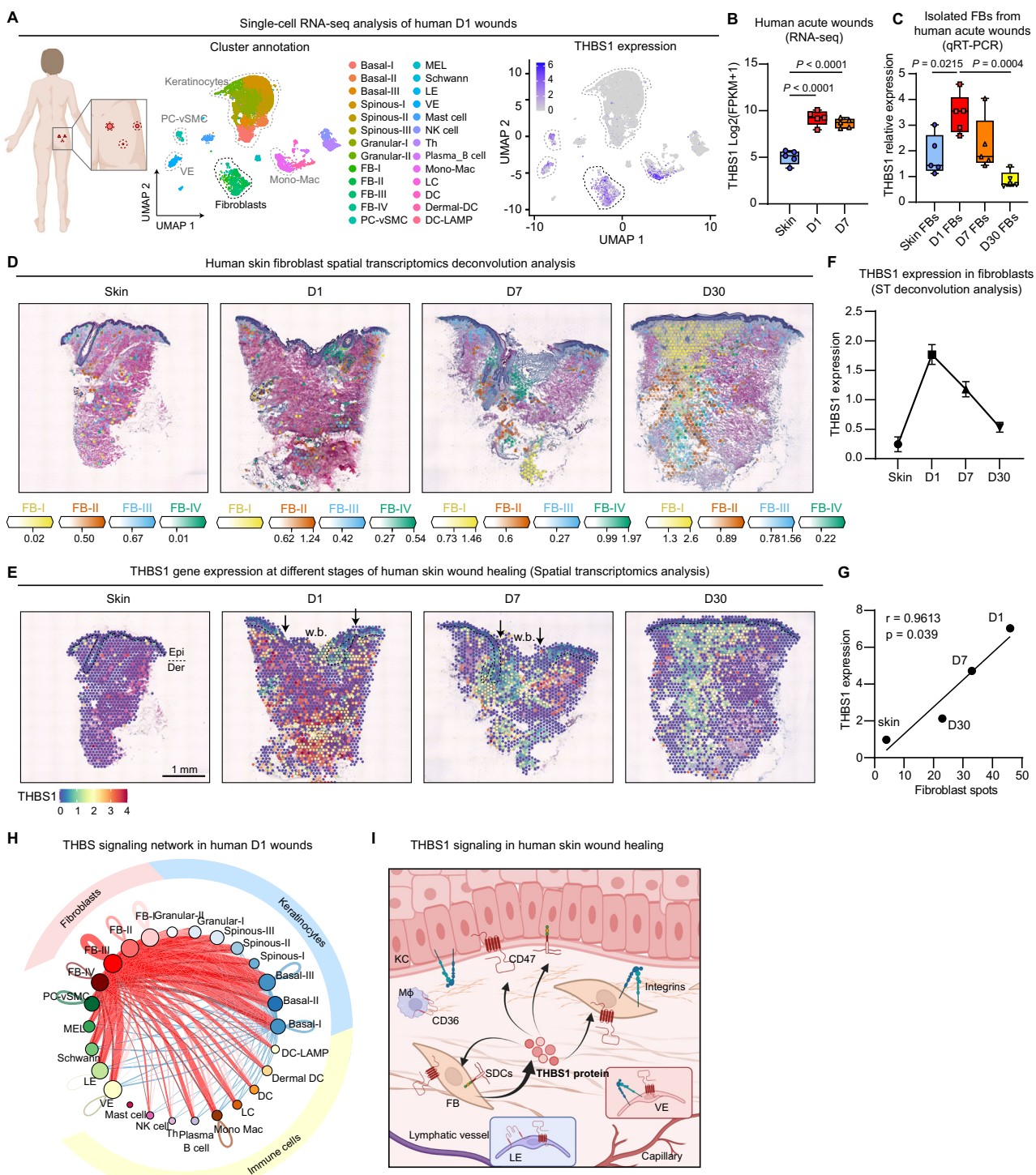

**Fig. 4 | Dynamic *THBS1* expression during human skin wound healing.** The human in vivo wound healing model was analyzed by single-cell RNA-seq (**A, H, I**), bulk RNA-seq (**B**, qRT-PCR **C**), and spatial transcriptomics (ST) **D-G. A** UMAP plot of 16,098 cells from the day1 (D1) wounds of three healthy donors, color-coded by cell type (left). *THBS1* expression is shown (right). *THBS1* expression in human skin and wound tissues (*n* = 5 donors) **B** and in CD90+ dermal cells isolated by magnetic activation cell sorting from these tissues (*n* = 5 donors) **C**. In each box plot, the center line indicates the median, the edges of the box represent the first and third quartiles, and the whiskers extend to the minimum and maximum values. **D** Spatial transcriptomic deconvolution analysis of fibroblast sub-clusters. **E** Spatial feature plots showing *THBS1* expression: epidermal-dermal junctions are indicated with

dashed lines, and arrows point wound edges. (**F**) THBS1 expression in ST deconvolution fibroblast spots (skin, *n* = 28 spots; D1, *n* = 55 spots; D7, *n* = 44 spots; D30, *n* = 64 spots). (**G**) Correlation of THBS1 expression with fibroblast spots. **H** THBS-mediated cell-cell communication in human day-1 acute wounds. Edge width is proportional to the inferred interaction strength. The edge color is consistent with the signaling source. **I** Schematic illustration of THBS1 signals in human wounds. Data are presented as means ± SEM **F**. One-way ANOVA (**B, C**). Pearson's correlation test **G**. Figure 4I was created with BioRender.com released under a Creative Commons Attribution-NonCommercial-NoDerivs 4.0 International license. Source data are provided as a Source Data file.

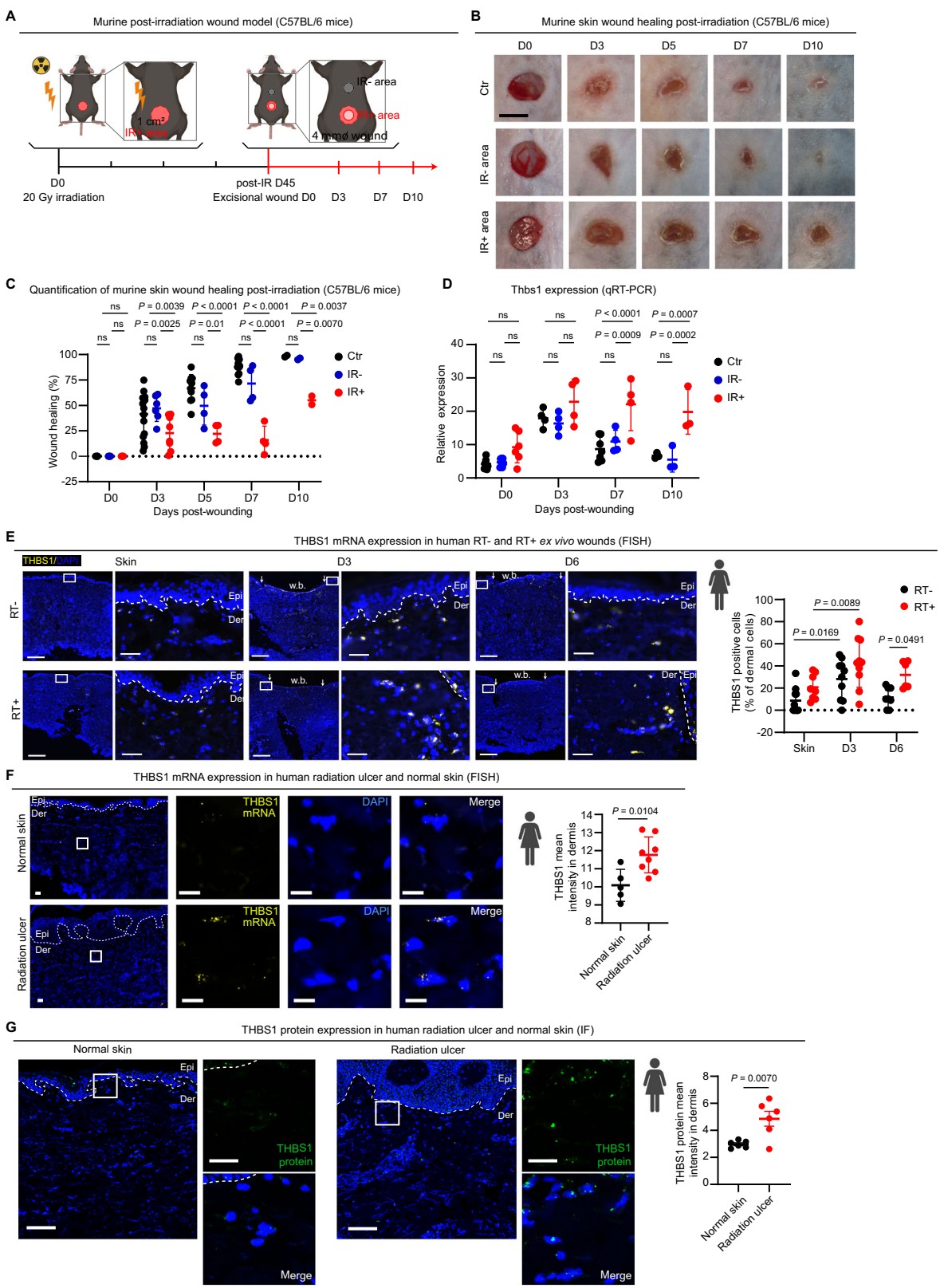

fibroblast expression of ECM genes (COL1A1, COL3A1, FN1, and ELN) and ACTA2, as well as cell migration as shown in scratch wound assays, suggesting that the compromised functions of fibroblasts in the RT⁺ skin may be attributed to the aberrant overexpression of *THBS1* (Fig. 6B–G). Conversely, we silenced *THBS1* expression with siRNAs in both RT⁻ and RT⁺ fibroblasts and then treated the cells with TGF-β to induce their differentiation into myofibroblasts (Fig. 6H). Intriguingly,

*THBS1* silencing significantly (P < 0.01) augmented the expression of the myofibroblast marker *ACTA2* in RT⁺ fibroblasts, while no significant effect was observed in the donor-matched RT⁻ fibroblasts (Fig. 6I). Moreover, we found that reducing *THBS1* expression increased migration in RT⁺ fibroblasts only when treated with TGF-β, indicating TGF-β's role in activating THBS1 and exacerbating functional abnormalities in RT⁺ cells (Fig. 6J, Supplementary Fig. 9B). Therefore,

**Fig. 5 | Aberrant *THBS1* expression in wounds of late irradiated skin.**
**A** Illustration of the murine post-irradiation wound model (C57BL/6 mice).
**B** Representative images of murine skin wound closure post-irradiation (C57BL/6 mice) (Ctr, $n = 38$; IR-, $n = 16$; IR +, $n = 18$ wounds). Scale bars: 4 mm.
**C** Quantification of wound healing at day 0, 3, 5, 7, and 10 post-wounding in the non-irradiated control mice (Ctr), non-irradiated area (IR-), and irradiated (IR+) murine skin (Ctr, $n = 38$; IR-, $n = 16$; IR+, $n = 18$ wounds). **D** qRT-PCR analysis of *Thbs1* at day 0, 3, 7, and 10 post-wounding in the Ctr, IR-, and IR+ murine wound dermal cells (Ctr, $n = 14$; IR- and IR+, $n = 11$ wounds). **E** Representative FISH images and quantification of THBS1 signals in human RT- and RT+ ex vivo wounds 0, 3 and 6 days post-wounding (RT- skin, RT- and RT+ D3, $n = 10$; RT+ skin, RT- D6, $n = 8$; RT+ D6, $n = 6$). Scale bars: 500 μm or 20 μm in the zoom-in area. Representative FISH (**F**) and IF **G** images and quantification of THBS1 in radiation ulcers ($n = 8$ in **F**; $n = 6$ in **G**) and surrounding skin ($n = 5$). Scale bars: 100 μm or 20 μm in the zoom-in area. White rectangles highlight the zoom-in areas, and dotted lines indicate epidermal-dermal junctions. Data are presented as means ± SEM **C**, or as means ± SD (**E–G**). two-way ANOVA **C–E**, or unpaired two-tailed student's t-test (**F**, **G**). ns: not significant. Figure 5A was created with BioRender.com released under a Creative Commons Attribution-NonCommercial-NoDerivs 4.0 International license. Source data are provided as a Source Data file.

the enhanced migratory effect of THBS1 inhibition is more pronounced with TGF-β treatment.

To probe the mechanisms by which *THBS1* overexpression (OE) influences fibroblast functions, we conducted RNA-seq on human fibroblasts with CRISPR/dCas9-SAM activated *THBS1* expression (Supplementary Fig. 9C, Supplementary Data 10). GO analysis of the differentially expressed genes indicated that biological processes like epithelial-mesenchymal transition, TGF-β signaling, inflammation (TNFα and IL-6 signaling), apoptosis, and hypoxia were predominantly enriched among the 138 downregulated genes, while only the interferon-gamma response was enriched among the 45 upregulated genes post THBS1 OE (Supplementary Fig. 9C). Additionally, we evaluated the impacts of key signaling pathways, including p38, JNK, ERK, PKC, STAT3, PI3K, and EGFR, on THBS1 OE effects. Our findings indicated that an ERK inhibitor most effectively counteracted the suppressive influence of THBS1 OE on ECM gene and *ACTA2* expression (Supplementary Fig. 9D, E). Blocking ERK signaling also mitigated the reduced migration of fibroblasts associated with THBS1 OE (Fig. 6K). These results highlighted ERK as a pivotal downstream signal mediating THBS1's effects. Supporting this, our western blotting results showed that THBS1 OE activated ERK signaling, evidenced by increased phosphorylation of ERK1/2 (Supplementary Fig. 9F). Overall, our findings indicate that heightened *THBS1* expression results in significant changes in gene expression and key signaling pathways essential for fibroblast function and wound healing.

Based on this compelling evidence, we investigated whether targeting the aberrant overexpression of *THBS1* could enhance the healing capacity of late-irradiated human skin. To assess this, we combined THBS1 antibody dissolved in PBS with 30% pluronic F-127 gel, using PBS alone with the gel as a control. We then topically applied the gel with or without THBS1 antibody to human ex vivo wounds on RT+ and RT- skin. Initially, we used a high dose of THBS1 antibody (e.g., 66.7 μg/mL) immediately after injury (D0), which blocked wound closure in RT- skin (Supplementary Fig. 9G). We propose that early upregulation of THBS1 is crucial for wound healing, while its persistent expression is detrimental. Therefore, we optimized the THBS1 antibody dose to 0.2 μg/mL and applied it two days post-wounding to block THBS1 at the late stage, but not the early stage, of wound healing. This regimen showed clear pro-healing effects in RT+ skin without affecting RT- skin (Fig. 6L, Supplementary Fig. 9H, I). These findings support the safety profile of the THBS1 blocking antibody, indicating it can specifically target late-irradiated skin without impacting normal skin, provided the treatment dose and timing are optimized.

Given the enhanced regulatory activity of RUNX in RT+ fibroblasts, particularly in driving the expression of post-radiation poised genes, we also investigated RUNX1's role in fibroblasts. Knocking down RUNX1 expression with siRNA significantly increased migration and ECM gene expression (COL3A1 and ELN) in RT+ fibroblasts, but not in RT- fibroblasts (Fig. 6M–P). Additionally, applying this siRNA topically to human ex vivo wounds, along with a transfection reagent, significantly reduced RUNX1 expression compared to wounds treated with control siRNAs (Supplementary Fig. 9K). RUNX1 silencing decreased THBS1 but increased ECM expression (FN1) in RT+ ex vivo

wounds (Supplementary Fig. 9L, M). Importantly, RUNX1 silencing improved healing in RT+ ex vivo wounds without affecting RT- wounds (Supplementary Fig. 9J, N). These findings indicate that RUNX1 knockdown mirrors the effects of THBS1 inhibition.

Together, our findings strongly suggest that the maladaptive radiation memory, especially its key effector THBS1, represents a promising therapeutic target for mitigating the late-onset adverse effects of RT on human skin (Fig. 7).

## Discussion

Radiotherapy, a common cancer treatment, often damages skin and affects postoperative outcomes. Our study compared skin from irradiated (RT+) and non-irradiated (RT-) areas of cancer patients, revealing an altered chromatin state in RT+ dermal fibroblasts, leading to impaired tissue repair. We found that inhibiting THBS1, a gene epigenetically primed in RT+ fibroblasts, accelerates wound healing in RT+ skin. This contributes to understanding LAE pathogenesis and suggests new treatments to mitigate radiotherapy-induced skin toxicity.

This study deepens our understanding of epigenetic memories in two significant ways. Firstly, we demonstrate that dermal fibroblasts, a long-lived and quiescent cell type in human skin, have the capacity to retain epigenetic memories for extended periods. This finding expands the concept of epigenetic memory beyond immune cells and epithelial stem cells, introducing a new cell type into the paradigm[13]. Although fibroblasts have been observed to exhibit memory related to their anatomical locations, their ability to retain long-term memories of other environmental stimuli remains less clear[48]. Secondly, we provide compelling evidence that maladaptive epigenetic memory is implicated in human disease. While epigenetic memory can help us to cope with new threats more efficiently, it is also suspected to play detrimental roles in the pathology of diseases such as chronic inflammatory disorders and cancers[49–52]. Validating this concept in humans poses challenges, as identifying the primary stimuli that establish pathologically relevant memory is complicated by the complexity of our exposome. However, in cancer patients treated with RT, a distinct and well-controlled primary stimulus, we have a unique opportunity to track the long-term impact of radiation memory in humans. Most intriguingly, we demonstrate that the erasure of such maladaptive radiation memory by targeting THBS1 can mitigate the late onset toxicity associated with RT. This discovery has broad implications for leveraging epigenetic memory in disease diagnosis and treatment.

After the primary stimulus, i.e., RT, the *THBS1* promoter undergoes chromatin remodeling and becomes primed, characterized by the presence of H3K4me1 histone modification. This modification is known to persist in epigenetic memory domains long after the stimulus is removed[29]. However, the *THBS1* gene remains transcriptionally inert in skin fibroblasts until cells encounter a secondary stimulus, such as TGF-β1 released during skin injury. In RT+ fibroblasts, the permissive chromatin state of the *THBS1* promoter facilitates faster recruitment of the transcription factor RUNX1, resulting in a more potent and sustained expression of *THBS1*. This aberrant *THBS1* expression adversely affects fibroblast motility and contractility

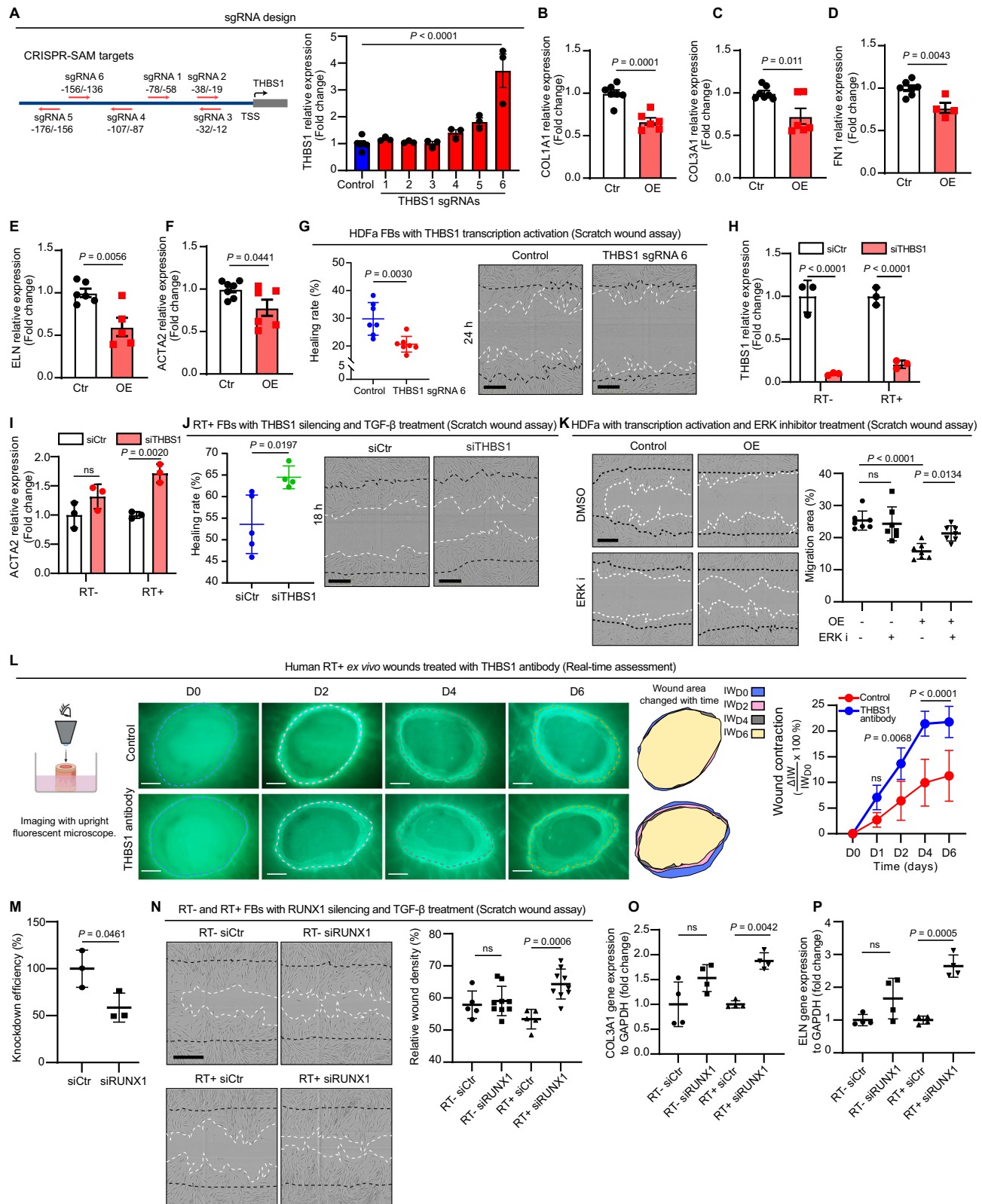

**A** sgRNA design

CRISPR-SAM targets

**G** HDFa FBs with THBS1 transcription activation (Scratch wound assay)

**H**

**J** RT+ FBs with THBS1 silencing and TGF-β treatment (Scratch wound assay) **K** HDFa with transcription activation and ERK inhibitor treatment (Scratch wound assay)

**L** Human RT+ *ex vivo* wounds treated with THBS1 antibody (Real-time assessment)

**N** RT- and RT+ FBs with RUNX1 silencing and TGF-β treatment (Scratch wound assay)

during wound repair, as observed in both in vivo mouse and ex vivo human post-radiation wound models.

A major question still unanswered is how the long-term radiation memory is established in fibroblasts. Upon RT-induced DNA damage, chromatin accessibility and histone modifications change at many genomic loci, creating a more accessible and permissive environment for DNA repair machinery to act on the damaged DNA[53]. While many of these epigenetic changes are transient and integral to the DNA repair process, some persist and manifest as chromatin damage scars[54]. It is plausible to consider that the enhanced accessibility at the *THBS1* promoter represents one such enduring chromatin alteration, contributing to the aberrant response of RT[+] fibroblasts to subsequent

**Fig. 6 | Modulation of THBS1 enhances the healing capacity of late irradiated human skin. A** Illustration of *THBS1* sgRNA design and qRT-PCR analysis of *THBS1* expression in human fibroblasts transfected with CRISPR/Cas9-SAM plasmids ($n = 3$). qRT-PCR of ECM genes **B**–**E** and *ACTA2* (**F**) in fibroblasts with THBS1 overexpression (OE). (**B, C**) Ctr, $n = 7$, OE, $n = 5$; (**D**) Ctr, $n = 7$, OE, $n = 4$; (**E**) Ctr, $n = 6$, OE, $n = 5$; (**F**) Ctr, $n = 7$, OE, $n = 6$. **G** Scratch wound assays of fibroblasts with *THBS1* OE ($n = 7$). Representative images of wounds are shown: black and white dashed lines indicate the wound edges at 0 and 24 hours, respectively. Scale bars: 300 μm. qRT-PCR analysis of *THBS1* (**H**) and *ACTA2* (**I**) in RT⁺ and RT⁻ fibroblasts with *THBS1* silencing and TGF-β1 treatment ($n = 3$). **J** Scratch wound assays of RT⁺ fibroblasts with *THBS1* silencing and TGF-β1 treatment (siCtr, $n = 5$, siTHBS1, $n = 4$). Representative images of wounds 18 hours after scratching are shown. Scale bars: 300 μm. **K** Scratch wound assays of fibroblasts with *THBS1* OE and treated with or

not with ERK inhibitor (U0126). Representative images of wounds 12 hours after scratching are shown ($n = 7$). Scale bars: 300 μm. **L** Representative images of ex vivo wounds on RT⁺ skin treated with or not with anti-THBS1 antibodies (Control, $n = 5$; THBS1 antibody, $n = 4$). Wound contraction was evaluated by measuring the change of areas within the initial wound edges (indicated with dashed lines) on days 0-6 post-injury. Scale bars: 500 μm. qRT-PCR of *RUNX1* **M**), *COL3A1* (**O**), and *ELN* (**P**) in RT⁻ and RT⁺ fibroblast transfected with siRUNX1 and treated with or not with TGF-β, $n = 3$ **M**), $n = 4$ (**O, P**). **N** Scratch wound assays of RT⁻ and RT⁺ fibroblasts transfected with siRUNX1 (RT⁻ or RT⁺ with siCtr, $n = 5$; RT⁻ or RT⁺ with siTHBS1, $n = 9$). Representative images of wounds 10 hours after scratching are shown. Scale bars: 300 μm. Data are presented as means ± SD. One-way ANOVA (**A**), unpaired two-tailed student's t-test (**B**–**G, J, M**), two-way ANOVA (**H, I, K, L, N**–**P**). ns: not significant. Source data are provided as a Source Data file.

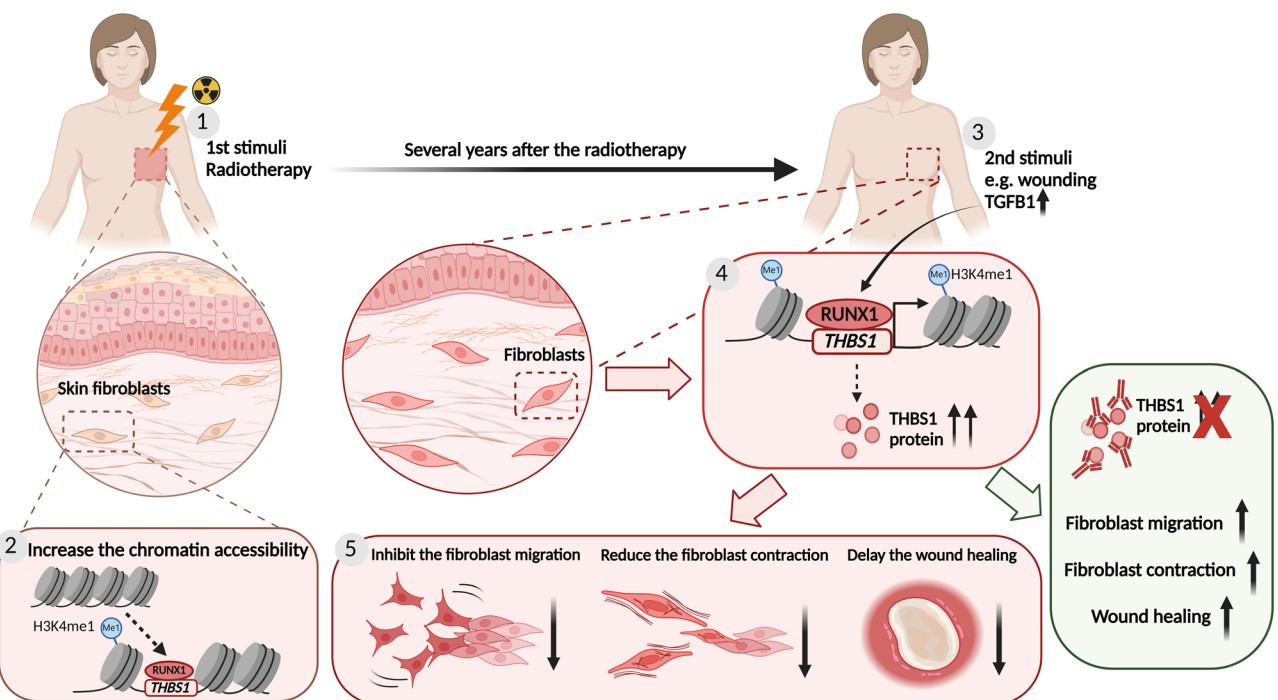

**Fig. 7 | Schematic summary of the study.** Skin fibroblasts in cancer patients who undergo radiotherapy exhibit enduring epigenetic alterations, including heightened chromatin accessibility at the *THBS1* gene locus. Following skin injury, such as during surgery, the TGF-β signaling pathway triggers RUNX1-dependent transcription of *THBS1*. The elevated and sustained expression of *THBS1* in RT⁺

fibroblasts hampers cellular motility, contractility, and delays the healing process. However, the inhibition of THBS1 enhances fibroblast functionality and facilitates tissue repair, indicating a prospective therapeutic approach for addressing radiation ulcers. Figure 7 was created with BioRender.com released under a Creative Commons Attribution-NonCommercial-NoDerivs 4.0 International license.

challenges. Further investigations are required to elucidate the potential mechanistic connections between DNA repair and the plasticity of the epigenome in irradiated human skin over the long term.

Not only in wound repair, THBS1 also plays critical roles in fibroblasts in systemic sclerosis[55], hypertrophic scarring[56], and keloid[57]. In skin fibrosis, *THBS1* expression is upregulated, promoting fibroblast migration, proliferation, and ECM deposition[55–57]. Interestingly, a clinical trial with systemic sclerosis patients showed that THBS1 expression rapidly declined in patients' skin after the treatment with Fresolimumab, a high-affinity neutralizing antibody targeting TGFβ1[58]. Moreover, THBS1 has been shown to enhance cancer cell invasion in head and neck squamous cell carcinoma (SCC) and esophageal SCC[59,60], while inhibiting tumor vascularization and progression in oral and cutaneous SCC[61]. Additionally, elevated THBS1 expression has been observed in irradiated blood vessels compared to non-irradiated ones long after RT exposure[62]. Although the current study focuses on wound repair, the RT-induced epigenetic alteration of the *THBS1* gene may also contribute to other late-onset adverse effects of RT, such as

fibrosis and secondary skin tumors, highlighting the need for further investigation.

Beyond affecting fibroblasts, the elevation of THBS1 in radiation-exposed skin likely exerts a substantial influence on a variety of cellular players and mediators that are crucial for wound healing. We found that THBS1, predominantly produced by fibroblasts, may interact with keratinocytes, angiogenic cells, and immune cells via receptors such as CD47 within human wounds. THBS1 is known to inhibit the proliferation, migration, and survival of vascular endothelial cells, restricting angiogenesis that is essential for wound repair[36]. It also regulates the bioavailability of pivotal growth factors (like bFGF and VEGF) and enzymes (such as MMPs), thereby modulating ECM and various cellular functions[33]. Moreover, THBS1 directly binds and activates latent TGF-β1, indicating that inhibiting THBS1 could offer a therapeutic approach for fibrotic conditions[63]. Notably, disrupting the THBS1-CD47 interaction has been demonstrated to protect normal tissues from the adverse effects of radiotherapy and chemotherapy by fostering protective autophagy and anabolic metabolic repair,

simultaneously enhancing the immune destruction of cancer cells[64]. Given its role as an innate immune checkpoint, CD47 has emerged as a central target in the development of cancer immunotherapies[65]. The growing interest in CD47 inhibitors is driven by their dual potential: enhancing the immune system's response against tumors and protecting healthy tissue during and after radiation therapy. Our findings further reinforce the latter benefit.

Purifying fibroblasts from patient skin is challenging due to their heterogeneity and the lack of universal markers[66]. We primarily used an explant outgrowth approach[20,67,68] and also CD90+ magnetic-activated cell sorting (Fig. 4C)[66]. The explant outgrowth approach is widely accepted for fibroblast isolation, with purity confirmed by marker analysis[20,67,68]. CD90 is also commonly used for sorting viable fibroblasts[66], and our scRNA-seq data show high CD90 expression across all fibroblast clusters in human skin. However, CD90 is also expressed in mesenchymal stem cells and endothelial cells[69–72], and not all fibroblasts express CD90[73]. A FACS-based negative-selection strategy for sorting fibroblasts exists but requires large samples[74]. Since no universally accepted method yields pure fibroblast populations, we initially used the explant outgrowth approach and then confirmed the role of fibroblasts in radiation-driven epigenetic changes through single-cell and imaging data.

In summary, our study reveals that dermal fibroblasts possess a long-term radiation memory manifested as enduring epigenetic alterations, which compromises their ability to respond effectively to new challenges. By functionally blocking THBS1, a gene associated with radiation memory that hampers the wound repair capacity of fibroblasts, we were able to improve wound healing of the previously irradiated patients' skin. These exciting findings highlight the potential of addressing maladaptive radiation memory as a promising approach to prevent and reverse late-onset RT toxicities.

## Methods

### Human RT⁻ and RT⁺ skin sample collection and analysis

The collection and usage of human skin samples were approved by the Stockholm Regional Ethics Committee (Stockholm, Sweden). All participants gave their written consent, and the study was conducted according to the Declaration of Helsinki's principles.

Paired tissue biopsies were collected from surplus skin from breast cancer patients undergoing autologous-tissue breast reconstruction ($n = 46$ donors, Supplementary Data 1) at the Department of Plastic and Reconstructive Surgery at Karolinska University Hospital (Stockholm, Sweden). These patients underwent a mastectomy followed by external beam RT with a total dose of 40–60 Gy high-energy X-rays. The length of the period from the end of the RT to the breast reconstruction surgery varied from one to 12 years among the patients. During the reconstruction surgery, previously irradiated skin was collected from one side of the breast (RT⁺) and non-irradiated skin (RT⁻) from the other side of the breast or abdomen from each patient (Supplementary Data 1). We also collected matched skin samples from three breast cancer survivors: areas with surgery 9–25 months prior but no radiotherapy (S+, abdomen) and areas without surgery and radiotherapy (S-, breast) (Supplementary Data 1). Dermal fibroblasts were isolated from these RT⁻/RT⁺ and S−/S+ skin biopsies.

Moreover, to monitor the in vivo gene expression changes of human skin wound healing, we developed a human wound healing model at the Karolinska University Hospital. We created three full-depth wounds, each 3 mm in diameter and extending into the subcutaneous adipose tissue, on the skin of healthy volunteers ($n = 18$ donors, Supplementary table 1) using a biopsy punch with a circular blade. Post-wounding, the donors returned to our clinic on three occasions: day one (D1), day seven (D7), and day 30 (D30). During each visit, we used a 6 mm biopsy punch to harvest wound-edge tissue, selecting a different wound at each time point for collection.

### Post-irradiation wound model

The protocols of murine experiments were approved by the Comité de Protection des Animaux de l'Université Laval (CPAUL), Cégep de Sainte-Foy Animal Protection Committee (Québec, Canada), and the North Stockholm Ethical Committee for Care and Use of Laboratory Animals (Stockholm, Sweden). All animals were housed in accordance with the procedures delineated in the Guide for the Care and Use of Laboratory Animals. Animals were maintained in a 12 hour light/dark cycle and were provided with food and water. Animals were assigned randomly to experimental groups.

**Post-irradiation wound model using CD-1 mice.** The protocol of murine irradiated skin and excisional wound model is detailed in ref. 45. On the day of irradiation, the animals were anesthetized with ketamine (80 mg/kg) - xylazine (8 mg/kg) - acepromazine (1 mg/kg). Unique doses of 6MV photons were delivered to the back skin of CD-1 mice (45, 60, or 80 Gy). Skin toxicity recovered over four weeks, at which time 8 mm-biopsy punches were used to create full-thickness excisional wounds. Non-irradiated control animals were also included in the wound healing study. The wounds were splinted with silicone rings and allowed to heal under moist conditions[45]. After 33 days, the wounded tissues were harvested for histological analyses. Formalin-fixed samples embedded in paraffin were used for FISH analysis.

**Post-irradiation wound model using C57BL/6 mice.** Mice were anesthesia with 1.5% isoflurane (Cat. 002185, Zoetis, UK) by inhalation. We performed focal irradiation on the shaved back skin of male C57BL/6 mice using XStrahl CIX3 irradiator (Xstrahl, Georgia) at Karolinska Institutet. Murine skin was exposed to 20 Gy irradiation with a 0.5 mm copper filter to maximum reduce the irradiation penetration through the skin. The irradiation area is 1 cm² using a collimator. While 20 Gy caused no visible skin damage in CD-1 mice, it induced radio-dermatitis in C57BL/6 mice[75,76]. Body weight was measured at every check to monitor the systemic effects of irradiation. We monitored the acute radiation effects (erythema, desquamation, ulceration, evaluated with RTOG scores[46]) appearing around 5 days and peaking at 14 days post-irradiation (IR), with macroscopic recovery by 37 days. We created excisional full-thickness wounds (4 mm in diameter) at both irradiated (IR⁺) and non-irradiated (IR⁻) sites 45 days post-IR, once acute effects had subsided, and also on non-irradiated control mice (Ctr). During the first two days, the mice were received s.c. buprenorphine (0.03 mg/kg) twice a day for relieving pain and distress caused by the wounding. We monitored wound closure by taking images of wounds on day three, day five, day seven, and day ten. The wound closure (%) was quantified as wound area$_{time point}$/wound area$_{D0}$ × 100%. Mice were euthanized with a $CO_2$ fill rate of 30-70% of the chamber volume, and skin biopsies at wound sites and intact areas were collected for qRT-PCR and ATAC-seq analysis.

### Fluorescence assisted cell sorting (FACS)

Four-millimeter punch biopsies were collected from Pdgfra-H2Be-GFP male mice one and seven days post-irradiation and from non-irradiated controls. After separating the epidermis and dermis using 5 U/mL dispase II solution (Cat. 17105041, Gibco), the dermis was cut into small pieces and dissociated into single-cell suspensions using a human enzyme mixture from a whole skin dissociation kit (Cat. 130-101-540, Miltenyi Biotec). The isolated dermal cells were resuspended in 300 μL FACS buffer (PBS containing 10% FBS). The single-cell suspension was incubated with SYTOX™ Blue Dead Cell Stain (1:10,000 dilution in PBS, Cat. S34857, Invitrogen) for 5 minutes at room temperature. FACS was performed using a BD FACSAria Fusion Sansa (BD Biosciences), and cells were collected into FACS buffer (Supplementary Fig. 10).

## Human ex vivo wound model

Human ex vivo wound model was performed as previously described[16–18]. We utilized a biopsy punch with a circular blade (2 mm in diameter) to create partial-thickness wounds, ensuring that the wounds did not extend below the dermis layer, on RT⁻ and RT⁺ human skin collected post-surgery. These wounds were excised from the skin using a 6 mm biopsy punch. After the subcutaneous fat was removed, the excised wound tissues were then placed into a 12-well cell culture plate. The Dulbecco's Modified Eagle Medium high glucose, DMEM (Cat. 11965092, Gibco, Waltham, MA) supplemented with 10% fetal bovine serum (Cat. 2567819RP, Gibco), and antibiotics (1x penicillin and streptomycin, Cat. 15140122, Gibco) was added (800 μl per well) around the tissue, so the epidermal surface was exposed to the air to create a liquid-air interface and cultured at 37 °C in a humidified atmosphere of 5% $CO_2$.

THBS1 antibody (Cat. MA5-13377, Invitrogen) was diluted in PBS to concentrations of 66.7 μg/mL or 0.2 μg/mL. This antibody mixture was then combined with 30% pluronic F-127 gel (Cat. P2443, Sigma-Aldrich, St Louis, MO) in a 1:2 volume ratio. As a control, PBS without THBS1 antibody was mixed with 30% pluronic F-127 gel in the same ratio. Five μL of the THBS1 antibody mixture or control mixture was applied topically to wounds immediately after injury or two days post-wounding. The treatment was repeated every two days until day 6. 0.1 μg of siRUNX1 (Cat. L-003926-00-0005, Dharmacon) or non-targeting siRNAs (Cat. D-001810-0X, Dharmacon), along with in vivo-jetPEI (Cat. 201-10 G, Polyplus-transfection, France), were separately diluted in 2.5 μl of a 5% glucose solution and then combined. This mixture was topically applied to ex vivo wounds immediately post-injury and again on day two and four. Wound samples were collected five or six days after injury for histological analysis.

We monitored ex vivo wound closure with CellTracker™ Green CMFDA Dye (Cat. C2925, Invitrogen, Waltham, MA) as previously described[17]. Briefly, 4 μl dye (50 μM) was added to each wound tissue and incubated at 37 °C with 5% $CO_2$ for 30 minutes. The tissue was washed with PBS and imaged with a Nikon eclipse Ni-E fluorescence microscope. The wound areas were quantified by using ImageJ. The initial wound edges were demarcated with dashed lines. The areas of fluorescent staining inside these dashed lines represent newly formed epithelial tongues, indicative of the re-epithelialization process[17]. Wound contraction was measured by assessing the changes of regions within the initial wound edge ($IW_{time\ point}$) over time, i.e., wound contraction (%) = $\Delta IW_{time\ point}/IW_{D0} \times 100\%$.

## Cell isolation and culture

We isolated RT⁺ and RT⁻ fibroblasts from the patient skin using an explant outgrowth approach[20]. In brief, the skin was cleansed with 70% ethanol and PBS, followed by taking 6 mm punch biopsies and removing the adipose tissue. Biopsies were rinsed in PBS, incubated overnight in 5 U/mL dispase II (Cat. 17105041, Gibco, Waltham, MA) at 4 °C, and the dermis was then separated from the epidermis and sectioned. The dermal pieces were placed in a culture dish with fibroblast growth medium (DMEM, 10% fetal bovine serum, antibiotics), changing the medium every three days until 70% confluency. Cells were then trypsinized and passaged, using only those propagated for ≤ 5 passages in experiments.

## In vitro irradiation of fibroblasts

Human dermal fibroblasts were irradiated at room temperature for 8 Gy of radiation using the XStrahl CIX2 irradiator (Xstrahl, Georgia) at Karolinska Institutet or the Scandritronix radiator (Scanditronix, Vislanda, Sweden) at Stockholm University. The Scandritronix radiator was equipped with a 137Cs source (activity 33.3 TBq as of June 1985). Three hours, one day, and six days after the irradiation, cells were treated with 5 ng/mL recombinant human TGFβ1 (Cat. 11343161, ImmunoTools, Friesoythe, Germany) for 24 hours.

## Ex vivo explant migration assay

The ex vivo explant migration assay was performed as previously described[77]. Briefly, 3 mm punch biopsies were collected from the paired human RT⁻ and RT⁺ skin, and the dermis was separated from the epidermis as described above. Dermis pieces were placed in cell culture plates and overlaid with fibroblast growth medium. Fibroblast outgrowth was analyzed ten days later by measuring the length from the migrated edge to the tissue (average of three measurements per sample).

## Cell migration assays

Cells were plated on ImageLock 96-well plates (Cat. 4379, Essen Bioscience, Ann Arbor, MI) and adhered overnight. To inhibit cell proliferation, we treated cells with 5 μg/mL mitomycin C (Cat. J63193.MA, Thermo Scientific Chemicals, Waltham, MA) for 2 hours, and the cell monolayer was scratched using the IncuCyte wound maker (Essen BioScience). Cells were imaged every 2 hours using the IncuCyte ZOOM imaging system, and cell migration was quantitated with Incu-Cyte ZOOM 2018A software (Essen BioScience) or ImageJ software (Bethesda, Maryland).

## Cell proliferation assays

**CyQuant assay.** RT⁻ and RT⁺ fibroblasts were seeded in 96-well plates (Cat. 83.3924, Sarstedt) and allowed to attach for 12 h. Cell proliferation was assessed by fluorometric quantification of DNA using CyQUANT Proliferation Assay Kit (Cat. C7026, Invitrogen) according to the manufacturer's instructions. *InCucyte live-cell imaging*: RT⁻ and RT⁺ fibroblasts were plated on ImageLock 96-well plates (Cat. 4379, Essen Bioscience, Ann Arbor, MI) and adhered overnight. Cells were imaged every two hours using the IncuCyte ZOOM imaging system, and cell proliferation was quantitated with IncuCyte ZOOM 2018A software (Essen BioScience).

## siRNA transfection

Fibroblasts at 70% confluence were transfected with a 60 nM predesigned siRNA targeting *THBS1* (siTHBS1, Cat. s14100, Invitrogen) or control siRNAs (siCtr, Cat. AM4611, Invitrogen); 20 nM predesigned siRNA targeting RUNX1 (siRUNX1, Cat. L-003926-00-0005, Dharmacon) or ON-TARGETplus non-targeting siRNAs (Cat. D-001810-0X, Dharmacon) with Lipofectamine™ RNAiMAX Transfection Reagent (Cat. 13778075, Invitrogen). Six hours post-transfection, the medium was switched to fibroblast growth medium for an additional 24 hours.

## CRISPR-mediated transcriptional activation

To mediate efficient transcriptional activation at endogenous genomic *THBS1* loci, we used a CRISPR/Cas9 Synergistic Activation Mediator (SAM) system. The core component of this system is comprised of two plasmids, lentiMS2-P65-HSF1_Hygro (Plasmid #61426, Addgene, Cambridge, Massachusetts) and lentiSAM v2 (Puro) (Plasmid #92062, Addgene). Six single guide RNAs (sgRNAs) were designed using an online tools CRISPOR: http://crispor.tefor.net/ and CHOPCHOP: http://chopchop.cbu.uib.no/. Individual sgRNA expression plasmid was constructed by annealing the oligonucleotides pair and then ligating them to a BsmBI-v2 (Cat. R0739S, New England Biolabs, Ipswich, MA) digested lentiMS2-P65-HSF1_Hygro backbone. Primers were designed using an online tool Benchling (https://benchling.com/). The sequences of sgRNAs and primers are listed in Supplementary table 4. Human dermal fibroblasts, adult (HDFa, Cat. C0135C, Gibco) were transfected with Lipofectamine™ 3000 Transfection Reagent (Cat. L3000008, Invitrogen) and harvested 48 hours post-transfection for qRT-PCR analysis of THBS1.

To further investigate the potential impacts of the key signaling pathways, the transfected HDFa were treated with a variety of signaling inhibitor for 24 hours, including 10 μM p38-inhibitor (SB203580, Merck), 10 μM JNK-inhibitor (SP600125, Santa Cruz Biotechnology),

10 µM ERK-inhibitor (U0126, Calbiochem), 100 nM PKC-inhibitor (Ro-31-8220, Santa Cruz Biotechnology), 5 µM STAT3-inhibitor (WP1066, Calbiochem), 1 µM PI3K-inhibitor (Wortmannin, W1628, Calbiochem), and 500 nM EGFR-inhibitor (PD153035, Calbiochem).

## Protein extraction and Western blot

Fibroblast protein lysates were extracted using radio-immunoprecipitation assay (RIPA) buffer (Cat. 89900, Thermo Scientific) supplemented with protease inhibitor. Protein concentrations were measured using the BCA Protein Assay Kit (Cat. 23252, Thermo Scientific). The total protein was separated in TGX precast protein gels (Bio-Rad), then transferred onto a nitrocellulose membrane. Blots were probed with rabbit phosphoERK1/2 antibody (1:1000, Thr202/Tyr204, 197G2, Cat. 4377S, Cell Signaling Technology) and rabbit ERK1/2 antibody (1:1000, Cat. 06-182, EMD Millipore). Thereafter, the blots were incubated with anti-rabbit HRP-conjugated secondary antibodies (P0447, 1:5000 dilution, DAKO). β-actin expression was visualized by using an HRP-coupled anti-human actin antibody (1:10000, Cat. A3854, Sigma-Aldrich, St.Louis, MO). Protein band densities were quantified using Image lab software (Bio-Rad).

## Masson's trichrome staining and imaging

Tissue samples were fixed in 4% paraformaldehyde (Cat. HL96753.1000, HistoLab, Askim, Sweden) overnight at 4 °C, placed in 70% ethanol, dehydrated and embedded in paraffin, and cut in 8 µm thickness. Tissue sections were stained by Trichrome Stain (Masson) Kit (Cat. HT15, Sigma-Aldrich) and imaged using a Nikon eclipse Ni-E bright field microscope (Tokyo, Japan).

## Collagen fibre alignment by FIBRAL analysis

We utilized the collagen alignment tool FIBRAL to analyze RT⁻ and RT⁺ histological images. Using a combination of L*a*b color conversion, image enhancement and Fourier domain analysis, a single orientation metric known as the alignment coefficient can be formulated (0-1, with 0 representing a random distribution and 1 correlating to a perfectly aligned case). FIBRAL was modified in this paper for applications to Masson's trichrome staining images. To isolate the collagen fibers from each image, the full a-channel was superimposed onto the -ve portion of the b-channel. To quantify the density of collagen fibers in each image, a simple pixel area algorithm was employed. Using the enhanced grayscale image exported through FIBRAL, the total area of fibrous tissue was represented by the fraction of the image with non-zero pixel values. In this system, a low alignment index indicates the presence of fibers in a random orientation, typical of non-scarred skin, whereas a high alignment index suggests fibers are arranged in the collagen structure in a linear pattern, as seen in scarring.

## Fluorescence In situ hybridization (FISH)

In situ hybridization probes for human and mouse *THBS1* (Hs-THBS1, Cat No. 42658 and Mm-Thbs1, Cat No. 457891) were designed and synthesized by Advanced Cell Diagnostics (ACD, Silicon Valley, CA). Tissues were prepared by following the manufacturer's instructions. After paraffin removal, the slides were incubated in hydrogen peroxide, target retrieval reagent, and protease plus (ACD), following incubation with hybridization probes for two hours at 40 °C in HybEZ™ II Hybridization System using RNAscope® Multiplex Fluorescent Reagent Kit v2 (Cat. 323100, ACD). The hybridization signals were amplified via sequential hybridization of amplifiers and probes. Probe signals were visualized on Zeiss AxioScan.Z1 Slide Scanner (Oberkochen, Germany) and analyzed with Zen 3.4 software (Zeiss).

To visualize the expression of *Thbs1* mRNA and Pdgfra protein in the murine skin and wounds post-irradiation, mRNA and protein co-detection was performed by using RNA-Protein Co-Detection Ancillary kit (Cat. 323180, ACD). Briefly, after paraffin removal, the slides were incubated in hydrogen peroxide and target retrieval reagent, followed by anti-Pdgfra antibody (5 µg/mL, cat. AF1062, R&D) incubation at 4 °C for overnight. After the second fixation with 10% neutral formalin buffer and Protease Plus treatment, the sections were stained using the RNAscope® Multiplex Fluorescent protocol. Finally, an Alexa Fluor 647 conjugated secondary antibody was applied to visualize Pdgfra, with results observed using the Zeiss AxioScan.Z1 Slide Scanner (Oberkochen, Germany).

## Immunofluorescence staining

Paraffin-embedded tissue sections were deparaffinized and rehydrated by passage through xylene and graded ethanol series. After antigen retrieval in citric acid buffer (10 mM, pH 6.0), sections were blocked with 5% bovine serum albumin (BSA, Cat. 9414, Sigma-Aldrich) in Tris-buffered saline with 0.1% Tween-20 (TBST). Sections were incubated overnight at 4 °C with primary antibody targeting THBS1 protein (1:100 dilution, cat. sc-59887, Santa Cruz Biotechnology, Dallas, TX, or 1:100 dilution, cat. ab267388, Abcam, Cambridge, UK), or anti-Pdgfra protein (10 µg/mL, cat. AF1062, R&D), or anti-Phospho-SMAD2 (Ser465, Ser467) protein (1:50, cat. 44-244 G, Thermofisher), followed by incubation with Alexa Fluor 555, 647 conjugated secondary antibody (Cat. A-31570, A-31572, A-21447, Invitrogen) in 1:1000 dilution in 1% BSA buffer. Sections were counter-stained with ProLong™ Diamond Anti-fade Mountant with DAPI (Cat. P36971, Invitrogen). Immuno-fluorescence staining was visualized using a Nikon eclipse Ni-E fluorescence microscope or Zeiss LSM900-Airy Confocal microscope.

## Chromatin immunoprecipitation (ChIP)

RT⁻ and RT⁺ fibroblasts were treated with 5 ng/mL recombinant human TGFβ1 (Cat. 11343161, ImmunoTools) in the fibroblast culture medium for 24 hours. Cells were crosslinked with 1% formaldehyde (Cat. 28908, Thermo Scientific) for 10 minutes and quenched with 0.125 M glycine (Cat. 50046, Sigma-Aldrich). MAGnify Chromatin Immunoprecipitation System kit (Cat. 492024, Applied Biosystems) was used for ChIP according to the manufacturer's instructions. Briefly, 200,000 cells per sample were collected and lysed, following DNA sonication to achieve 200-500 bp fragments using Bioruptor UCD-200 (Diagenode, Seraing, Belgium). Protein A/G Dynabeads were mixed with RUNX1-targeting antibody (2.5 µg per reaction, Cat. ab272456, Abcam) or H3K4me1 (2 µg per reaction, Cat. ab8895, Abcam). Sonicated cell lysates were incubated with the antibody-coated beads for two hours at 4 °C, followed by washing, reverse crosslinking, and DNA purifying. Samples were analyzed by qPCR with primers designed to span the RUNX1-binding sites at the THBS1 promoter region (Supplementary table 4).

## Magnetic activation cell sorting

Fibroblasts were isolated from human skin and acute wound tissues with magnetic activation cell sorting (MACS). Fresh tissue samples were washed 2–3 times in PBS and incubated in 5 U/mL dispase II solution (Cat. 17105041, Gibco) supplemented with antibiotics (1x penicillin and streptomycin, Cat. 15140122, Gibco) overnight at 4 °C. The epidermis was separated from the dermis. The dermis was incubated in the enzyme mix from the whole skin dissociation kit (Cat. 130-101-540, Miltenyi Biotec) for 3 hours and further processed by Medicon tissue disruptor (BD Biosciences, Stockholm, Sweden). The dermal cell suspension was incubated with CD90 microbeads (Cat. 130-096-253, Miltenyi Biotec), and CD90⁺ fibroblasts were isolated with MACS MS magnetic columns according to the manufacturer's instructions (Miltenyi Biotec)[66]. The isolated fibroblasts were used for qRT-PCR analysis directly without cell culture.

## RNA extraction and qRT-PCR

Total RNA was extracted from fibroblasts using Trizol, followed by cDNA synthesis with RevertAid First Strand cDNA Synthesis Kit (Cat. K1621, Thermo Scientific, Waltham, MA). Specific premixed primers

and probes were predesigned by Integrated DNA Technologies (IDT, Leuven, Belgium) for the detection of *THBS1, ACTA2, FN1, ELN, CDKN1A, PCNA, COL1A1, COL3A1, RUNX1, 18S, GAPDH, Thbs1, Cdkn1a, Gapdh*, and *Actb*. Gene expression was determined by TaqMan expression assays (Cat. 4304437, ThermoScientific) or SYBR™ Green master mix (Cat. 4367659, ThermoScientific) and normalized based on the values of the housekeeping gene *GAPDH, 18S, Gapdh, or Actb*. The comparative 2ΔΔCT method was used for the quantification of gene expression. All reactions were run by QuantStudio 6 or 7 (Applied Biosystems, Waltham, MA). Information for all the primers used in this study is listed in Supplementary table 4.

### ATAC-seq library preparation, sequencing, and analysis

ATAC-seq was performed as previously described[78]. Briefly, 50,000 cells per sample were subjected to nuclei extraction and library preparation. The libraries were sequenced by Illumina NovaSeq SP-100 (2 x 50bp) or NovaSeq X Plus (2×150 bp) at the National Genomics Infrastructure at SciLifeLab Stockholm.

Raw data were processed using Trimmomatic v0.36[79] by removing reads of low quality and with a length of fewer than 30 nucleotides. The PCR duplicated reads were excluded using Picard (v2.20.4) tools after mapping to GRCh38 human or GRCm39 mouse reference genome using Bowtie2 (v2.3.5.1)[80]. The uniquely mapped reads were shifted, and read coverages were then normalized with a method of RPKM and converted into bigWig format for IGV visualization[81] using the deepTools (v3.3.2)[82] with bamCoverage function. Peaks were called on each sample individually using MACS2 (v2.2.6) with the default parameter except for using the BAMPE option[83]. Peaks overlapping with the repetitive regions in ENCODE blacklist downloaded from UCSC Table Browser were filtered out using the BEDTools suite (v2.29.2)[84]. Peaks were further extended to 500 bp windows centered on the summits, avoiding the bias of differential accessibility (DA) analysis due to the varying lengths of peaks. The peaks were annotated using the ChIPseeker package (v1.40.0) with a promoter region ranging from -3K (upstream) to 3 K (downstream) bp of the transcription starting site[85]. DA peaks in RT⁻ and RT⁺ fibroblasts from patients and IR⁺ and IR⁻ fibroblasts from mice were analyzed using DESeq2 (v1.44.0)[86]. DA domains were defined as an adjusted *p*-value < 0.05. RT⁺ up domains were defined as log2(fold change) >0 and an adjusted *p*-value < 0.05 in RT⁺ fibroblasts. RT⁻ up domains were defined as log2(fold change) <0 and an adjusted *p*-value < 0.05 in RT⁺ fibroblasts. For the ATAC-seq of in vitro irradiated human fibroblasts, library sizes were normalized using the 'DGEList' and 'calcNormFactors' functions from the edgeR package (v 3.40.2)[87], then with 'quantile' normalization from the Limma package (v 3.54.2)[88]. DA peaks were analyzed with t-test or edgeR[87]. Irradiation-induced peaks were defined as *p*-value < 0.05, log2foldchange > 0, and normalized readout counts > 2. Gene ontology (GO) analysis for peak-related genes was performed using Metascape[89], showing significant GO terms with *p*-value < 0.05. For motif analysis of ATAC peaks, the HOMER (v4.11) function findMotifsGenome was used with default parameters to identify enriched sequence motifs matching known transcription factor (TF) binding sites[22]. TOBIAS software version 0.14.0 with functions of ATACCorrect, FootprintScores, and BINDetect, was used to predict differential TF binding scores between RT- and RT⁺ fibroblasts with a bound *p*-value = 0.01[23]. TF-TF gene regulation networks were created using TOBIAS's 'CreateNetwork' function.

### RNA-seq library preparation and data analysis

After RNA extraction and library construction, the libraries were sequenced on an Illumina Hiseq 4000 platform or DNBSEQ, and 150-bp paired-end reads were generated for the following analysis. Raw sequencing reads were trimmed for adaptors using Trimmomatic[79]. Clean reads were mapped to the human reference genome

(GRCh38.p13), coupled with the comprehensive gene annotation file (GENCODEv34) using hisat2[90]. Gene expression was then quantified by calculating unique mapped fragments to exons by using the feature count from the Subread package (v2.0.0)[91]. Differential expression genes (DEGs) were analyzed across different conditions with DESeq2 test and two-way ANOVA test[79].

### Single-cell RNA library preparation, sequencing, and analysis

After separating epidermis and dermis of the skin or wound samples using 5 U/mL dispase II solution (Cat. 17105041, Gibco), we digested the epidermis in 0.025% trypsin-EDTA (Gibco) for 15 minutes at 37 °C, which was quenched with defined trypsin inhibitor (Cat. R007100, Gibco), and strained through a 70 μm filter. Red blood cells and dead cells were removed by red blood cell lysis solution kit (Cat. 130-094-183, Miltenyi Biotec, Bergisch Gladbach, Germany) and a dead cell removal kit (Cat. 130-090-101, Miltenyi Biotec), respectively. Dermis was cut into small pieces and further dissociated into single-cell suspension using a human enzyme mixture from a whole skin dissociation kit (Cat. 130-101-540, Miltenyi Biotec). Epidermal and dermal cells were combined in a 1:1 ratio, and libraries were constructed using a 10x chromium system with chemistry v3. Libraries were then sequenced with the Illumina NovaSeq 6000 sequencer to generate 150-base pair paired-end reads. Raw single-cell sequencing data were processed using the standard 10X Cell Ranger (v5.0.1) analysis workflow, including demultiplexing, aligning to the GRCh38 human reference genome, barcode counting, and unique molecular identifier (UMI) quantification. The doublets of cells predicted by Scrublet (v0.2.3)[92] and DoubletFinder (v2.0.4)[93] were excluded. The clean filtered feature barcode matrices were used as input into a Seurat pipeline[94]. Within the Seurat, we removed mitochondrial genes, hemoglobin genes, ribosomal genes, genes expressed in less than ten cells, and cells with less than 500 detected genes, less than 1000 UMIs, and with more than 20% mitochondrial gene expression. Finally, 16,098 cells from day one post-wounding were retained for all the subsequent analyses. The data were first normalized using the SCTransform (v0.4.1)[95] function. Uniform manifold approximation and projection (UMAP) plots were generated using the 'RunUMAP' function with the first 40 harmonies. The clusters were obtained using the FindNeighbors and FindClusters functions with a resolution of 0.8. The cluster marker was identified using the function 'FindAllMarker'. The cell types were annotated according to the overlaps between the cluster markers and well-known signature genes of each cell type from previous studies. The ligand-receptor (L-R) analysis was performed by the CellChat package[44] to access the potential cell-cell crosstalk among different cell types.

### Single-cell multi-omics gene expression and ATAC sequencing

For single-cell isolation, we used the Whole Skin Dissociation Kit – human (MACS CAT. 130-101-540). Nuclei extraction followed 10X Genomics protocol, and libraries were prepared using their Single Cell Multiome ATAC + Gene Expression protocol. Sequencing was done on Illumina NovaSeq X 300 (2 x 150 bp) at the National Genomics Infrastructure, SciLifeLab, Stockholm.

Data mapping employed 10X Cell Ranger ARC (v2.0.2) with a GRCh38 human reference genome and GENCODE v44 annotation. We used scDblFinder (v1.15.3) to identify doublets and processed data via Signac (v1.12.0)[96] and Seurat (v5.0.1)[97] workflows. Cells with fewer than 1000 or more than 100,000 ATAC fragments, and fewer than 1000 or more than 25,000 gene counts were filtered, as were cells with nucleosome enrichment more than 2 or transcriptional start site enrichment less than 1. A total of 7,299 cells from RT⁻ skin and 6,748 cells from RT⁺ skin were used for analyses. Gene expression was normalized with SCTransform[95], with PCA and UMAP (dim = 1:25) for dimensionality reduction. Peaks within each dataset were identified using MACS2[98]. The ATAC sequencing data was normalized with term-frequency inverse-document-frequency (TFIDF) and was subjected to

dimensional reduction by singular value decomposition (SVD) of the TFIDF matrix and UMAP embedding (dim = 2:14) with RunSVD and RunUMAP functions, respectively. Cluster markers were identified using the 'FindAllMarker' function. Cell types were annotated based on the overlaps between cluster markers and well-established signature genes for each cell type from prior studies.

## Spatial transcriptomics (ST)

Human skin and wound tissues were gently washed with cold PBS and embedded in optimal cutting temperature compound (OCT, Cat. 4583, Sakura Finetek USA, Torrance, CA) and snap-frozen on dry ice. The samples were then processed for the ST experiment by the Visium Spatial platform of 10x Genomics as per the manufacturer's instructions. Cryosections were cut and mounted onto the ST arrays and stored at a − 80 °C freezer. The tissue was dehydrated and stained with haematoxylin and eosin staining to assess the morphology and quality. After permeabilization, reverse transcription and second-strand synthesis were performed on the slides. cDNA Library preparation, clean up, and indexing were conducted following standard procedures. The pooled libraries are sequenced on NovaSeq6000 S4-200 (Illumina), generating ~300 M reads per section. The raw ST data were processed using the standard Space Ranger pipeline (version 1.2) with the GRCh38 human reference genome and GENCODE v38 gene annotations and visualized by BBrowser (BioTuring). Spatial deconvolution analysis were performed with 'cell2location.models' from Cell2location package[43] to train and project the cell types of scRNA-seq into the spatial Visum data with parameters: N_cells_per_location=20, detection_alpha=20. The deconvoluted cell types in each spot were plotted using the top 5% quantile of cell abundance of posterior distribution.

## Analysis of published single-cell RNA-seq data

The FASTQ files for mouse skin samples at day 4 and day 7 post-wounding were retrieved from GSE188432[40]. Additionally, FASTQ files for mouse skin samples at day 1, day 3, day 7, day 14, and day 27 post-wounding were obtained from (https://zenodo.org/records/10013141)[39]. The scRNA-seq data were processed using a standard Seurat pipeline, as previously described. The subcluster of cells with high Pdgfra expression was considered as fibroblasts. Violin plots and dot plots were generated using the 'VlnPlot' and 'DotPlot' functions.

## Statistical information

The number of biological replicates used in each experiment is indicated in the respective method sections and figure legends. Comparison between groups was performed using paired or unpaired student's t-test, one-way analysis of variance (ANOVA), or two-way ANOVA. The two-tailed Mann-Whitney U test was used as the significance test for cell counts in IF staining of murine skin and wounds post-irradiation. Correlation analysis was performed by using Pearson's correlation test. The Wald test was used for differential analysis (Supplementary Data 2–4, 6–10). A Hypergeometric test was used for enrichment analysis (Supplementary table 2, Supplementary table 3, Supplementary Data 5). Differences were considered statistically significant when $P < 0.05$. Statistical analysis was performed using GraphPad Prism software version 9 (San Diego, CA).

## Reporting summary

Further information on research design is available in the Nature Portfolio Reporting Summary linked to this article.

# Data availability

The raw and processed sequencing data generated in this study have been deposited in the Gene Expression Omnibus (GEO) database: ATAC-seq data (GSE254753), RNA-seq data (GSE254756), and single-cell multiome ATAC + RNA-seq data (GSE254758). Bulk RNA-seq data, single-cell RNA-seq data, and spatial transcriptomics data of human skin and wound samples have been published and can be accessed via GSE174661[28]. For the single-cell RNA-seq and spatial transcriptomics analysis of human skin and wound samples, the gene count data can be accessed via GSE241132 and GSE241124 at the GEO database; the raw sequencing data has been deposited at the European Genome–phenome Archive (EGA) under the accession code EGAD50000000813 and is under controlled access due to patient privacy concerns. Specifically, data is available for academic non-commercial research purposes only and is subject to review of a project proposal by a data access committee, entering into an appropriate data access agreement and subject to any applicable ethical approvals. A response to the request for access is typically provided within 10 working days after the committee has received the relevant project proposal and all other required information. The access to the data will expire on the third anniversary of the effective date of the agreement. Single-cell RNA-seq data of mouse skin and wound samples have been published and can be accessed via GSE188432[40] and from https://zenodo.org/records/10013141[39]. Source data are provided with this paper.

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

## Acknowledgements

We thank all the tissue donors participating in this study. We appreciate technical support from Helena Griehsel and Britta Krynitz (Karolinska University Hospital), as well as Cindy J Hayward, Josée Langevin, and Josée Galarneau (Université Laval/Centre de Recherche sur le Cancer). The computations/data handling was enabled by resources in projects sens2020010 and SNIC2019/8-262 provided by the Swedish National Infrastructure for Computing (SNIC) at UPPMAX, partially funded by the Swedish Research Council through grant agreement no. 2018-05973. This study was supported by Swedish Research Council grant (No. 2020-01400, to N.X.L.), Ragnar Söderbergs Foundation (Grant No. M31/15, to N.X.L.), Welander and Finsens Foundation (Hudfonden) (to N.X.L.), Ming Wai Lau Centre for Reparative Medicine (to N.X.L.), LEO Foundation (No. LF-OC-22-001035 and LF-AW_EMEA-20-400022, to N.X.L.), Cancerfonden (No. 20 0930 Pj, to N.X.L.), Karolinska Institutet (to N.X.L.), Cancer Research Funds of Radiumhemmet (No. 161072, to M.H.), the Swedish Society of Medicine (No. SLS-886621, to M.H.), The Stockholm County Council (No. FoUI-962332, to M.H.), Sigrid Jusélius Foundation (to M.P.), Canadian Institutes for Health Research (CIHR), Institute of Musculoskeletal Health and Arthritis catalyst Grant (No. 384224/151708, to J.F. and L.A.), the Swedish Radiation Safety Authority (Grant No. SSM2014-4016, to S.H.), and the Region Normandy, Caen, France, RIN ARCHADE CHOxTRaCC project (No. 20E06142-00018053, to S.H.).

## Author contributions

X.B., M.P., M.H., and N.X.L. conceived the study and designed the experiments. X.B., M.P., J.G., and Y.C. conducted the experiments, analyzed results, and prepared figures. Z.L., L.L., and Y.C. analyzed the

sequencing data. T.S. and S.H. helped with the in vitro irradiation experiments. E.E. and M.G. provided the PdgfraH2B-eGFP mouse line. X.B., C.D., J.F., and L.A. performed the murine irradiated skin and excisional wound experiments. C.B., M.C., and J.C. performed the FIBRL analysis of collagen fiber alignment in human skin. X.B. and M.M. constructed plasmids. M.H. and P.S. helped with human skin sample collection and collection of clinical data. X.B., M.P., and N.X.L. prepared the original draft, with feedback from all authors. L.L., J.G., and Y.C. made equal contributions.

## Funding

## Competing interests
The authors declare no competing interests.

## Additional information

[1]Dermatology and Venereology Division, Department of Medicine Solna, Center for Molecular Medicine, Karolinska Institutet, Stockholm, Sweden. [2]Department of Molecular Biosciences, The Wenner-Gren Institute, Stockholm University, Stockholm, Sweden. [3]Centre de recherche en organogénèse expérimentale de l'Université Laval / LOEX, Québec, QC, Canada. [4]Division of Regenerative Medicine, CHU de Québec-Université Laval Research Centre, Québec, QC, Canada. [5]Institute of Mechanical, Process and Energy Engineering, School of Engineering and Physical Sciences, Heriot-Watt University, Edinburgh EH14 4AS, UK. [6]Department of Cell and Molecular Biology, Karolinska Institutet, Solna, Sweden. [7]Centre for Inflammation Research, Institute for Regeneration and Repair, 4–5 Little France Drive, University of Edinburgh, Edinburgh EH16 4UU, UK. [8]Department of Physics, Université Laval/Centre de Recherche sur le Cancer, Université Laval/Centre de recherche du CHU de Québec, Québec, QC, Canada. [9]ABTE/ToxEMAC laboratory, University of Caen Normandy, Advanced Resource Center for HADrontherapy in Europe (ARCHADE), Caen, France. [10]Department of Surgery, Faculty of Medicine, Université Laval, Québec, QC, Canada. [11]Department of Plastic and Reconstructive Surgery, Karolinska University Hospital, Stockholm, Sweden. [12]Department of Molecular Medicine and Surgery, Karolinska Institutet, Stockholm, Sweden. [13]These authors contributed equally: Xiaowei Bian, Minna Piipponen. [14]These authors jointly supervised this work: Martin Halle, Ning Xu Landén. ✉e-mail: martin.halle@regionstockholm.se; ning.xu@ki.se

