## [Peer Review File · Nature Communications]

Epigenetic memory of radiotherapy in dermal fibroblasts
impairs wound repair capacity in cancer survivorsReviewers' Comments:

Reviewer #1:

Remarks to the Author:

Piipponen et al.

In the manuscript entitled “Epigenetic memory of radiotherapy in dermal fibroblasts impairs wound repair capacity in cancer survivors,” Piipponen and colleagues describe a mechanism where skin fibroblasts retain epigenetic memory from irradiation that leads to a long-term compromised wound healing capacity. Using skin biopsies of previously irradiated (RT+, 1-12yr post-irradiation) and non-irradiated (RT-) sites from breast cancer patients, the authors performed ex vivo wound healing assays and showed that previously irradiated skin has significant delays in closing wounds, as well as associated defects in wound-responsive fibroblast migration and transcription, suggesting a long-term memory of irradiation within this population. Consistently, the authors saw increased chromatin accessibility within a small subset of genomic loci in RT+ fibroblasts, which were further enriched for RUNX1-binding signatures. Digging into specific memory target genes in RT+ fibroblasts, the authors found that THBS1, which has previously been correlatively associated with delayed wound healing, is retained in a primed state post-irradiation that can be activated via the injury-induced cytokine TGF β . Indeed, inhibiting THBS1 using antibody blockade and the CRISPR-Cas9 system rescued delayed ex vivo healing in RT+ skin.

The finding that fibroblasts can retain epigenetic memories of stress builds upon the idea that such memory is a concept that is not specific to only immune cells. In addition, the TGF β -THBS1 pathway that the authors propose is responsible for delayed wound healing post-irradiation may be interesting as a therapeutic avenue to correct critical late-adverse events following irradiation therapies. Despite these potential advancements, the authors' characterizations of memory are weak and would need to be strengthened substantially to be informative. Additionally, the experiments done thus far are not sufficient to justify the mechanisms they propose. Overall, the work is not up to the level expected of Nature Communications readers.

The following are the major weaknesses:

1. The extent to which epigenetic memories in RT+ fibroblasts develop due to irradiation itself v. preceding mastectomies and/or cancer is currently unclear.

- o The authors should expand their murine irradiation model to more faithfully control for the effects of tumors and surgery – i.e., at least perform ATAC-seq on fibroblasts after tumor + mastectomy + irradiation, then compare how this regimen v. irradiation alone alters long-term chromatin accessibility away from a healthy (non-treated) baseline. If this is not feasible, the authors should still perform ATAC-seq on murine and/or in vitro human fibroblasts isolated post-irradiation alone to globally determine whether long-lasting epigenetic marks are observed over healthy controls, and the extent to which these may be consistent with those seen in humans.

- o The authors should also include how the epigenetic landscape looks like during radiation therapy in mouse fibroblasts and in vitro in human fibroblasts. That way the readers can see what peaks are specifically opened as a result of irradiation and remain open (i.e. “remember” irradiation) over time.

- o Some epigenetic memories may be obscured if irradiation (and/or cancer or mastectomies, as described above) exerts systemic effects throughout patient tissues. Healthy controls should be

included alongside paired-skin controls throughout experiments, at least for the murine model if not possible in humans.

2. Related to the above comment, the mouse irradiated skin and wound model needs to be much better characterized.

- o There is no figure to show whether the presented murine irradiation model truly delays long-term wound-healing capacity.

- o The rationale for assessing skin at D33 is unclear, especially since the authors argue earlier that THBS1 is acutely induced during wounding (Fig. 4). Likewise, the authors also do not demonstrate whether THBS1 follows similar expression patterns through wounding in mice as in humans.

- o The authors' observation that THBS1 expression remains high in post-irradiated then wounded skin (D33) is generally unconvincing.

- o Presented quantifications were done on the entire skin. How about just fibroblasts (e.g. by co-staining THBS1 with known fibroblast markers)? Further, if the authors propose that it is because of TGF β triggering RUNX1 dependent transcription of THBS1, they should show that there is increased and prolonged TGF β signal (e.g. via pSMAD2/3 staining or Western blots) and/or increased RUNX1 binding to THBS1 in fibroblasts of these mice.

- o It is difficult to distinguish the THBS1 signal in presented FISH images from autofluorescence. The authors should validate their results with qPCR of fibroblasts, both at D33 and earlier wound timepoints (e.g. a time course between D1-7).

3. The presented epigenetic data needs to be more thoroughly fleshed out:

- o Genome tracks should be shown to illustrate differentially accessible peaks (i.e. to complement Fig. 2B). Parameters to define differential accessibility should also be included in the main text for clarity.

- o Given the relatively low number of differential peaks between RT+ v. RT- fibroblasts, it will be of interest to perform global footprinting to predict genome-wide differences in TF binding which may not be reflected in chromatin accessibility alone.

- o Fig. 3B: some peaks are cut off; this figure should be remade using a consistent scale across all groups that allow for all peaks to be displayed in full.

- o If feasible, the authors should perform full sequencing experiments for RUNX1 (i.e. instead of just ChIP-qPCR), as these will allow for more convincing results with regards to RUNX1's role in mediating epigenetic memories in fibroblasts, e.g. via RUNX1 signal intensity across retained chromatin-accessible v. background domains and tracks to better visualize binding.

- o If feasible, the authors should also perform a whole RNA-seq (i.e. instead of just qPCR for THBS1) on RT+ v. RT- fibroblasts post-wounding (or at least post-TGF β) to determine the full set of genes that are poised post-irradiation, all of which may contribute to delayed wound closure.

4. The authors use TGF β throughout the manuscript to mimic wounding in vitro. However, the specific relevance of TGF β toward activating RT+ fibroblast memory/THBS1 as a memory target gene should be more thoroughly explored.

- o Whether the in vitro TGF β concentrations used by the authors recapitulates physiological levels during wounding (e.g. measured via ELISA) is unclear. In this vein, is THBS1 induction post-irradiation + TGF β dependent on TGF β dose?

5. More detail should be included on the authors' ex vivo wound healing model, especially since this system has not been previously described and can represent a valuable tool for the field to study human skin injury in controlled settings.
 - o Why is the CMFDA dye specifically useful to trace wound healing, e.g. over brightfield imaging?
 - o Wound closure kinetics are unclear – do wounds fully close in this model? D2 looks somewhat similar to D5 in both RT+ and RT- conditions (Fig. 1C). The authors should also quantify wound area over time and associated rate constants of wound closure, which will provide a more holistic metric of overall closure through the entire wound-healing process (see Naik S & Larsen SB, Nature 2017).
6. The authors use dermal explants to determine migration defects in RT+ fibroblasts (Fig. 1F). It will be worthwhile to repeat this experiment using whole-skin explants, which will allow the authors to determine whether keratinocytes themselves display enhanced migration (e.g. via K14 staining).
7. Figure 4 should be placed into the supplementary, as this only addresses THBS1 dynamics during normal wound healing and not post-irradiation.
8. Whether fibroblast-harbored epigenetic memory itself contributes to rescued wound healing upon THBS1 inhibition is not demonstrated (Fig. 6).
 - o Is wound healing rescued upon knocking out the open peaks that RUNX1 binds to which proposedly mediate enhanced THBS1 expression? Does knockout/knockdown of RUNX1 itself in fibroblasts recapitulate these effects?
 - o Lastly, for Figure 6D, the authors measure wound healing in RT+ fibroblasts with and without THBS1 knocked out but all in the presence of TGF β . How would RT+ fibroblasts with and without THBS1 knocked out behave without TGF β treatment?
9. The authors mention that clinical examination/histology revealed no noticeable differences in RT+ v. RT- skin, but do not include any figure.

Reviewer #2:

Remarks to the Author:

Very interesting and thorough study examining whether dermal fibroblasts retain epigenetic memory priming them towards dysregulated wound healing even years after the triggering insult. The authors focus on THBS1 as a candidate upregulated gene following secondary injury years after radiation therapy, and are convincingly show that neutralizing this protein promotes normal wound healing. This is an important piece of work for patients who may have complications related to prior radiation therapy, and underlines the potential of addressing epigenetic memory as an exciting therapeutic angle. Going a bit more in depth into how cellular crosstalk and spatial localization may promote this dysregulation would enhance the impact of the paper, described more in attached document.

General comments:

Very interesting and thorough study examining whether dermal fibroblasts retain epigenetic memory priming them towards dysregulated wound healing even years after the triggering insult. The authors focus on THBS1 as a candidate upregulated gene following secondary injury years after radiation therapy, and are able to convincingly show that neutralizing this protein may promote normal wound healing. This is an important piece of work for patients who may have complications related to prior radiation therapy, and underlines the potential of addressing epigenetic memory as an exciting therapeutic angle. Going a bit more in depth into how cellular crosstalk and spatial localization may promote this dysregulation would enhance the impact of the paper, described more below.

Major comments:

Curious if any major differences were observed between length of time between radiation and skin biopsy, mostly in regards to functional capacity of dermal fibroblasts. Do those cells that have had more time to 'rest' demonstrate as much of a functional defect as those that have seen radiation more recently? It would be very interesting to see if there would be any correlation there.

The authors state that RT+ and RT- skin does not display any differences at homeostasis, could this data be included in supplemental? Also it could be interesting to stain with something like picosirius red to determine if ECM composition/organization remains similar in both groups.

Was there any difference in proliferation between RT- and RT+ cells?

It's an interesting result that TGFb treatment had a muted effect in RT+ cells as far as activation/ECM production. How do you think this relates back to the functional changes seen in explant migration/wound healing? Further, THBS1 has been shown to be linked to latent TGFb activation, this might be worth mentioning in the discussion and could also explain the antifibrotic effects of THBS1 inhibition (<https://www.ncbi.nlm.nih.gov/pmc/articles/PMC6015530/>)

Figure 1 demonstrates that RT+ fibroblasts have a decreased migration capacity, yet are shown in Figure 2 to highly express RUNX which is involved in cell migration. How do you reconcile these data?

In the experiments treating RT- fibroblasts to ionizing radiation, did you also examine the epigenome? Would be interesting to see if similar epigenetic alterations would be seen in this shorter timespan compared to cells from irradiated patient tissue.

Regarding the ST data, a bit of a missed opportunity to take better advantage of the spatial component. Could you look at whether high concentrations of THBS1 expression correlate more with fibroblasts/PCvSMC etc at different timepoints, or with other microanatomical niches? May be a bit difficult to do with the low resolution of the 10X platform (which should be mentioned as a limitation), but there are methodologies to integrate sc-seq and ST, which would really enhance this data. Fig. 4E then could act as ST validation in supplementary.

THBS1 crosstalk is an important point, but missing a more thorough discussion on the impact of its increased signaling in RT+ skin and how this may impact clinical outcomes and inform immunotherapies outside of THBS1 neutralization.

Minor comments:

Fig. 1C is a bit confusing – the edge of fluorescent staining suggests that the wounds are actually closing faster than the dotted line would suggest.

Line 137 – are the 97 less accessible peaks necessarily RT- up, or are they just the least accessible in the RT+ fibroblasts? I'm assuming they are the highest accessible in the RT- group based on Fig 2D, but needs clarification.

Line 238 – a bit unclear whether this group performed these experiments based on the wording

Line 282 - Please spell out late-onset adverse effects (LAE)

Discussion line 348: might be more accurate to say that we can 'address' the maladaptive radiation memory vs 'erasing' it

Reviewer #3:

Remarks to the Author:

In their manuscript, “Epigenetic memory of radiotherapy in dermal fibroblasts impairs wound repair capacity in cancer survivors,” Piipponen, Bian et al. provide a valuable molecular and functional study to help understand the mechanisms of late-onset radiotherapy toxicities with focus on epigenetic regulators. By comparing paired skin biopsies from irradiated and non-irradiated sites in breast cancer survivors, the authors found impaired wound healing and impaired fibroblast function after radiotherapy and demonstrated solid evidence of a persistently altered chromatin landscape in irradiated fibroblasts. The authors identified THBS1 as a potential mediator of this phenotype fibroblasts and validated its importance in mouse and human wound models. Remarkably, the authors demonstrated the ability of anti-THBS1 antibodies to promote wound closure in an ex vivo irradiated skin model. Overall, this work will be of interest to the fields of cancer and radiation biology, providing important insights into the mechanisms of late radiation toxicity and demonstrating the clinical promise of targeting these processes to reverse the long-term side effects of radiotherapy. However, I have several concerns that should be addressed and suggestions to strengthen the findings prior to publication.

1. The time from end of radiotherapy to tissue collection varies from 1-12 years, which is a relatively large range. According to Table S1, different patients were selected for various experiments in the study. Did the authors consider any criteria for assigning patients to different experiments? For example, only five patients (line 132) with a narrower range of time since the end of radiation were assigned to the ATAC-seq experiment. Could authors include additional cases (such as cases 37 and 13, collected 120 and 129 months after radiation, respectively) to compare how epigenetic changes persist at later timepoints after radiotherapy?
2. The authors identified THBS1 by analyzing genes with altered chromatin accessibility in irradiated fibroblasts that were also differentially expressed during the normal wound-healing process (Day1 or Day7 wounds vs. donor-matched skin). This seems counter-intuitive because they were interested in genes that lead to impaired wound healing. Can the authors explain their rationale for this approach? What genes are differentially expressed during wound healing in irradiated versus non-irradiated skin?
3. For most of their histological analyses, the authors utilized FISH for THBS1 rather than immunofluorescence for the THBS1 protein. Although they demonstrated significant differences by quantification, it is difficult to appreciate significant differences in the presented images. It may be easier to appreciate the differences if the THBS1 and DAPI images are displayed separately instead of overlaid. In Figure S4, the authors show that they can perform immunofluorescence for the THBS1 protein, but the difference between groups appears less substantial. It would be helpful to show that protein levels of THBS1 are significantly different in their mouse model (Fig. 5B) and/or in radiation ulcers (Fig.5F).
4. The authors provide impressive data that an anti-THBS1 antibody promotes wound healing after radiation exposure in ex vivo wounds. However, it is unclear how the control group was treated in the anti-THBS1 antibody experiment in Fig. 6E. Was the gel that was mixed with the THBS1 antibody added to the controls? It is possible that the gel promotes wound healing in the absence of the antibody. It would substantially strengthen the manuscript if the authors could demonstrate that the anti-THBS1 antibody promotes wound healing in vivo in their murine wound model.

5. The authors provide compelling evidence for THBS1's role in delayed wound healing after radiotherapy. However, the downstream mechanism of how increased THBS1 expression blocks fibroblast migration, contraction, and wound healing is unclear. Can the authors comment on why THBS1 expression could affect these processes?

Minor Notes

1. Line 94: The authors state that patients underwent external beam radiotherapy to a total dose of 50 Gy. However, the methods and Table S1 show that there was a range of doses from 40 to 60 Gy. This should be consistent.
2. How were the wounds created for the human wound healing model? Can the authors clarify the difference between full depth and partial-thickness wounds?
3. For the GO enrichment networks in Figure 2C and S1, it would be helpful to explain what the size and colors of the nodes represents in the legend.
4. For the scRNA-seq analysis in Figure 4A, the authors identified 26 cell types based on their clustering strategy, including 3 basal, 3 spinous, and 2 granular keratinocyte populations and 4 fibroblast populations. Are these sub-populations biologically meaningful or just a byproduct of the resolution chosen for their clustering approach? Have these sub-populations been described previously in the literature and is there a difference in THBS1 expression between them?
5. Why isn't there a D0 timepoint for the 45 Gy treatment in Figure 5C?

Point by point response:

Reviewer #1: expertise in wound healing biology and epigenetic memory, single cell omics and ATAC-seq:

Piipponen et al. In the manuscript entitled “Epigenetic memory of radiotherapy in dermal fibroblasts impairs wound repair capacity in cancer survivors,” Piipponen and colleagues describe a mechanism where skin fibroblasts retain epigenetic memory from irradiation that leads to a longterm compromised wound healing capacity. Using skin biopsies of previously irradiated (RT+, 1-12yr post-irradiation) and non-irradiated (RT-) sites from breast cancer patients, the authors performed *ex vivo* wound healing assays and showed that previously irradiated skin has significant delays in closing wounds, as well as associated defects in wound-responsive fibroblast migration and transcription, suggesting a long-term memory of irradiation within this population. Consistently, the authors saw increased chromatin accessibility within a small subset of genomic loci in RT+ fibroblasts, which were further enriched for RUNX1-binding signatures. Digging into specific memory target genes in RT+ fibroblasts, the authors found that THBS1, which has previously been correlatively associated with delayed wound healing, is retained in a primed state post-irradiation that can be activated via the injury-induced cytokine TGF β . Indeed, inhibiting THBS1 using antibody blockade and the CRISPR-Cas9 system rescued delayed *ex vivo* healing in RT+ skin.

The finding that fibroblasts can retain epigenetic memories of stress builds upon the idea that such memory is a concept that is not specific to only immune cells. In addition, the TGF β -THBS1 pathway that the authors propose is responsible for delayed wound healing post-irradiation may be interesting as a therapeutic avenue to correct critical late-adverse events following irradiation therapies. Despite these potential advancements, the authors' characterizations of memory are weak and would need to be strengthened substantially to be informative. Additionally, the experiments done thus far are not sufficient to justify the mechanisms they propose. Overall, the work is not up to the level expected of NatureCommunications readers.

R1: We are immensely thankful to Reviewer #1 for the exceptional recommendations provided. In light of these insightful suggestions, we have undertaken substantial new experiments and analyses. These include further ATAC-seq, RNA-seq, and ChIP-seq on irradiated fibroblasts, as well as single-cell ATAC-seq and RNA-seq comparing RT⁺ versus RT⁻ patient skin. Additionally, we have refined our FISH and IF analyses on both murine and patient tissues post-radiation, conducted more comprehensive cellular functional studies, and expanded our *ex vivo* wound models *etc.*. These enhancements have significantly solidified the mechanism we propose and have furthered our comprehension of epigenetic memory.

The following are the major weaknesses:

1. The extent to which epigenetic memories in RT⁺ fibroblasts develop due to irradiation itself v. preceding mastectomies and/or cancer is currently unclear.

- o The authors should expand their murine irradiation model to more faithfully control for the effects of tumors and surgery – i.e., at least perform ATAC-seq on fibroblasts after tumor + mastectomy + irradiation, then compare how this regimen v. irradiation alone alters long-term chromatin accessibility away from a healthy (non-treated) baseline. If this is not feasible, the authors should

still perform ATAC-seq on murine and/or *in vitro* human fibroblasts isolated post-irradiation alone to globally determine whether long-lasting epigenetic marks are observed over healthy controls, and the extent to which these may be consistent with those seen in humans.

- The authors should also include how the epigenetic landscape looks like during radiation therapy in mouse fibroblasts and *in vitro* in human fibroblasts. That way the readers can see what peaks are specifically opened as a result of irradiation and remain open (i.e. “remember” irradiation) over time.

R1.1a, b: We concur with the Reviewer on the importance of distinguishing the specific effects of irradiation from the potential influences of cancer and mastectomy. Following the Reviewer's advice, we performed ATAC-seq on healthy human fibroblasts *in vitro* after irradiation at one (D1) and seven days (D7) post-treatment (**new Fig. 2C**). These time points were chosen based on the expression dynamics of CDKN1A and PCNA in the treated fibroblasts, to capture the peak response to irradiation and the late stage of DNA damage repair, respectively. (**Fig. 3H, I, and new Fig. S4E, F**)^{1, 2}. Our ATAC-seq analysis revealed increased chromatin accessibility in 2000 genes at D1 and 2765 genes at D7 compared to non-irradiated cells ($\log_2FC > 0$, $p\text{-value} < 0.05$) (**new Fig. 2D**). Notably, 23 of the 59 genes that showed increased accessibility in RT⁺ fibroblasts compared to RT⁻ fibroblasts from cancer survivors (RT⁺ up) were more accessible post-irradiation in either D1 or D7 samples (**new Fig. 2D**). Additionally, the THBS1 gene exhibited increased accessibility only in D7-irradiated samples, not in D1, suggesting the epigenetic changes in the THBS1 gene occur during the DNA damage repair process (**new Fig. 2D**). Consistent with this, THBS1 expression was more markedly induced in irradiated cells than in non-irradiated cells when treated with TGF-13 six days post-irradiation. This difference was not observed in fibroblasts treated with TGF-13 three hours and one-day post-irradiation (**new Fig. 3K-M, Fig. S4H**). Despite complexities such as the differences between *in vivo* and *in vitro* fibroblast states and between clinical and experimental radiation exposures, these new ATAC-seq data suggest that some of the memory domains identified in cancer survivors are directly associated with radiotherapy.

Clinically, obtaining skin solely irradiated from healthy individuals is challenging and would not be ethically sound. However, we consider the influence of previous surgery negligible since the collected RT⁺ skin samples were selected from areas distant from the previous cancer and mastectomy scars, but still within the irradiated zone (**new Fig. S1A**). Regarding the use of a murine model to replicate these clinical conditions, it is both technically and ethically difficult to create a model involving breast cancer followed by sequential treatments of mastectomy, irradiation, and re-wounding. Furthermore, the extent to which such a murine model could accurately represent breast cancer patients' complex conditions is debatable.

- Some epigenetic memories may be obscured if irradiation (and/or cancer or mastectomies, as described above) exerts systemic effects throughout patient tissues. Healthy controls should be included alongside paired-skin controls throughout experiments, at least for the murine model if not possible in humans.

R1.1c: The paired collection of RT⁺ and RT⁻ samples from the same donors is a key strength of our study design, significantly reducing donor variability—a major hurdle in studying clinical samples—and allowing for the correlation of molecular and clinical phenotypes individually (**new Fig. S1C, J, K, new Fig. S2B, new Fig. S4A-C**). While we acknowledge the Reviewer's point about

potential limitations of this design in detecting systemic effects of irradiation, surgeries, or cancer, clinical evidence shows that late-onset adverse effects (LAEs) of radiotherapy on the skin are primarily local, occurring at the previous irradiation site. This suggests that the systemic impacts of previous treatments or cancer are less likely to be the root cause of these LAEs.

In response to the Reviewer's advice, we have incorporated additional healthy controls into our study. For instance, we compared the *ex vivo* wound healing capabilities of unirradiated skin (RT⁻) from cancer survivors with the skin from healthy donors. We observed no significant differences between these groups, whereas the disparity between RT⁺ and RT⁻ skin was evident (**new Fig. 1D, E, Fig. S1D**). Additionally, our expanded experiments include ATAC-seq analysis of *in vitro* fibroblasts post-irradiation, using samples from healthy donors (**new Fig. 2C**). This analysis corroborated most of the radiotherapy-related epigenetic changes observed in cancer survivors. We also noted similar gene expression dynamics post-irradiation in fibroblasts from the RT⁻ area of cancer survivors (RT⁻ FBs, **Fig. 3G-J**) and from healthy donors (HDFa, **new Fig. S4E-G**). Our collective clinical and experimental findings provide no evidence of systemic effects on the skin from prior treatments or cancer. Therefore, we did not pursue a further analysis of such systemic impacts at the epigenomic level, given their minimal relevance to the pathology of local LAEs.

2. Related to the above comment, the mouse irradiated skin and wound model needs to be much better characterized.

o There is no figure to show whether the presented murine irradiation model truly delays long-term wound-healing capacity.

R1.2a: The murine irradiation-wound model, which demonstrates delayed healing in previously irradiated skin versus non-irradiated skin, has been recently published³ (**Figure 1 for reviewers**). Not only the healing speed, the granulation tissue of the wounds created in irradiated skin was less organized, more heterogeneous and featured a lower collagen content³. In the current study, we conducted FISH and novel IF analyses on these murine skin and wound tissues, complementing the wound healing kinetics and histological analyses previously performed³. In the revised manuscript, we describe the model and the published results further (see page 15, the 2nd paragraph), alongside the results of additional FISH and protein co-labeling performed during the revision.

Figure 1 for reviewers. adapted from the Figure 7 in³ *C. Diaz et al., Ionizing Radiation Mediates Dose Dependent Effects Affecting the Healing Kinetics of Wounds Created on Acute and Late Irradiated Skin. Surgeries 2, 35-57, 2021. Healing kinetics of excisional wounds created on irradiated skin. (A) Macroscopic images showing the reduction in wound size over the course of 33 days after surgery.*

Measurements of global wound closure were made on such images for each wound, using ImageJ software. (B) Kinetics and (C) Global final wound closure values as a function of the dose received, reflecting the severity of the radiodermatitis that developed during four weeks prior to excisional surgery and splinting of the wounds. Statistics: mean \pm SD. $n = 3-5$ mice/group, One-way ANOVA with Dunnett's multiple comparisons test ** $p < 0.01$. * Symbol indicates significance between control and other groups.

○The rationale for assessing skin at D33 is unclear, especially since the authors argue earlier that THBS1 is acutely induced during wounding (Fig. 4). Likewise, the authors also do not demonstrate whether THBS1 follows similar expression patterns through wounding in mice as in humans.

R1.2b: Previous research has thoroughly investigated Thbs1 expression patterns post-wounding in mice through *in situ* hybridization and immunostaining techniques 4, 5, 6, 7. Complementing this, we have depicted the trajectory of Thbs1 expression in fibroblasts during wound healing by employing recent single-cell RNA-seq data from murine model 8, 9, as illustrated in the **new Fig. S6A, B**. Further, our investigations into a murine wound healing model using both qRT-PCR (**new Fig. S6C**) and single-cell RNA-seq (**Fig. S6D**) confirmed that THBS1 expression dynamics in mice parallel those in human tissues (**Fig. 4A-G**). Specifically, THBS1 is minimally expressed in unwounded skin, surges rapidly post-injury, and subsequently reverts to baseline during the remodeling phase. In our study, we selected day 33 for sampling to capture the pronounced contrast in wound healing outcomes between non-irradiated (IR-) and irradiated (IR+) conditions. At this juncture, we observed a resolution of Thbs1 expression in IR- samples, whereas IR+ samples maintained heightened Thbs1 levels (**new Fig. 5B-D, Fig. S7A, B**). Therefore, this time point provides valuable insight into the late effects of radiation, which results in the sustained expression of THBS1 throughout the wound healing process.

○The authors' observation that THBS1 expression remains high in post-irradiated then wounded skin (D33) is generally unconvincing.

○Presented quantifications were done on the entire skin. How about just fibroblasts (e.g. by co-staining THBS1 with known fibroblast markers)? Further, if the authors propose that it is because of TGF β 3 triggering RUNX1 dependent transcription of THBS1, they should show that there is increased and prolonged TGF β 3 signal (e.g. via pSMAD2/3 staining or Western blots) and/or increased RUNX1 binding to THBS1 in fibroblasts of these mice.

○It is difficult to distinguish the THBS1 signal in presented FISH images from autofluorescence. The authors should validate their results with qPCR of fibroblasts, both at D33 and earlier wound timepoints (e.g. a time course between D1-7).

R1.2c-e: Following the reviewer's suggestion, we co-stained Thbs1 (using FISH and immunofluorescence, IF) with the fibroblast marker Pdgfra (using IF), revealing increased Thbs1 mRNA and protein expression in IR+ versus IR- murine wound fibroblasts (**new Fig. 5B-D, new Fig. S7A, B**)¹⁰. We also performed IF co-staining for pSMAD2, a TGF β 3 signaling effector, and Pdgfra, observing elevated and prolonged TGF β 3 signaling in IR+ fibroblasts compared to IR- ones (**new Fig. S7C, D**). This difference was notable even in IR+ mouse skin before wounding. Time constraints during the revision period limited us from conducting additional rounds of the murine radiation-wound model to gather early-timepoint samples. Nevertheless, we conducted a thorough

analysis of our existing murine tissue samples. Enhancements in our methodology, including Pdgfra co-staining, improved Thbs1 FISH protocol, and new Thbs1 IF data, robustly demonstrate the abnormal expression pattern of Thbs1 in post-irradiated then wounded murine skin.

3. The presented epigenetic data needs to be more thoroughly fleshed out:

- Genome tracks should be shown to illustrate differentially accessible peaks (i.e. to complement Fig. 2B). Parameters to define differential accessibility should also be included in the main text for clarity.

R1.3a: Following the Reviewer's advice, we have enriched our original Figure 2B by incorporating representative Integrative Genomics Viewer (IGV) snapshots that showcase genomic locations with ATAC-seq signal tracks, highlighting a few more differentially accessible (DA) peaks (**new Fig. 3B, Fig. S2C, D**). Additionally, we have included heatmaps that depict the ATAC-seq signal intensity (**new Fig. S2A**). The parameters to define DA peaks have now been incorporated into the main text (page 8, the 2nd paragraph).

- Given the relatively low number of differential peaks between RT+ v. RT- fibroblasts, it will be of interest to perform global footprinting to predict genome-wide differences in TF binding which may not be reflected in chromatin accessibility alone.

R1.3b: We agree with the Reviewer. In the previous version of the manuscript, we have already performed TOBIAS transcription factor (TF) footprinting analysis of the ATAC-seq data (**Fig. 2G, H**). We observed a strong enrichment in the differential binding score for RUNX footprints in the RT+ fibroblasts compared to the RT- fibroblasts, suggesting the role of RUNX1 in the epigenetic changes associated with late irradiation.

- Fig. 3B: some peaks are cut off; this figure should be remade using a consistent scale across all groups that allow for all peaks to be displayed in full.

R1.3c: We have revised **Fig. 3B** by adjusting the scale, enabling all peaks to be displayed in full.

- If feasible, the authors should perform full sequencing experiments for RUNX1 (i.e. instead of just ChIP-qPCR), as these will allow for more convincing results with regards to RUNX1's role in mediating epigenetic memories in fibroblasts, e.g. via RUNX1 signal intensity across retained chromatin-accessible v. background domains and tracks to better visualize binding.

R1.3d: In line with the Reviewer's suggestion, we attempted RUNX1 ChIP-seq on RT- and RT+ fibroblasts, both with and without TGF-13 treatment. Unfortunately, the samples without TGF-13 treatment did not yield sufficient DNA for sequencing. We proceeded with the TGF-13-treated samples, which also had a low DNA yield, necessitating an additional PCR amplification step prior to sequencing. From this process, we identified 685 genes with RUNX1 binding, and the specificity of our ChIP assay was validated by the significant enrichment of RUNX binding motifs in the pulled down DNA (**Figure 2 for reviewers**). Among these, RUNX1 binding was noted at six genes containing poised chromatin domains post-radiotherapy (RT+ up domains).

Despite these findings, we acknowledge that the quality of the resulting ChIP-seq data might not meet the publication standards due to the presence of excessively broad peaks (> 1 kb in width) upon visual inspection of normalized read pileups in a genome browser. This could be due to PCR over-amplification of the limited ChIP DNA, potentially caused by low RUNX1 levels or its weak genome binding. Hence, ChIP-qPCR, due to its sensitivity, may be a better choice for assessing RUNX1's genomic interaction than ChIP-seq. Our focused ChIP-qPCR data, while not offering a comprehensive view of RUNX1's genomic binding, provides precise evidence of RUNX1's interaction with the THBS1 locus, reinforcing our findings.

Figure 2. RUNX1 ChIP-seq analysis in human fibroblasts. (A) Heatmap showing global RUNX1 occupancy. (B) HOMER motif analysis of the pulled-down DNA. (C) Comparison of the RUNX1 binding genes identified by the ChIP-seq with the genes containing poised chromatin domains post-radiotherapy (RT+ up domains).

o If feasible, the authors should also perform a whole RNA-seq (i.e. instead of just qPCR for THBS1) on RT+ v. RT- fibroblasts post-wounding (or at least post-TGF13) to determine the full set of genes that are poised post-irradiation, all of which may contribute to delayed wound closure.

R1.3e: During revision, we performed RNA-seq on fibroblasts from five patients. Results showed minor gene expression differences between RT⁺ and RT⁻ cells (**new Fig. S3A**). Notably, 93% of genes with altered chromatin accessibility in RT⁺ cells were either not expressed or not differentially expressed compared to RT⁻ cells, suggesting that the epigenetic memory of radiotherapy was not unfolded at the homeostatic cell state (**new Fig. S3B**). This aligns with the similar clinical and histological profiles of RT⁺ and RT⁻ skin (**new Fig. S1A-C**).

As it is not feasible to obtain RT⁺ vs. RT⁻ fibroblasts post-wounding from patients, we used TGF- β , an important signal for fibroblasts in wound healing (see **R1.4**), to treat patient-derived fibroblasts. RNA-seq analysis showed that ECM-related genes were more induced by TGF-13 in RT⁻ cells, while ribosome biogenesis-related genes were more upregulated in RT⁺ cells (**new Fig. S3C**). TGF-13 treatment activated 14 of the 59 genes with poised chromatin post-radiotherapy, with six overlapping with genes differentially expressed in human wounds (**new Fig. 2K, L**). Our data also

demonstrated lower ECM production in RT⁺ fibroblasts compared to RT⁻ cells after TGF13 stimulation (**new Fig. 1J, Fig. S1H, I**). These new RNA-seq data reveal how RT⁺ fibroblasts differ from RT⁻ cells in their response to TGF-13 signals, shedding light on the molecular alterations that impair RT⁺ fibroblasts' function in wound repair. However, genes poised post-radiotherapy but unresponsive to TGF-13 could also be crucial in wound healing and other late-onset adverse effects (LAEs) like fibrosis and cancer. A deeper understanding of radiotherapy's epigenetic memory and its impact on various LAEs necessitates further research.

To further improve the characterization of epigenetic memory, we also used the 10x Genomics Chromium Single Cell Multiome ATAC+Gene Expression platform on paired RT⁺ and RT⁻ skin from cancer survivors (**new Fig. S5**). This method provided concurrent gene expression and epigenomic data from single cells in patients' skin *in vivo*. Among the 14 skin cell types, we found that the THBS1 promoter region is more accessible in the RT⁺ fibroblasts compared to the RT⁻ cells, while its mRNA expression remains similar between these two groups. Also, despite similar RUNX expression levels in RT⁺ and RT⁻ fibroblasts, RUNX family motif activity was higher in RT⁺ cells, supporting RUNX's role in the long-term impact of radiotherapy.

4. The authors use TGF13 throughout the manuscript to mimic wounding *in vitro*. However, the specific relevance of TGF13 toward activating RT⁺ fibroblast memory/THBS1 as a memory target gene should be more thoroughly explored.

- Whether the *in vitro* TGF13 concentrations used by the authors recapitulates physiological levels during wounding (e.g. measured via ELISA) is unclear. In this vein, is THBS1 induction post-irradiation + TGF13 dependent on TGF13 dose?

R1.4: TGF-13, a key regulator of fibroblast functions in wound healing ^{11, 12, 13}, is known to enhance expression ^{14, 15} and activity ¹⁶ of RUNX1 and induce THBS1 expression in fibroblasts ¹⁷. Our new RNA-seq data from TGF-13-treated fibroblasts reveal that 27% of wound healing signature genes are upregulated by TGF-13 in human fibroblasts (**new Fig. 2K**). Additionally, 14 of the 59 genes with poised chromatin domains post-radiotherapy are activated by TGF-13 treatment (**new Fig. 2K**), validating the use of TGF-13 to simulate wound healing and trigger RT⁺ fibroblasts' epigenetic memory.

Considering previous research typically uses 0.1 to 10 ng/ml TGF-13 for *in vitro* fibroblast treatment and the fact that TGF-13 levels in wound fluid of breast cancer patients can reach 10-20 ng/ml 24 hours post-surgery ¹⁸, we tested various TGF-13 concentrations (0.01-5 ng/ml) on human RT⁺ and RT⁻ fibroblasts. We found that a 5 ng/ml concentration, similar to physiological levels in human wounds, was the minimum to induce THBS1 expression in RT⁺ fibroblasts without affecting RT⁻ cells (**new Fig. S4D**), and this concentration was used in our study.

5. More detail should be included on the authors' *ex vivo* wound healing model, especially since this system has not been previously described and can represent a valuable tool for the field to study human skin injury in controlled settings.

- Why is the CMFDA dye specifically useful to trace wound healing, e.g. over brightfield imaging?

o Wound closure kinetics are unclear – do wounds fully close in this model? D2 looks somewhat similar to D5 in both RT+ and RT- conditions (Fig. 1C). The authors should also quantify wound area over time and associated rate constants of wound closure, which will provide a more holistic metric of overall closure through the entire wound-healing process (see Naik S & Larsen SB, Nature 2017).

R1.5: We agree with Reviewer that the *ex vivo* wound healing model is an invaluable tool for translationally relevant preclinical wound healing research. Established protocols have been published^{19, 20, 21}, and we have also provided more technical details in the revised method part.

The identification of the newly formed epithelial tongue via brightfield imaging is problematic, which hinders accurate wound size determination. The use of the fluorescent vital dye CMFDA to label epithelial cells within wounds circumvents this issue, allowing for longitudinal monitoring of wound closure and facilitating evaluation of re-epithelialization and tissue contraction²⁰.

Ex vivo wounds can achieve full re-epithelialization. Fig. 1C shows initial wound edges with dotted lines to gauge contraction. Within these markers, the epithelial tongue is visible, with day 5 (D5) wounds showing greater re-epithelialization than day 2 (D2) wounds. Following the Reviewer's advice, healing rate constants for the *ex vivo* wound model have been calculated and are presented in the **new Fig. 1E**²².

6. The authors use dermal explants to determine migration defects in RT+ fibroblasts (Fig. 1F). It will be worthwhile to repeat this experiment using whole-skin explants, which will allow the authors to determine whether keratinocytes themselves display enhanced migration (e.g. via K14 staining).

R1.6: In our skin explant assay, epidermal tissue was utilized for the isolation of keratinocytes. Employing scratch wound assays, we have demonstrated that keratinocytes derived from RT+ skin exhibit significantly slower migration than those from RT- skin (**new Fig. S1G**). Our lab is currently engaged in a project investigating the mechanisms behind this disparity in keratinocytes.

7. Figure 4 should be placed into the supplementary, as this only addresses THBS1 dynamics during normal wound healing and not post-irradiation.

R1.7: The results in Figure 4 emphasize the crucial role of fibroblast producing THBS1 in human skin wound healing. This underscores the significance of our finding that THBS1 expression is epigenetically primed in RT+ fibroblasts. Additionally, the insights from Figure 4 have received commendations from other reviewers, who suggested enhancing the spatial transcriptomics analysis and delving into the effects of THBS1-mediated cell-cell communication in the wound healing. In light of this supportive feedback, we are inclined to retain Figure 4 as a main figure in this manuscript.

8. Whether fibroblast-harbored epigenetic memory itself contributes to rescued wound healing upon THBS1 inhibition is not demonstrated (Fig. 6).

○ Is wound healing rescued upon knocking out the open peaks that RUNX1 binds to which proposedly mediate enhanced THBS1 expression? Does knockout/knockdown of RUNX1 itself in fibroblasts recapitulate these effects?

R1.8a: We acknowledge the Reviewer's point on the need to show that epigenetic memory in fibroblasts contributes to improved wound healing with THBS1 inhibition. However, deleting RUNX1 binding peaks at THBS1 loci may be inappropriate due to potential unintended effects, such as loss of THBS1 expression or altered chromatin dynamics. These could complicate the interpretation of results. Therefore, we followed the Reviewer's second advice, to knockdown RUNX1 expression in fibroblasts with siRNA. Silencing RUNX1 expression resulted in a significant increase in migration and extracellular matrix production (COL1A3 and ELN) of RT⁺ fibroblasts, but not in RT⁻ fibroblasts (**new Fig. 6M-P**). Additionally, topical application of this siRNA, along with a transfection reagent on human *ex vivo* wounds, led to a marked reduction in RUNX1 expression compared to wounds treated with control siRNAs (**new Fig. S8J**). The silencing of RUNX1 decreased THBS1 while it increased ECM expression in RT⁺ *ex vivo* wounds (**new Fig. S8K, L**). Notably, RUNX1 silencing was observed to improve the healing of *ex vivo* wounds in RT⁺ skin, with no significant impact on RT⁻ skin wounds (**new Fig. S8I, M**). These new data demonstrate that RUNX1 knockdown can recapitulate the effects of THBS1 inhibition, reinforcing the idea that fibroblast-harbored epigenetic memory itself contributes to rescued wound healing upon THBS1 inhibition.

○ Lastly, for Figure 6D, the authors measure wound healing in RT⁺ fibroblasts with and without THBS1 knocked out but all in the presence of TGF β 3. How would RT⁺ fibroblasts with and without THBS1 knocked out behave without TGF β 3 treatment?

R1.8b: Additional experiments were conducted similarly to those in previous Figure 6D (now Fig. 6J), but without TGF- β 3 treatment. Interestingly, we found that reducing THBS1 expression increased migration in RT⁺ fibroblasts only when treated with TGF- β 3, indicating TGF- β 3's role in activating THBS1 and exacerbating functional abnormalities in RT⁺ cells (**Fig. 6J, new Fig. S8B**). Therefore, the enhanced migratory effect of THBS1 inhibition is more pronounced with TGF β 3 treatment.

9. The authors mention that clinical examination/histology revealed no noticeable differences in RT⁺ v. RT⁻ skin, but do not include any figure.

R1.9: In the revised manuscript, we included representative clinical and histological images of both RT⁺ and RT⁻ skin from patients (**new Fig. S1A-C**). Macroscopically, the RT⁺ and RT⁻ skin appeared similar, with no noticeable differences. We also performed histological analyses using Masson's trichrome staining on both skin samples and assessed collagen fiber orientation using the FIBRAL application²³. This system classifies a low alignment index as indicative of randomly oriented fibers, typical in non-scarred skin, and a high index as aligned fibers, characteristic of scarring²³. Our results reveal no significant difference in collagen fiber alignment between paired RT⁺ and RT⁻ skin samples. Additionally, the disparities between these two sites tend to lessen over time following radiotherapy (**new Fig. S1B, C**).

Reviewer #2, expertise in wound healing, fibroblasts and ST:

Very interesting and thorough study examining whether dermal fibroblasts retain epigenetic memory priming them towards dysregulated wound healing even years after the triggering insult. The authors focus on THBS1 as a candidate upregulated gene following secondary injury years after radiation therapy, and are able to convincingly show that neutralizing this protein may promote normal wound healing. **This is an important piece of work for patients who may have complications related to prior radiation therapy, and underlines the potential of addressing epigenetic memory as an exciting therapeutic angle.** Going a bit more in depth into how cellular crosstalk and spatial localization may promote this dysregulation would enhance the impact of the paper, described more below.

Major comments:

1. Curious if any major differences were observed between length of time between radiation and skin biopsy, mostly in regards to functional capacity of dermal fibroblasts. Do those cells that have had more time to 'rest' demonstrate as much of a functional defect as those that have seen radiation more recently? It would be very interesting to see if there would be any correlation there.

R2.1: We appreciate the Reviewer's insight, which prompted a more in-depth investigation during our revision. We explored the correlation between fibroblast migration, *ex vivo* wound healing in RT⁺ and RT⁻ skin, and the length of time (LoT) post-radiotherapy, but found no significant association (**new Fig. S1J, K**). Additionally, we used the FIBRAL application²³ for a quantitative analysis of collagen fiber orientation in paired RT⁺ and RT⁻ skin (see **R2.2**). Interestingly, the variation in collagen alignment between RT⁺ and RT⁻ samples significantly decreased with LoT (**new Fig. S1C**). To investigate if the chromatin accessibility differences may change with LoT, we performed additional ATAC-seq comparing RT⁺ and RT⁻ cells from two more donors (cases 17 and 26 in Table S1) with 108 and 84 months post-radiotherapy, respectively, during the revision. However, we did not find any apparent correlation between the epigenetic changes with LoT (**new Fig. S2A, B**). Moreover, we observed that the disparity in THBS1 mRNA expression between RT⁺ and RT⁻ decreased over time, independent of donor age or radiation dose during RT (**new Fig. S4A-C**). Collectively, these data imply that some long-term impacts of radiation exposure may diminish over time in cancer survivors.

2. The authors state that RT⁺ and RT⁻ skin does not display any differences at homeostasis, could this data be included in supplemental? Also it could be interesting to stain with something like picosirius red to determine if ECM composition/organization remains similar in both groups.

R2.2: In the revised manuscript, we included representative clinical and histological images of both RT⁺ and RT⁻ skin from patients (**new Fig. S1A, B**). Macroscopically, the RT⁺ and RT⁻ skin appeared similar, with no noticeable differences. We also performed histological analyses using Masson's trichrome staining on both skin samples and assessed collagen fiber orientation using the FIBRAL application²³. This system classifies a low alignment index as indicative of randomly oriented fibers, typical in non-scarred skin, and a high index as aligned fibers, characteristic of scarring²³. Our results reveal no significant difference in collagen fiber alignment between paired RT⁺ and RT⁻ skin

samples. Additionally, the disparities between these two sites tend to lessen over time following radiotherapy (**new Fig. S1B, C**).

3. Was there any difference in proliferation between RT- and RT+ cells?

R2.3: We utilized the IncuCyte live-cell imaging system and conducted CyQuant cell proliferation assays to evaluate cell proliferation. However, our analysis revealed no detectable differences in the proliferation rates between the paired RT⁻ and RT⁺ fibroblasts (**new Fig. S1E, F**).

4. It's an interesting result that TGF β treatment had a muted effect in RT⁺ cells as far as activation/ECM production. How do you think this relates back to the functional changes seen in explant migration/wound healing? Further, THBS1 has been shown to be linked to latent TGF β activation, this might be worth mentioning in the discussion and could also explain the antifibrotic effects of THBS1 inhibition (<https://www.ncbi.nlm.nih.gov/pmc/articles/PMC6015530/>)

R2.4: TGF-13 is a crucial signal for wound healing, known to trigger fibroblast migration, contraction, and ECM production—essential functions for wound repair^{11, 12, 13}. Our findings indicate that, in response to TGF-13 treatment, RT⁺ fibroblasts exhibit a diminished response compared to RT⁻ fibroblasts, as evidenced by the lower induction of ACTA2 and ECM component expression (**new Fig. 1I, J, Fig. S1H, I**). The significance of fibroblast-synthesized ECM in wound healing is well-established, with fibronectin recognized for promoting fibroblast migration both autocrinally and paracrinally²⁴. ACTA2 serves as a marker for myofibroblasts, which are integral to wound contraction^{25, 26}. Therefore, the impaired expression of these critical genes in RT⁺ fibroblasts may contribute to their reduced migration and wound healing capabilities. We have expanded our discussion to include these insights. Moreover, we have provided a more detailed analysis of our results in the context of THBS1's established functions, such as its ability to activate latent TGF-13, which is further elaborated in **R2.8** (page 21, the 2nd paragraph).

5. Figure 1 demonstrates that RT⁺ fibroblasts have a decreased migration capacity, yet are shown in Figure 2 to highly express RUNX which is involved in cell migration. How do you reconcile these data?

R2.5: RUNX1 is a pivotal transcription factor (TF) essential for the proliferation and differentiation of mesenchymal progenitor cells¹⁴. Foster *et al.* have also demonstrated that fibroblasts become activated during wound healing through increased chromatin accessibility for RUNX1²⁷.

Our ATAC-seq data shows RT⁺ fibroblasts have enhanced RUNX motifs in open chromatin areas and increased differential binding for RUNX footprints compared to RT⁻ cells (**Fig. 2F, G**). Additionally, we performed Single Cell Multiome ATAC+Gene Expression platform on paired RT⁺ and RT⁻ skin during revision. This method reveals that, despite similar RUNX expression levels in RT⁺ and RT⁻ fibroblasts (**new Fig. S5D**), RUNX family motif activity was higher in RT⁺ cells, supporting RUNX's role in the long-term impact of radiotherapy (**new Fig. S5E**). This is corroborated by our new RNA-seq analysis showing comparable RUNX gene expression in both RT⁻ and RT⁺ fibroblasts (**new Fig. S2F**).

Gene Ontology analysis reveals RUNX1 targets in RT⁺ fibroblasts, like THBS1, DACH1, and GNAI12, negatively affect cell migration (**Fig. 2E, I, Table S5 and S7**). During revision, we also knocked down RUNX1 in both RT⁺ and RT⁻ fibroblasts, observing enhanced migration and ECM production in RT⁺ fibroblasts and improved healing in RT⁺ *ex vivo* wounds, without significantly affecting RT⁻ samples (**new Fig. 6M-P, Fig. S8I-M**).

These new findings, combined with existing knowledge, imply that RUNX1 predominantly governs the genes with epigenetic changes from radiotherapy. While RUNX1 is vital for wound healing, in RT⁺ fibroblasts, it appears to activate genes with epigenetic memory that undermine the fibroblasts' ability to repair wounds.

6. In the experiments treating RT⁻ fibroblasts to ionizing radiation, did you also examine the epigenome? Would be interesting to see if similar epigenetic alterations would be seen in this shorter timespan compared to cells from irradiated patient tissue.

R2.6: Following the Reviewer's advice, we performed ATAC-seq analysis in human fibroblasts exposed to ionizing radiation at a dose of 8 Gy and then being collected at one (D1) and seven days (D7) post-treatment (**new Fig. 2C**). These time points were chosen based on the expression dynamics of CDKN1A and PCNA in the treated fibroblasts, to capture the peak response to irradiation and the late stage of DNA damage repair, respectively (**Fig. 3H, I, new Fig. S4E, F**)^{1, 2}.

Our ATAC-seq analysis revealed increased chromatin accessibility in 2000 genes at D1 and 2765 genes at D7 compared to non-irradiated cells ($\log_2FC > 0$, $p\text{-value} < 0.05$) (**new Fig. 2C**). Notably, 23 of the 59 genes that showed increased accessibility in RT⁺ fibroblasts from cancer survivors (RT⁺ up) were more accessible post-irradiation in either D1 or D7 samples (**new Fig. 2D**). Additionally, the THBS1 gene exhibited increased accessibility only in D7-irradiated samples, not in D1, suggesting the epigenetic changes in the THBS1 gene occur during the DNA damage repair process (**new Fig. 2D**). Consistent with this, THBS1 expression was more markedly induced in irradiated cells than in non-irradiated cells when treated with TGF- β 3 six days post-irradiation. This difference was not observed in fibroblasts treated with TGF- β 3 three hours and one day post-irradiation (**new Fig. 3L, and new Fig. S4H**).

Despite complexities such as the differences between *in vivo* and *in vitro* fibroblast states and between clinical and experimental radiation exposures, these new ATAC-seq data suggest that some of the memory domains identified in cancer survivors are directly associated with ionizing radiation and can be observed in shorter timespan after radiation.

7. Regarding the ST data, a bit of a missed opportunity to take better advantage of the spatial component. Could you look at whether high concentrations of THBS1 expression correlate more with fibroblasts/PCvSMC etc at different timepoints, or with other microanatomical niches? May be a bit difficult to do with the low resolution of the 10X platform (which should be mentioned as a limitation), but there are methodologies to integrate sc-seq and ST, which would really enhance this data. Fig. 4E then could act as ST validation in supplementary.

R2.7: Heeding the Reviewer's advice, we employed the cell2location model-based probabilistic method²⁸ to deconvolute the ST data using our scRNA-seq results, which elucidated the spatial distribution of fibroblasts with single-cell precision (**new Fig. 4D**). Our analysis reveals a spike in

THBS1 expression in fibroblasts at Day 1 post-wounding, which then returns to baseline by Day 30 (**new Fig. 4E, F**). Additionally, we observed a positive correlation between THBS1 expression and the number of fibroblasts in human skin and wounds (**new Fig. 4G**). This expression trajectory is consistent with our qRT-PCR data on THBS1 from fibroblasts isolated from human skin and wound tissues (**Fig. 4C**) and aligns with THBS1 FISH results (**Fig. 4E, now relocated to Fig. S6G**).

8. THBS1 crosstalk is an important point, but missing a more thorough discussion on the impact of its increased signaling in RT+ skin and how this may impact clinical outcomes and inform immunotherapies outside of THBS1 neutralization.

R2.8: We concur with the Reviewer on the fundamental importance of THBS1 crosstalk in its functional capacities, and we recognize the necessity of comprehensively understanding this interaction network to ascertain its influence on wound healing in RT+ skin. To this end, we have expanded our analysis in the revised manuscript to include a detailed illustration of THBS1 expression and its receptors across different cell types within human wounds (**new Fig. 4I, new Fig. S6E**), building upon what was initially shown in Fig. 4H. Moreover, we have augmented the discussion with the following paragraph (page 21, the 2nd paragraph):

'Beyond affecting fibroblasts, the elevation of THBS1 in radiation-exposed skin likely exerts a substantial influence on a variety of cellular players and mediators that are crucial for wound healing. We found that THBS1, predominantly produced by fibroblasts, may interact with keratinocytes, angiogenic cells, and immune cells via receptors such as CD47 within human wounds. THBS1 is known to inhibit the proliferation, migration, and survival of vascular endothelial cells, restricting angiogenesis that is essential for wound repair²⁹. It also regulates the bioavailability of pivotal growth factors (like bFGF and VEGF) and enzymes (such as MMPs), thereby modulating ECM and various cellular functions³⁰. Moreover, THBS1 directly binds and activates latent TGF β 1, indicating that inhibiting THBS1 could offer a therapeutic approach for fibrotic conditions³¹. Notably, disrupting the THBS1-CD47 interaction has been demonstrated to protect normal tissues from the adverse effects of radiotherapy and chemotherapy by fostering protective autophagy and anabolic metabolic repair, simultaneously enhancing the immune destruction of cancer cells³². Given its role as an innate immune checkpoint, CD47 has emerged as a central target in the development of cancer immunotherapies³³. The growing interest in CD47 inhibitors is driven by their dual potential: enhancing the immune system's response against tumors and protecting healthy tissue during and after radiation therapy. Our findings further reinforce the latter benefit.'

Minor comments:

9. Fig. 1C is a bit confusing – the edge of fluorescent staining suggests that the wounds are actually closing faster than the dotted line would suggest.

R2.9: In this figure, we aim to direct the readers' focus towards wound contraction, as indicated by the alterations of the original wound margins, which are demarcated by dotted lines. The areas of fluorescent staining inside these dotted lines on Days 2 and 5 represent the newly formed epithelial tongues, indicative of the re-epithelialization process²⁰. We have provided further clarification on this point in the revised manuscript (see the revised Methods section, under the sub-title 'Human *ex vivo* wound model').

10. Line 137 – are the 97 less accessible peaks necessarily RT- up, or are they just the least accessible in the RT+ fibroblasts? I'm assuming they are the highest accessible in the RT- group based on Fig 2D, but needs clarification.

R2.10: In our analysis, we discerned 97 chromatin domains that exhibited reduced accessibility in RT+ fibroblasts relative to RT- fibroblasts. The revised text now reads: '...we identified 74 peaks showing increased accessibility [$\log_2(\text{fold change, FC}) > 0$, false discovery rate (FDR) < 0.05, denoted as RT+ up domains] and 97 peaks with decreased accessibility ($\log_2\text{FC} < 0$, FDR < 0.05, denoted as RT- up domains) in RT+ fibroblasts in comparison to RT- fibroblasts.'

11. Line 238 – a bit unclear whether this group performed these experiments based on the wording

R2.11: We have revised this text as: 'Our study reveals the lasting impact of RT on wound healing in breast cancer survivors, supporting our recent findings on the extended effects of radiation on wound repair observed in a mouse model⁴³. ' (page 15, the 2nd paragraph)

12. Line 282 - Please spell out late-onset adverse effects (LAE) **R2.12:** We have revised this text as suggested (page 18, the 4th paragraph).

13. Discussion line 348: might be more accurate to say that we can 'address' the maladaptive radiation memory vs 'erasing' it.

R2.13: We have revised this text as suggested (page 22, the 2nd paragraph).

Reviewer #3, expertise in molecular mechanisms of radiotherapy effects:

In their manuscript, "Epigenetic memory of radiotherapy in dermal fibroblasts impairs wound repair capacity in cancer survivors," Piipponen, Bian et al. provide a valuable molecular and functional study to help understand the mechanisms of late-onset radiotherapy toxicities with focus on epigenetic regulators. By comparing paired skin biopsies from irradiated and non-irradiated sites in breast cancer survivors, the authors found impaired wound healing and impaired fibroblast function after radiotherapy and demonstrated solid evidence of a persistently altered chromatin landscape in irradiated fibroblasts. The authors identified THBS1 as a potential mediator of this phenotype fibroblasts and validated its importance in mouse and human wound models. Remarkably, the authors demonstrated the ability of anti-THBS1 antibodies to promote wound closure in an ex vivo irradiated skin model. **Overall, this work will be of interest to the fields of cancer and radiation biology, providing important insights into the mechanisms of late radiation toxicity and demonstrating the clinical promise of targeting these processes to reverse the long-term side effects of radiotherapy.** However, I have several concerns that should be addressed and suggestions to strengthen the findings prior to publication.

1. The time from end of radiotherapy to tissue collection varies from 1-12 years, which is a relatively large range. According to Table S1, different patients were selected for various experiments in the study. Did the authors consider any criteria for assigning patients to different experiments? For example, only five patients (line 132) with a narrower range of time since the end of radiation were assigned to the ATAC-seq experiment. Could authors include additional cases (such as cases 37 and 13, collected 120 and 129 months after radiation, respectively) to compare how epigenetic changes persist at later timepoints after radiotherapy?

R3.1: In our initial approach, we did not impose specific selection criteria for assigning patients to different experimental groups, as the study unfolded concurrently with sample collection. A significant majority of the patients enrolled in this study, approximately 84%, were within 18-60 months post-radiotherapy, a group that is well-represented in our ATAC-seq analysis. Pursuant to the Reviewer's suggestion, we expanded our bulk ATAC-seq to include fibroblasts from cases 17 and 26, which were collected 108 and 84 months post-radiotherapy, respectively. (We don't have enough cells left for the cases 13 and 37.) Our data show that the epigenetic alterations are still present at these samples with extended intervals after radiotherapy (**new Fig. S2A**). We did not find a clear correlation between the overall differences in chromatin accessibility between RT⁺ and RT⁻ fibroblasts and the time elapsed post-radiotherapy (**new Fig. S2B**). Moreover, our revision included analysis of RT⁺ and RT⁻ skin from a cancer survivor 81 months post-radiotherapy, using the single-cell Multiome ATAC+Gene Expression platform. This provided simultaneous gene expression and epigenomic profiles within individual cells of the patient's skin. Notably, we found increased accessibility of the THBS1 promoter region in RT⁺ fibroblasts compared to RT⁻ cells, corroborating our bulk ATAC-seq findings (**new Fig. S5C**).

2. The authors identified THBS1 by analyzing genes with altered chromatin accessibility in irradiated fibroblasts that were also differentially expressed during the normal wound-healing process (Day1 or Day7 wounds vs. donor-matched skin). This seems counter-intuitive because they were interested in genes that lead to impaired wound healing. Can the authors explain their rationale for this approach? What genes are differentially expressed during wound healing in irradiated versus non-irradiated skin?

R3.2: The genes with altered chromatin accessibility in RT⁺ fibroblasts may contribute to various functional impairments in late-onset adverse effects (LAEs), necessitating further investigation in different pathological contexts. To prioritize the key effectors of radiotherapy's epigenetic memory impacting wound healing, we focused on genes showing expression changes (either up- or down-regulation) during wound repair. We hypothesize that genes without expression changes are less likely to be involved in wound repair. As a result, we identified 16 genes with increased chromatin accessibility in RT⁺ fibroblasts that also exhibit expression changes during wound healing (**new Fig. 2K, L**). The epigenetic changes in these genes are thus more likely to disrupt the wound healing process.

Although comparing gene expression during wound healing between irradiated and non-irradiated skin would be ideal, acquiring post-wounding RT⁺ versus RT⁻ skin samples from clinical settings is not practical. In our revision, we conducted RNA-seq on paired RT⁺ and RT⁻ fibroblasts and found that 93% of genes with altered chromatin accessibility in RT⁺ cells were either not expressed or not differentially expressed compared to RT⁻ cells, suggesting that the epigenetic memory of

radiotherapy was not unfolded at the homeostatic cell state (**new Fig. S3A, B**). Therefore, we treated the RT⁺ and RT⁻ fibroblasts with TGF- β , an important signal for fibroblasts in wound healing^{11, 12, 13}, and performed RNA-seq analysis (**new Fig. S3C**). TGF- β treatment activated 14 of the 59 genes with poised chromatin post-radiotherapy, with six overlapping with genes differentially expressed in human wounds (**new Fig. 2K, L**). This strategy allowed us to pinpoint THBS1 as a poised gene, exhibiting a more robust TGF β -induced expression in RT⁺ fibroblasts compared to RT⁻ cells (**Fig. 3F, new Fig. S3C**).

3. For most of their histological analyses, the authors utilized FISH for THBS1 rather than immunofluorescence for the THBS1 protein. Although they demonstrated significant differences by quantification, it is difficult to appreciate significant differences in the presented images. It may be easier to appreciate the differences if the THBS1 and DAPI images are displayed separately instead of overlaid. In Figure S4, the authors show that they can perform immunofluorescence for the THBS1 protein, but the difference between groups appears less substantial. It would be helpful to show that protein levels of THBS1 are significantly different in their mouse model (Fig. 5B) and/or in radiation ulcers (Fig. 5F).

R3.3: We opted for FISH over immunofluorescence (IF) because FISH detects THBS1 mRNA, allowing for a more sensitive detection of THBS1 synthesis during wound healing. In the revised study, we conducted IF using a new THBS1 antibody with improved specificity on both the mouse model and human radiation ulcers (**new Fig. 5D, H, new Fig. S7B**). Following Reviewer's suggestion, we've also separated the THBS1 and DAPI images in FISH and IF experiments to enhance visualization (**new Fig. 5B, D, G, H, new Fig. S7A-C**). Moreover, the adoption of a dual-staining technique for Thbs1 with the fibroblast marker Pdgfra, along with refinements to the Thbs1 FISH protocol, has markedly improved the clarity of Thbs1 FISH images in the revised manuscript (**new Fig. 5B, new Fig. S7A**).

4. The authors provide impressive data that an anti-THBS1 antibody promotes wound healing after radiation exposure in *ex vivo* wounds. However, it is unclear how the control group was treated in the anti-THBS1 antibody experiment in Fig. 6E. Was the gel that was mixed with the THBS1 antibody added to the controls? It is possible that the gel promotes wound healing in the absence of the antibody. It would substantially strengthen the manuscript if the authors could demonstrate that the anti-THBS1 antibody promotes wound healing *in vivo* in their murine wound model.

R3.4: We appreciate the Reviewer's input regarding the anti-THBS1 antibody treatment experiment (**Fig. 6E, now relocated at Fig. 6L**), where both the experimental and control groups were treated with F-127 gel. This has been clarified in the manuscript's results section as follows: 'To evaluate this, we mixed THBS1 antibody dissolved in PBS or PBS buffer alone with 30% pluronic F-127 gel. Then we topically applied the gel with or without THBS1 antibody to human *ex vivo* wounds created on both RT⁺ and RT⁻ skin (Fig. 6L, Fig. S8H)'. The methodology is now more thoroughly detailed in the 'Human *ex vivo* wound model' section under Methods.

We acknowledge the Reviewer's point on the significance of assessing the therapeutic and preventative efficacy of THBS1 antibody using a murine *in vivo* model as a critical step towards clinical application. This necessitates testing new anti-mouse THBS1 antibodies, as our current anti-human THBS1 antibody does not cross-react with murine tissues and optimizing dosage and

timing for these new antibodies. The complexities of establishing the murine irradiated skin and excisional wound model also present a substantial challenge, which cannot be addressed within the timeframe of this revision. In addition, our revision uncovered that silencing RUNX1—the transcription factor responsible for THBS1 expression and other genes related to epigenetic memory—can hasten *ex vivo* wound healing (**new Fig. S8I, M**). Furthermore, disrupting the THBS1-CD47 interaction presents a promising avenue for radioprotective therapies, aligning with the growing focus on developing CD47 inhibitors for cancer immunotherapies (discussed in detail at page 21, the 2nd paragraph). A follow-up study, given sufficient time and resources, is essential to thoroughly investigate these diverse strategies for altering the epigenetic memory of radiotherapy in both human and murine models. This research will further solidify the basis for clinical translation.

5. The authors provide compelling evidence for THBS1's role in delayed wound healing after radiotherapy. However, the downstream mechanism of how increased THBS1 expression blocks fibroblast migration, contraction, and wound healing is unclear. Can the authors comment on why THBS1 expression could affect these processes?

R3.5: THBS1 is a multifunctional molecule with context-dependent roles. To elucidate the mechanisms by which increased THBS1 expression influences fibroblast activities and wound healing, we conducted RNA-sequencing analysis on human fibroblasts with THBS1 expression activated by the CRISPR/dCas9-SAM system (**new Fig. S8C**). This analysis revealed 45 genes upregulated and 138 genes downregulated (p -value <0.05) following THBS1 overexpression (OE). Gene Ontology analysis indicated that biological processes like epithelial-mesenchymal transition, TGF- β signaling, inflammation (TNF α and IL6 signaling), apoptosis, and hypoxia were predominantly enriched among the downregulated genes, while only the interferon gamma response was enriched among the upregulated genes post THBS1 OE (**new Fig. S8C**). Further, through qRT-PCR, we confirmed decreased expression of ECM-related genes (FN1, ELN, COL3A1, and COL1A1) and ACTA2 in fibroblasts with THBS1 OE (**new Fig. 6B-F**). The significance of fibroblast-produced ECM in wound healing is well-documented, particularly fibronectin's role in enhancing fibroblast migration²⁴. ACTA2 serves as a marker for myofibroblasts, which are integral to wound contraction^{25, 26}.

Additionally, we evaluated the potential impacts of key signaling pathways, including p38, JNK, ERK, PKC, STAT3, PI3K, and EGFR, on THBS1 OE effects. Our findings indicate that an ERK inhibitor most effectively counteracts the suppressive influence of THBS1 OE on ECM gene and ACTA2 expression (**new Fig. S8D-E**). Blocking ERK signaling also mitigated the reduced migration of fibroblasts associated with THBS1 OE (**new Fig. 6K**). These results highlight ERK as a pivotal downstream signal mediating THBS1's effects. Supporting this, we observed that THBS1 OE activates ERK signaling, evidenced by increased phosphorylation of ERK1/2, as demonstrated in our Western blotting results (**new Fig. S8F**). Collectively, our results demonstrate that enhanced THBS1 expression leads to notable alterations in gene expression and signaling pathways critical to fibroblast function and wound healing.

Minor Notes

1. Line 94: The authors state that patients underwent external beam radiotherapy to a total dose of 50 Gy. However, the methods and Table S1 show that there was a range of doses from 40 to 60 Gy. This should be consistent.

R3.6: We have revised the text in the 1st paragraph of the result section as 'These individuals had previously received external beam RT with a total dose of 40-60 Gy at least one year prior to the reconstruction surgery.'

2. How were the wounds created for the human wound healing model? Can the authors clarify the difference between full depth and partial-thickness wounds?

R3.7: We have revised the text in the methods section to clarify how the human wound healing model was created and the difference between full depth and partial-thickness wounds in the human *in vivo* and *ex vivo* wound models, respectively.

In our human wound healing model, we crafted three full-depth wounds, each 3 mm in diameter and extending into the subcutaneous adipose tissue, on the skin of healthy volunteers using a biopsy punch equipped with a circular blade. Post-wounding, the donors returned to our clinic on three occasions: day one (D1), day seven (D7), and day thirty (D30). During each visit, we harvested tissue from the periphery of the wounds using a 6 mm biopsy punch, selecting a different one of the three wounds at each time point for tissue collection.

In our study of an *ex vivo* human wound model, we utilized a biopsy punch with a circular blade (2 mm in diameter) to create partial thickness wounds, ensuring that the wounds did not extend below the dermis layer, on RT⁻ and RT⁺ human skin collected post-surgery. These wounds were excised from the skin using a 6 mm biopsy punch. After the subcutaneous fat was removed, the excised wound tissues were then placed into a 12-well cell culture plate.

3. For the GO enrichment networks in Figure 2C and S1, it would be helpful to explain what the size and colors of the nodes represents in the legend.

R3.8: The color coding of each node in the diagram corresponds to different GO (Gene Ontology) biological process categories. The size of each node reflects the quantity of genes that are enriched in the respective biological process. We have revised the figure legend to better explain these elements.

4. For the scRNA-seq analysis in Figure 4A, the authors identified 26 cell types based on their clustering strategy, including 3 basal, 3 spinous, and 2 granular keratinocyte populations and 4 fibroblast populations. Are these sub-populations biologically meaningful or just a byproduct of the resolution chosen for their clustering approach? Have these sub-populations been described previously in the literature and is there a difference in THBS1 expression between them?

R3.9: These sub-populations are biologically significant, as evidenced by their distinct marker gene expression profiles (**Figure 3 for reviewers**). Similar cell states have been previously documented in literature [34, 35 for fibroblasts, 36, 37 for keratinocytes]. In a separate project analysing human skin wound samples through both scRNA-seq and spatial transcriptomics, we explored their

differentiation states and functional characteristics. This extensive dataset and related analysis will be the subject of a forthcoming manuscript, which is not included here due to the spatial constraints and specific research focus of the current study.

In the revised version of this manuscript, we have incorporated **new Fig. S6E**, which illustrates the expression of THBS1 and its receptors across various cell types in human wounds, as revealed by scRNA-seq analysis. We observed that THBS1 expression is predominantly in fibroblasts, with lower levels in macrophages and angiogenic cells. Within the fibroblast sub-clusters, FB-III (papillary fibroblasts) showed the highest expression of THBS1, while FB-IV (proliferating fibroblasts) had the least expression (**new Fig. S6F**)^{34, 35}.

Figure 3. Dot plot showing the expression of distinctive markers for each cell population.

5. Why isn't there a D0 timepoint for the 45 Gy treatment in Figure 5C?

R3.10: We have included these samples (D0 timepoint for the 45 Gy treatment) and performed the Thbs1 and Pdgfra co-staining in the revised manuscript (**new Fig. 5B, Fig. S7A**) (see also **R3.3**).

References

1. Dabin J, Fortuny A, Polo SE. Epigenome Maintenance in Response to DNA Damage. *Mol Cell* **62**, 712-727 (2016).
2. Hunt CR, *et al.* Histone modifications and DNA double-strand break repair after exposure to ionizing radiations. *Radiat Res* **179**, 383-392 (2013).
3. Diaz C, *et al.* Ionizing Radiation Mediates Dose Dependent Effects Affecting the Healing Kinetics of Wounds Created on Acute and Late Irradiated Skin. *Surgeries* **2**, 35-57 (2021).
4. Agah A, Kyriakides TR, Lawler J, Bornstein P. The lack of thrombospondin-1 (TSP1) dictates the course of wound healing in double-TSP1/TSP2-null mice. *Am J Pathol* **161**, 831-839 (2002).
5. DiPietro LA, Nissen NN, Gamelli RL, Koch AE, Pyle JM, Polverini PJ. Thrombospondin 1 synthesis and function in wound repair. *Am J Pathol* **148**, 1851-1860 (1996).

1. Joost S, *et al.* Single-Cell Transcriptomics of Traced Epidermal and Hair Follicle Stem Cells Reveals Rapid Adaptations during Wound Healing. *Cell Rep* **25**, 585-597 e587 (2018).
2. Raugi GJ, Olerud JE, Gown AM. Thrombospondin in early human wound tissue. *J Invest Dermatol* **89**, 551-554 (1987).
3. Correa-Gallegos D, *et al.* CD201(+) fascia progenitors choreograph injury repair. *Nature* **623**, 792-802 (2023).
4. Vu R, *et al.* Wound healing in aged skin exhibits systems-level alterations in cellular composition and cell-cell communication. *Cell Rep* **40**, 111155 (2022).
5. Muhl L, *et al.* Single-cell analysis uncovers fibroblast heterogeneity and criteria for fibroblast and mural cell identification and discrimination. *Nat Commun* **11**, 3953 (2020).
6. Knoedler S, *et al.* Fibroblasts - the cellular choreographers of wound healing. *Front Immunol* **14**, 1233800 (2023).
7. Lichtman MK, Otero-Vinas M, Falanga V. Transforming growth factor beta (TGF-beta) isoforms in wound healing and fibrosis. *Wound Repair Regen* **24**, 215-222 (2016).
8. Wang XJ, Han G, Owens P, Siddiqui Y, Li AG. Role of TGF beta-mediated inflammation in cutaneous wound healing. *J Investig Dermatol Symp Proc* **11**, 112-117 (2006).
9. Kim W, *et al.* RUNX1 is essential for mesenchymal stem cell proliferation and myofibroblast differentiation. *Proc Natl Acad Sci U S A* **111**, 16389-16394 (2014).
10. Klunker S, *et al.* Transcription factors RUNX1 and RUNX3 in the induction and suppressive function of Foxp3+ inducible regulatory T cells. *J Exp Med* **206**, 2701-2715 (2009).
11. Ito Y, Miyazono K. RUNX transcription factors as key targets of TGF-beta superfamily signaling. *Curr Opin Genet Dev* **13**, 43-47 (2003).
12. Pal SK, *et al.* THBS1 is induced by TGFB1 in the cancer stroma and promotes invasion of oral squamous cell carcinoma. *J Oral Pathol Med* **45**, 730-739 (2016).
13. Scherer SD, *et al.* TGF-beta1 Is Present at High Levels in Wound Fluid from Breast Cancer Patients Immediately Post-Surgery, and Is Not Increased by Intraoperative Radiation Therapy (IORT). *PLoS One* **11**, e0162221 (2016).
14. Li X, Xu Landen N. Evaluation of MicroRNA Therapeutic Potential Using the Mouse In Vivo and Human Ex Vivo Wound Models. *Methods Mol Biol* **2193**, 67-75 (2021).
15. Nasir NAM, Paus R, Ansell DM. Fluorescent cell tracer dye permits real-time assessment of re-epithelialization in a serum-free ex vivo human skin wound assay. *Wound Repair Regen* **27**, 126-133 (2019).
16. Wilkinson HN, Kidd AS, Roberts ER, Hardman MJ. Human Ex vivo Wound Model and Whole-Mount Staining Approach to Accurately Evaluate Skin Repair. *J Vis Exp*, (2021).
17. Naik S, *et al.* Inflammatory memory sensitizes skin epithelial stem cells to tissue damage. *Nature* **550**, 475-480 (2017).

18. Rocliffe H, *et al.* MC1R reduces scarring and rescues stalled healing in a novel preclinical chronic wound model. *bioRxiv*, 2022.2011.2030.518516 (2023).
19. Li X, *et al.* Human dermal fibroblast migration induced by fibronectin in autocrine and paracrine manners. *Exp Dermatol* **23**, 682-684 (2014).
20. Desmouliere A, Chaponnier C, Gabbiani G. Tissue repair, contraction, and the myofibroblast. *Wound Repair Regen* **13**, 7-12 (2005).
21. Ibrahim MM, *et al.* Myofibroblasts contribute to but are not necessary for wound contraction. *Laboratory Investigation* **95**, 1429-1438 (2015).
22. Foster DS, *et al.* Integrated spatial multiomics reveals fibroblast fate during tissue repair. *Proc Natl Acad Sci U S A* **118**, (2021).
23. Kleshchevnikov V, *et al.* Cell2location maps fine-grained cell types in spatial transcriptomics. *Nature Biotechnology* **40**, 661-671 (2022).
24. Streit M, *et al.* Thrombospondin-1 suppresses wound healing and granulation tissue formation in the skin of transgenic mice. *EMBO J* **19**, 3272-3282 (2000).
25. Kyriakides TR, Maclachlan S. The role of thrombospondins in wound healing, ischemia, and the foreign body reaction. *J Cell Commun Signal* **3**, 215-225 (2009).
26. Murphy-Ullrich JE, Suto MJ. Thrombospondin-1 regulation of latent TGF-beta activation: A therapeutic target for fibrotic disease. *Matrix Biol* **68-69**, 28-43 (2018).
27. Isenberg JS, Roberts DD. THBS1 (thrombospondin-1). *Atlas Genet Cytogenet Oncol Haematol* **24**, 291-299 (2020).
28. Zhao H, *et al.* CD47 as a promising therapeutic target in oncology. *Front Immunol* **13**, 757480 (2022).
29. Deng C-C, *et al.* Single-cell RNA-seq reveals fibroblast heterogeneity and increased mesenchymal fibroblasts in human fibrotic skin diseases. *Nature Communications* **12**, 3709 (2021).
30. Philippeos C, *et al.* Spatial and Single-Cell Transcriptional Profiling Identifies Functionally Distinct Human Dermal Fibroblast Subpopulations. *J Invest Dermatol* **138**, 811-825 (2018).
31. Haensel D, *et al.* Defining Epidermal Basal Cell States during Skin Homeostasis and Wound Healing Using Single-Cell Transcriptomics. *Cell Rep* **30**, 3932-3947 e3936 (2020).
32. Wang S, *et al.* Single cell transcriptomics of human epidermis identifies basal stem cell transition states. *Nat Commun* **11**, 4239 (2020).

Reviewers' Comments:

Reviewer #1:

Remarks to the Author:

Rebuttal to Bian, Piipponen et al.

Bian, Piipponen et al. addressed some concerns raised during the first round of review, and their added experiments and clarifications generally improve the rigor of the initial submission. This being said, some important issues have not yet been resolved and are essential in order to justify the conclusions drawn and merit publication in Nature Communications.

Major Points:

1. The role of radiation, as opposed to e.g. previous cancer or surgery, in specifically mediating lasting epigenetic changes is still unclear and addressing this is central to the paper.
 - The authors say that previously irradiated (RT+) skin was only taken from areas within irradiated sites, but distant from previous tumors and mastectomy scars (Fig. S1A). However, the locations of these previous tumors are not made clear, and the mastectomies may have impacted skin tissue cells within larger affected areas of the resection procedures. The authors should perform ATAC-seq on dermal fibroblasts from patients who received a mastectomy alone (if feasible), or at least from mice post-irradiation but prior to wounding. This was suggested in the first round of revisions but it was not discussed in rebuttals. These assays will allow for assessment of long-term epigenetic changes due specifically to irradiation, which are essential for the authors' claims but still remain missing in the study.
2. Fig. 2C: The authors present an in vitro system to argue that healthy human dermal fibroblasts can mount an enduring epigenetic response to acute irradiation alone, lasting through at least D7 post-radiation. However, the rationale for choosing D7 as a post-radiation timepoint is based only on resolution of DNA damage-repair genes (*Cdkn1a* and *Pcna*, Fig. 3H/I and argued in rebuttal). The authors should repeat their post-irradiation ATAC-seq at a later timepoint, especially since fibroblasts seem to be engaging a continued, robust epigenetic response at D7 (~2800 regions with heightened accessibility v. ~2000 regions at D1, and only 59 in the analogous clinical post-irradiation RT+ fibroblasts).
3. The authors emphasize THBS1 as a central "memory gene" in functionally driving long-term impairment of wound healing. However, this role/the underlying regulation of THBS1 needs to be shown more thoroughly to justify this conclusion:
 - The authors claim that THBS1 remains in a heightened state of chromatin accessibility post-irradiation (Fig. 2D,3B), allowing for rapid (Fig. 3F,M) and persistent (Fig. 5C,E) transcription upon wounding that then confers delayed healing. However, these arguments are made across separate models, which need to be resolved.
 - Human:
 - The expression dynamics of THBS1 in the presented ex vivo and in vivo wounding studies appear contradictory. I.e., the authors observe no induction of THBS1 in normal (RT-) skin upon wounding ex vivo (Fig. S7E), but observe its transient induction in vivo (Fig 4B,C,F). This contradiction should be discussed in the manuscript, and more generally suggests that the presented murine model may be more representative of THBS1 dynamics upon wounding than the human ex vivo system, as the

former better recapitulates the transient wound-induced activation of THBS1 (Fig. S6). If so, the text needs to be revised to refrain from overinterpretation of the results.

- Through all the manuscript, the authors use TGF- β in vitro as a key mediator in wounding to infer differences in wound-response potential between RT+ and RT- fibroblasts. In Figure S3, they show 5 module genes where THBS1 is located in cluster M3 (by looking at Table S9). This module represents genes downregulated in RT- fibroblasts upon TGF- β , but that remain constant in RT+ with or without TGF- β . However, in Figure 3F, they did qPCR for THBS1 in presence and absence of TGF- β and showed a contradictory result whereby THBS1 is uniquely induced 24h after TGF- β in RT+ but not RT- fibroblasts. The timepoint at which the RNA-seq was performed is not specified, and more generally the authors should clarify these differences between RNA-seq and qPCR regarding THBS1 in the presence/absence of TGF- β .
- Fig. S8G: Considering that the authors show THBS1 is induced as a response to wounding and critical for long-term delays in wounding capacity, it seems strange that human wounds close at the same rate in RT- fibroblasts when treated with control or with THBS1 antibody. A satisfactory explanation as to why THBS1 may be dispensable for normal wound-healing in these contexts needs to be presented and discussed.
- Murine model: The authors characterize expression levels of THBS1 from D0-D10 in normal mouse wounding, both in whole-skin (Fig. S6C) and fibroblasts (Fig. S6D). How these patterns/expression levels are altered in post-irradiation settings (beyond %THBS1+ cells and prior to D33) is missing and should be addressed.

- Fig 3B: The presented tracks/peak-calls (black bars) around THBS1 look a bit strange:
- ATAC signal over the entire locus appears higher in RT+ fibroblasts (as opposed to within specific sub-domains), and peak calls do not overlay clearly distinguished peaks, and not all peaks appear consistently differential across respective RT+ vs. RT- comparisons. Given the importance of THBS1 toward the study, the authors should provide a table summarizing the read coverage over this locus and statistical test. The authors should also widen their track over a larger area to determine whether this overall increase in signal in RT+ cells is truly specific to the THBS1 locus (as the scATAC in Fig. S5 suggests), e.g. to mirror the ~Chr15:39850000-39600000 region shown in the scATAC. The method used to normalize bulk ATAC-seq read coverage for bigWigs should also be reported in the methods section.

Minor Points:

1. Fig. 2b: Distribution of accessible genomic sites from RT+ and RT- in pie charts do not represent reported values. Figure should be remade.

2. Fig. 2k: Replace 'and' by 'or' since they are representing union of DE genes in D1 with D7 in 'DE genes in human skin wound healing D1 and D17'.

3. Fig. 5A: The mouse irradiated skin and wound model still needs further characterization.

- The authors used a citation to state that there is significantly delayed wound healing in irradiated mouse skin compared to non-irradiated mouse skin. Given the importance of this model toward the study, the authors should add a figure showing delayed wound healing in their mouse model rather than only a citation.

4. The authors may consider rewording a few terms/phrases for clarity:

- “radiation wound models” (used throughout) should stress that wounding was only performed after acute irradiation (e.g. “post-radiation wound models”).
- “skin cells” (p5) may read as keratinocytes/epidermal stem cells rather than a variety of potential in the skin tissue, given its placement immediately after highlighting memory in epidermal stem cells (e.g. “...a variety of surviving skin cells after acute radiation...”).

Reviewer #2:

Remarks to the Author:

Thank you to the authors for the thoughtful and thorough resolution of the concerns raised in the initial review. Especially appreciated the quantification of collagen alignment in RT+/- skin biopsies, the expanded epigenetic analysis and ST deconvolution. The additional discussion of THBS1 and its multiple roles in crosstalk and impact on cellular functions has further improved the take-home message of the data. Overall, this paper has been significantly strengthened following revision, and this reviewer believes it will be a notable contribution to the field.

Reviewer #3:

Remarks to the Author:

I appreciate the authors' response to my comments, and I think the additions have substantially strengthened the manuscript. This work provides important insights into the mechanisms of late radiation toxicity and suggests promising approaches to reverse these long-term side effects. At this time, I do not have any further concerns, and I think the manuscript is suitable for publication.

Point by point response:

Bian, Piipponen et al. addressed some concerns raised during the first round of review, and **their added experiments and clarifications generally improve the rigor of the initial submission**. This being said, some important issues have not yet been resolved and are essential in order to justify the conclusions drawn and merit publication in Nature Communications.

We are immensely thankful to Reviewer #1 for the exceptional recommendations.

In response, we have conducted substantial new experiments and analysis. These include ATAC-seq on human fibroblasts from areas with and without previous surgery in three breast cancer survivors, representing very rare and difficult-to-collect clinical cases. Additionally, we performed ATAC-seq on *in vitro* irradiated human fibroblasts at two weeks post-irradiation. We also set up a murine radiation and wound healing model, performing ATAC-seq on murine dermal fibroblasts from irradiated and non-irradiated areas at 1- and 7-days post-radiation, along with qRT-PCR analysis of dermal *Thbs1* expression during wound healing in previously irradiated and non-irradiated murine skin. Moreover, we analyzed THBS1 expression in additional human *ex vivo* wound samples to evaluate THBS1 expression dynamics in this model. These findings have significantly solidified our conclusions and enhanced our understanding of epigenetic memory.

Major Points:

1. The role of radiation, as opposed to e.g. previous cancer or surgery, in specifically mediating lasting epigenetic changes is still unclear and addressing this is central to the paper.

- The authors say that previously irradiated (RT+) skin was only taken from areas within irradiated sites, but distant from previous tumors and mastectomy scars (Fig. S1A). However, the locations of these previous tumors are not made clear, and the mastectomies may have impacted skin tissue cells within larger affected areas of the resection procedures. The authors should perform ATAC-seq on dermal fibroblasts from patients who received a mastectomy alone (if feasible), or at least from mice post-irradiation but prior to wounding. This was suggested in the first round of revisions but it was not discussed in rebuttals. These assays will allow for assessment of long-term epigenetic changes due specifically to irradiation, which are essential for the authors' claims but still remain missing in the study.

R1: We appreciate the Reviewer's emphasis on distinguishing the effects of radiation from those of previous cancer or surgery. Due to ethical and surgical constraints together with limited access to the requested clinical material, we have put down considerable time to recruit suitable patients for the task, which has delayed the resubmission. Following the suggestion, we performed ATAC-seq on dermal fibroblasts from skin areas with surgery 9-25 months ago but no radiotherapy (S+) and from areas without surgery and radiotherapy (S-) in three breast cancer survivors. These samples are very rare, and our clinical team made significant efforts to collect them during the revision period.

The ATAC-seq analysis revealed that previous surgery induced increased chromatin accessibility in 16 genes, with only one gene, HOXA3, among the 59 genes with the RT+ up- domains (**new Fig. S2F, new Table S4**). Notably, THBS1 locus accessibility did not differ between S+ and S- fibroblasts, suggesting that **the lasting epigenetic changes in RT+ fibroblasts are unlikely due to previous surgery**.

Additionally, we conducted focal irradiation (IR) on the shaved back skin of C57BL/6 mice (see revised method section, page 41-42, line 749-762, **new Fig. S7C, D**). ATAC-seq was performed on dermal cells isolated from irradiated (IR+) and non-irradiated (IR-) skin one (D1) and seven days (D7) after IR. We identified 3675 genes and 9913 genes with increased accessibility in IR+ dermal cells compared to IR- cells ($\log_2FC > 0$, $p\text{-value} < 0.05$) on D1 and D7, respectively (**new Fig. S7G, new Table S12**). Notably, 34 of the 59 genes with the RT+ up domains in human fibroblasts, including THBS1, showed greater accessibility in murine cells post-irradiation.

Despite challenges such as differences between fibroblasts' *in vivo* and *in vitro* states, human-mouse variations, and differences in clinical and experimental radiation exposures, **these new ATAC-seq data confirm that the lasting epigenetic changes identified in cancer survivors' fibroblasts are directly associated with radiation.**

2. Fig. 2C: The authors present an *in vitro* system to argue that healthy human dermal fibroblasts can mount an enduring epigenetic response to acute irradiation alone, lasting through at least D7 post-radiation. However, the rationale for choosing D7 as a post-radiation timepoint is based only on resolution of DNA damage-repair genes (Cdkn1a and PcnA, Fig. 3H/I and argued in rebuttal). The authors should repeat their post-irradiation ATAC-seq at a later timepoint, especially since fibroblasts seem to be engaging a continued, robust epigenetic response at D7 (-2800 regions with heightened accessibility v. -2000 regions at D1, and only 59 in the analogous clinical post-irradiation RT+ fibroblasts).

R2: Following the reviewer's suggestions, we repeated the experiment and conducted ATAC-seq on human dermal fibroblasts two weeks (D14) after irradiation (IR). The ATAC-seq analysis revealed increased chromatin accessibility in 1441 genes in D14 IR+ fibroblasts compared to D14 IR- cells ($\log_2FC > 0$, $p\text{-value} < 0.05$) (**new Fig. 2C, new Table S5**). Notably, 9 out of 59 genes with RT+ up domains, including THBS1, maintained greater accessibility post-irradiation, indicating that the epigenetic response can persist for at least two weeks. Although the impact of IR diminishes at D14 compared to D7 (1441 vs. 2765 genes with increased chromatin accessibility induced by IR), it is more robust compared to RT+ fibroblasts from cancer survivors, which show only 59 genes with RT+ up domains. However, extending the culture period further (e.g., to 4 weeks post-IR) revealed signs of senescence in human primary fibroblasts, even in IR- controls. Therefore, it is challenging to find a later time point in this *in vitro* system to accurately reflect the resolving phase of IR effects observed in cancer survivors.

3. The authors emphasize THBS1 as a central "memory gene" in functionally driving long-term impairment of wound healing. However, this role/the underlying regulation of THBS1 needs to be shown more thoroughly to justify this conclusion:

- The authors claim that THBS1 remains in a heightened state of chromatin accessibility post-irradiation (Fig. 2D,3B), allowing for rapid (Fig. 3F,M) and persistent (Fig. 5C,E) transcription upon wounding that then confers delayed healing. However, these arguments are made across separate models, which need to be resolved.
- Human: The expression dynamics of THBS1 in the presented *ex vivo* and *in vivo* wounding studies appear contradictory. I.e., the authors observe no induction of THBS1 in normal (RT-) skin upon

wounding *ex vivo* (Fig. S7E), but observe its transient induction *in vivo* (Fig 4B,C,F). This contradiction should be discussed in the manuscript, and more generally suggests that the presented murine model may be more representative of THBS1 dynamics upon wounding than the human *ex vivo* system, as the former better recapitulates the transient wound-induced activation of THBS1 (Fig. S6). If so, the text needs to be revised to refrain from overinterpretation of the results.

R3.1: We appreciate the reviewer highlighting this important point. Previously, we measured dermal THBS1 expression by qRT-PCR in human RT⁻ vs. RT⁺ *ex vivo* wounds from two donors without technical replicates. In the revised manuscript, we assessed THBS1 expression in human *ex vivo* wound dermis from additional donors (n=4) and included more technical replicates (n=1-3) per experiment, increasing the reliability of our results (**new Fig. S8E**). Our new qRT-PCR analysis showed that dermal THBS1 expression transiently increased in RT⁻ skin during wound healing, peaking at D3 *ex vivo* wounds, as confirmed by FISH analysis (**Fig. 5E**). This aligns with results from the human *in vivo* wound healing model (**Fig. 4B, C, F**).

- Through all the manuscript, the authors use TGF- β *in vitro* as a key mediator in wounding to infer differences in wound-response potential between RT⁺ and RT⁻ fibroblasts. In Figure S3, they show 5 module genes where THBS1 is located in cluster M3 (by looking at Table S9). This module represents genes downregulated in RT⁻ fibroblasts upon TGF- β , but that remain constant in RT⁺ with or without TGF- β . However, in Figure 3F, they did qPCR for THBS1 in presence and absence of TGF- β and showed a contradictory result whereby THBS1 is uniquely induced 24h after TGF- β in RT⁺ but not RT⁻ fibroblasts. The timepoint at which the RNA-seq was performed is not specified, and more generally the authors should clarify these differences between RNA-seq and qPCR regarding THBS1 in the presence/absence of TGF- β .

R3.2: We apologize for the mislabeled gene modules in **Table S9 (New Table S10)**, which we have corrected in the revised manuscript. THBS1 is located in cluster M5, not cluster M3 (**Fig. S3C**). This module represents genes unchanged in RT⁻ fibroblasts upon TGF-13 treatment but significantly induced in RT⁺ fibroblasts by TGF-13, consistent with the qRT-PCR results in **Fig. 3F**. In this RNA-seq experiment (**Fig. S3C**), RT⁻ and RT⁺ fibroblasts were treated with TGF-13 for 24 hours, the same as in the qRT-PCR experiment (**Fig. 3F**). We have clarified this time point in the legend of Fig. S3C.

- Fig. S8G: Considering that the authors show THBS1 is induced as a response to wounding and critical for long-term delays in wounding capacity, it seems strange that human wounds close at the same rate in RT⁻ fibroblasts when treated with control or with THBS1 antibody. A satisfactory explanation as to why THBS1 may be dispensable for normal wound-healing in these contexts needs to be presented and discussed.

R3.3: Thbs1 has been shown to regulate tissue repair in various mouse models¹. Importantly, both Thbs1 deletion and overexpression result in delayed wound repair, indicating that the quantity and duration of its expression are crucial for tissue healing^{2, 3, 4}.

During the revision, we established post-radiation wound models in C57BL/6 mice (*see our reply to the minor point 3 and new Fig. S7C-J*). We demonstrated that dermal Thbs1 expression is rapidly induced upon injury in both previously irradiated (IR⁺) and non-irradiated (IR⁻) murine skin (**new Fig. S7J**). However, Thbs1 levels remain elevated in IR⁺ wounds even at the late healing

stage (seven and ten days post-wounding), whereas in IR- wounds, Thbs1 returns to basal levels (**new Fig. S7J**).

We propose that early upregulation of THBS1 is important for wound healing, while its persistent expression may be detrimental. Consistent with this, we found that treating with a higher dose of THBS1 antibody (e.g., 66.7 µg/mL) immediately after injury (D0) blocked *ex vivo* wound closure of RT⁻ skin (**new Fig. S9G**). Therefore, we optimized the THBS1 antibody dose to 0.2 µg/mL and administered it two days post-wounding to block THBS1 at the late stage, but not the early stage, of wound healing. This regimen showed clear pro-healing effects in RT⁺ skin without affecting the healing of RT⁻ skin (**Fig. 6L, Fig. S9H, I**).

These findings support the safety profile of the THBS1 blocking antibody, indicating it can specifically target late-irradiated skin without impacting normal skin, provided the treatment dose and timing are optimized. We have revised the method section to specify the THBS1 antibody treatment dose and time (page 43, line 786-787, 790-791) and discussed this issue on page 19, line 398-406.

- Murine model: The authors characterize expression levels of THBS1 from D0-D10 in normal mouse wounding, both in whole-skin (Fig. S6C) and fibroblasts (Fig. S6D). How these patterns/expression levels are altered in post-irradiation settings (beyond %THBS1⁺ cells and prior to D33) is missing and should be addressed.

R3.4: As discussed above, we have performed post-radiation wound models in C57BL/6 mice during this revision, and confirmed the reduced healing capacity of previously irradiated (IR⁺) and non-irradiated (IR⁻) murine skin (*see our reply to the minor point 3 and new Fig. S7C-J*). We found that dermal Thbs1 expression is rapidly induced upon injury in both previously irradiated (IR⁺) and non-irradiated (IR⁻) murine skin (three days post-wounding). However, in IR⁺ wounds, Thbs1 levels remain elevated even at the late healing stage (seven and ten days post-wounding), whereas in IR⁻ wounds, Thbs1 returns to basal levels (**new Fig. S7J**). Therefore, the previous irradiation leads to a persistent Thbs1 expression during wound healing, which is in line with our previous findings in Fig. 5.

- Fig 3B: The presented tracks/peak-calls (black bars) around THBS1 look a bit strange:

- ATAC signal over the entire locus appears higher in RT⁺ fibroblasts (as opposed to within specific sub-domains), and peak calls do not overlay clearly distinguished peaks, and not all peaks appear consistently differential across respective RT⁺ vs. RT⁻ comparisons. Given the importance of THBS1 toward the study, the authors should provide a table summarizing the read coverage over this locus and statistical test. The authors should also widen their track over a larger area to determine whether this overall increase in signal in RT⁺ cells is truly specific to the THBS1 locus (as the scATAC in Fig. S5 suggests), e.g. to mirror the -Chr15:39850000-39600000 region shown in the scATAC. The method used to normalize bulk ATAC-seq read coverage for bigWigs should also be reported in the methods section.

R3.5: As suggested by Reviewer, we have widened the track to cover a larger area around the THBS1 locus (-Chr15:39575225-39608038, **new Fig. 3B**). The entire THBS1 locus is broadly accessible, with ATAC signals generally higher in RT⁺ fibroblasts compared to RT⁻ fibroblasts, and

differences in several sub-domains were statistically significant (**new Table S3**). This overall increase in signal in RT⁺ cells is truly specific to the THBS1 locus.

In our study, peaks were called on each sample individually using MACS2 (v2.2.6)⁵ and extended to 500 bp windows centred on the summits to avoid bias in differential accessibility analysis due to varying peak lengths. We have provided a table summarizing read coverage over the THBS1 locus and related statistical test results (**new Table S3**).

Additionally, we have revised the methods section to include the normalization process for bulk ATAC-seq read coverage for bigWigs. Specifically, "the uniquely mapped reads were shifted, and read coverages were then normalized with a method of RPKM and converted into bigWig format for IGV visualization⁶ using the deepTools⁷" (see supplementary methods page 25, line 233-235).

Minor Points:

1. Fig. 2b: Distribution of accessible genomic sites from RT⁺ and RT⁻ in pie charts do not represent reported values. Figure should be remade.

R4: The previous Fig. 2B was confusing because the left and middle panels displayed all accessible genomic sites in RT⁻ and RT⁺ fibroblasts, respectively, while the right panel showed the number of differentially accessible (DA) loci between the RT⁻ and RT⁺ fibroblasts. These panels looked similar, potentially causing misunderstandings. In the revised manuscript, we moved the left and middle pie charts to Fig. S2A to avoid confusion.

2. Fig. 2k: Replace 'and' by 'or' since they are representing union of DE genes in D1 with D7 in 'DE genes in human skin wound healing D1 and D17'.

R5: We have changed the new Fig. 2J following Reviewer's suggestion.

3. Fig. 5A: The mouse irradiated skin and wound model still needs further characterization.

- The authors used a citation to state that there is significantly delayed wound healing in irradiated mouse skin compared to non-irradiated mouse skin. Given the importance of this model toward the study, the authors should add a figure showing delayed wound healing in their mouse model rather than only a citation.

R6: We agree with Reviewer and have included **new Fig. S7A, B** to illustrate the wound healing dynamics in irradiated versus non-irradiated mouse skin. In addition to the CD-1 post-radiation wound model developed at Julie Fradette's lab in Canada, we established another model at Xu Landén's lab in Sweden during the revision (**new Fig. S7C**). We performed focal irradiation (20 Gy) on the shaved back skin of C57BL/6 mice using a 1 cm collimator. While 20 Gy caused no visible skin damage in CD-1 mice (data not shown), it induced radiodermatitis in C57BL/6 mice^{8, 9}.

We monitored the acute radiation effects (erythema, desquamation, ulceration, evaluated with RTOG scores¹⁰) appearing around 5 days and peaking at 14 days post-irradiation (IR⁺), with macroscopic recovery by 37 days (**new Fig. S7D, E**). Additionally, we found that dermal expression of Cdkn1a, a marker for DNA damage repair, increased within one day and returned to baseline at approximately one-week post-IR (**new Fig. S7F**).

We created wounds (4 mm in diameter) at both irradiated (IR+) and non-irradiated (IR-) sites 45 days post-IR, once acute effects had subsided, and also on non-irradiated control mice (Ctr). We monitored wound healing and collected biopsies at various healing stages (**new Fig S7H**). Consistent with the CD-1 model results (**new Fig. S7A, B**), we observed significantly delayed wound healing and persistent high expression of Thbs1 in IR+ skin compared to IR- skin in the C57BL/6 murine model (**new Fig. S7H-J**). These findings further support our conclusion about the long-term impacts of radiotherapy on skin wound healing.

4. The authors may consider rewording a few terms/phrases for clarity:

- “radiation wound models” (used throughout) should stress that wounding was only performed after acute irradiation (e.g. “post-radiation wound models”).
- “skin cells” (p5) may read as keratinocytes/epidermal stem cells rather than a variety of potential in the skin tissue, given its placement immediately after highlighting memory in epidermal stem cells (e.g. “...a variety of surviving skin cells after acute radiation...”).

R7: We have revised these as suggested by Reviewer.

References

1. Kyriakides TR, Maclauchlan S. The role of thrombospondins in wound healing, ischemia, and the foreign body reaction. *J Cell Commun Signal* **3**, 215-225 (2009).
2. Agah A, Kyriakides TR, Lawler J, Bornstein P. The lack of thrombospondin-1 (TSP1) dictates the course of wound healing in double-TSP1/TSP2-null mice. *Am J Pathol* **161**, 831-839 (2002).
3. DiPietro LA, Nissen NN, Gamelli RL, Koch AE, Pyle JM, Polverini PJ. Thrombospondin 1 synthesis and function in wound repair. *Am J Pathol* **148**, 1851-1860 (1996).
4. Streit M, *et al.* Thrombospondin-1 suppresses wound healing and granulation tissue formation in the skin of transgenic mice. *EMBO J* **19**, 3272-3282 (2000).
5. Feng J, Liu T, Qin B, Zhang Y, Liu XS. Identifying ChIP-seq enrichment using MACS. *Nat Protoc* **7**, 1728-1740 (2012).
6. Robinson JT, *et al.* Integrative genomics viewer. *Nat Biotechnol* **29**, 24-26 (2011).
7. Ramirez F, Dundar F, Diehl S, Gruning BA, Manke T. deepTools: a flexible platform for exploring deep-sequencing data. *Nucleic Acids Res* **42**, W187-191 (2014).
8. Ejaz A, Epperly MW, Hou W, Greenberger JS, Rubin JP. Adipose-Derived Stem Cell Therapy Ameliorates Ionizing Irradiation Fibrosis via Hepatocyte Growth Factor-Mediated Transforming Growth Factor-beta Downregulation and Recruitment of Bone Marrow Cells. *Stem Cells* **37**, 791-802 (2019).
9. Xiao Z, *et al.* Protective effect of esculentoside A on radiation-induced dermatitis and fibrosis. *Int J Radiat Oncol Biol Phys* **65**, 882-889 (2006).

10. Cox JD, Stetz J, Pajak TF. Toxicity criteria of the Radiation Therapy Oncology Group (RTOG) and the European Organization for Research and Treatment of Cancer (EORTC). *Int J Radiat Oncol Biol Phys* **31**, 1341-1346 (1995).

Reviewers' Comments:

Reviewer #1:

Remarks to the Author:

Rebuttal to Bian & Piipponen et al. (Second Revision)

Bian & Piipponen et al. have addressed most major comments raised during the first and second rounds of review, and the authors' new experiments add substantial rigor to their conclusions. The revised study is now a good fit for Nature Communications. However, there is one potentially serious issue (unfortunately missed during the first round of review/rebuttals) and several minor points that should be addressed prior to publication.

Major issue:

- The authors mention only using CD90-targeting microbead enrichment on dispase-separated dermal tissue to isolate fibroblasts, but do not mention any additional selection steps and do not provide a citation for their isolation strategy. The authors do show that THBS1 is consistently epigenetically dysregulated in post-irradiated dermis, and that this dysregulation can be targeted to correct radiation-driven delays in wound healing (Figs. 6, S9). In addition, the authors' single cell and imaging experiments suggest that THBS1-driven wound dynamics are indeed driven by fibroblasts, though their single cell data also suggest that some myeloid and endothelial compartments can additionally express THBS1 at least early during wounding (e.g. Fig. 4, S5, S6). This is consistent with publications of others showing that CD90 is expressed in many different cell populations.

Typical purification schemes involve multiple cell markers. Most importantly, the purity of the authors' strategy for fibroblasts vs. other dermal compartments should have been validated with flow cytometry on populations isolated directly in vivo and in vitro cultures of isolated cells, both of which were used to investigate epigenetic changes within post-irradiated skin. This characterization of the populations isolated from the authors' cell isolation strategy seems essential. If the authors are unable to document purity, as seems likely based upon their own data and published literature, they need to rephrase their conclusions. Instead of stating that fibroblasts are definitive carriers of radiation/THBS1-driven epigenetic changes from the beginning of the manuscript (as in current Fig. 2), the authors should instead argue that dermal cells broadly carry these changes, then use their subsequent data (e.g. single cell and imaging) to propose that fibroblasts could be the key subset within this broader collection.

Minor:

Several minor points should also be addressed to improve clarity of the presented findings:

- o Fig. 2C: A parallel analysis comparing specific peaks, i.e. not just associated genes, should be provided at least in supplement. While differences between in vitro and in vivo settings are appreciated, it will be worthwhile to understand whether the chromatin accessibility changes post-in vitro irradiation and post-irradiation therapy reflect changes over the same specific chromatin domains vs. general convergence over the same genes, but within different domains.

o Fig. 2D: All text except for “Negative regulation of cell migration” is greyed and difficult to read, which should be changed for legibility.

o Fig. 2I: The presented experimental schematic is confusing. The authors should only include bulk RNA-seq, as these data feed into 2J-K, and omit the other assays listed, which only appear in later figures (i.e. scRNA-seq, Visium, and FISH). At least a simplified version of the wounding schematic can be reproduced when presenting results for each assay.

o Fig. 2K: The authors should adjust their scale, as several differential genes appear to have fold-changes of zero (white boxes).

o Fig. 3K: The timing of TGF- β addition is unclear in the provided schematic. This should be updated to reflect TGF- β addition at 3h, D1, or D6 (as stated in the legend), rather than at all three of these timepoints.

o Fig. 5A-D: The CD1 post-radiation model is still weak (reasons below), while the authors’ new C57BL/6 model is much more compelling. The authors should move the C57 model to the main figures, and place the CD1 data in supplement.

The justification for using a 28-day resolution period post-irradiation of CD1 mice is unclear, especially since the authors’ previous study shows signs of erythema at D28 post-radiation at all doses used in the present study (Diaz et al. Surgeries 2021, Fig. 2). Resolution of irradiation is much better characterized in the new C57 model, which shows clear resolution of macroscopic features of irradiation by the post-irradiation wounding timepoint (Fig. S7E,F).

o Fig. S7I-J (C57 model) clearly show a quantified, temporal wound-healing defect, as well as a failure to resolve the normally transient induction of Thbs1 in post-irradiated skin. These temporal characterizations are missing from the CD1 model.

o The authors should perform fibroblast-specific characterizations (i.e. at least PDGFR α /THBS1 co-stainings) in their C57 model to more conclusively refine conclusions to fibroblasts, and not just pan-dermal cells.

o Fig. S9G: “THBS1-antibody” should be explicitly stated instead of just “RT- antibody”.

Methods:

o Several package versions are missing, including for BEDTools, DESeq2, ChIPseeker, HOMER, deepTools, and Subread. Some specific functions and parameters are also missing throughout (e.g. for deepTools and HOMER), and should be explicitly listed (e.g. “the HOMER function findMotifsGenome was used with default parameters to identify enriched sequence motifs...”).

Point by point response:

Bian & Piipponen et al. have addressed most major comments raised during the first and second rounds of review, and the authors' new experiments add substantial rigor to their conclusions. The revised study is now a good fit for Nature Communications. However, there is one potentially serious issue (unfortunately missed during the first round of review/rebuttals) and several minor points that should be addressed prior to publication.

Major issue:

- The authors mention only using CD90-targeting microbead enrichment on dispase-separated dermal tissue to isolate fibroblasts, but do not mention any additional selection steps and do not provide a citation for their isolation strategy. The authors do show that THBS1 is consistently epigenetically dysregulated in post-irradiated dermis, and that this dysregulation can be targeted to correct radiation-driven delays in wound healing (Figs. 6, S9). In addition, the authors' single cell and imaging experiments suggest that THBS1-driven wound dynamics are indeed driven by fibroblasts, though their single cell data also suggest that some myeloid and endothelial compartments can additionally express THBS1 at least early during wounding (e.g. Fig. 4, S5, S6). This is consistent with publications of others showing that CD90 is expressed in many different cell populations.

Typical purification schemes involve multiple cell markers. Most importantly, the purity of the authors' strategy for fibroblasts vs. other dermal compartments should have been validated with flow cytometry on populations isolated directly in vivo and in vitro cultures of isolated cells, both of which were used to investigate epigenetic changes within post-irradiated skin. This characterization of the populations isolated from the authors' cell isolation strategy seems essential. If the authors are unable to document purity, as seems likely based upon their own data and published literature, they need to rephrase their conclusions. Instead of stating that fibroblasts are definitive carriers of radiation/THBS1-driven epigenetic changes from the beginning of the manuscript (as in current Fig. 2), the authors should instead argue that dermal cells broadly carry these changes, then use their subsequent data (e.g. single cell and imaging) to propose that fibroblasts could be the key subset within this broader collection.

R: We appreciate Reviewer 1 for highlighting this crucial issue. In this study, we isolated RT⁺ and RT⁻ fibroblasts from patient skin using an explant outgrowth approach¹⁻³ (see 'Methods' section: 'Cell isolation and culture', lines 601-609). The CD90⁺ positive dermal cells were exclusively used in the experiment depicted in **Fig. 4A** (see the first paragraph of 'supplementary methods': 'Magnetic activation cell sorting')⁴. We have clarified the cell isolation strategies employed in different experiments and provided relevant citations in the revised manuscript.

Fibroblasts, due to their heterogeneity and lack of universal pan-fibroblast surface markers, are challenging to purify⁴. CD90/Thy-1 is the most commonly used surface marker for sorting viable fibroblasts from human skin⁴. Our single-cell RNA-seq data also corroborate its specific high expression in all four fibroblast clusters in human skin and day-1 acute wounds (**Fig. 1** for the rebuttal letter). However, as the Reviewer pointed out, CD90 could be expressed in other cell types than fibroblasts, such as dermal mesenchymal stem cells and endothelial cells⁵⁻⁸, and the human dermis also contains a CD90⁻ fibroblast population⁹. To address this, a FACS-based negative-selection strategy has been established to sort total fibroblasts (dermal cells with CD45⁻, CD31⁻,

CD235-, CD106-, ITGA6-, and E-cadherin-), which requires large skin samples and is technically complex¹⁰.

Fig. 1. The expression of CD90/THY1 in various cell types within human skin and day-1 acute wounds is demonstrated through single-cell RNA sequencing.

Currently, the most prevalent methods to obtain primary fibroblasts from human skin involve enzymatic digestion of the dermis or outgrowth of cells from explanted tissue pieces (including commercially available human primary fibroblasts). Although these methods are not based on specific markers, the culture conditions (DMEM, 10% fetal bovine serum, antibiotics) can select for fibroblast growth, and other cell types are unlikely to survive under these conditions¹. The purity of fibroblasts isolated by these methods has been confirmed by immunofluorescence analysis of fibroblast markers, e.g., Serpin H1, F-actin, and Vimentin¹, high fibronectin but no cytokeratin and epidermal marker expression, and no chromosomal diversity after passage 20 shown by karyotyping analysis². Therefore, we opted for the explant outgrowth approach to isolate fibroblasts from the RT+ and RT- skin biopsies from cancer survivors in this study.

In conclusion, there are no universally agreed-upon methods that can yield complete and pure fibroblast populations. Therefore, we concur with the Reviewer that it is imprecise to state from the outset of the manuscript that ‘fibroblasts are definitive carriers of radiation/THBS1-driven epigenetic changes’. However, stating that ‘dermal cells broadly carry these changes’ might confuse readers, as we did use a widely accepted method of fibroblast isolation and culturing¹⁻³. In the revised manuscript, we address this issue by clarifying how we obtained fibroblasts from patient skin and acknowledging the limitations of the methods at the beginning of the manuscript. We then used our single-cell and imaging data to further substantiate that fibroblasts are the primary cell type carrying radiation/THBS1-driven epigenetic changes.

Minor:

Several minor points should also be addressed to improve clarity of the presented findings:

o Fig. 2C: A parallel analysis comparing specific peaks, i.e. not just associated genes, should be provided at least in supplement. While differences between *in vitro* and *in vivo* settings are appreciated, it will be worthwhile to understand whether the chromatin accessibility changes post-*in vitro* irradiation and post-irradiation therapy reflect changes over the same specific chromatin domains vs. general convergence over the same genes, but within different domains.

R: In response to the Reviewer's suggestion, we compared the differentially accessible (DA) chromatin domains identified in both *in vitro* and *in vivo* settings (**Fig. 2C** and **the revised Table S5**). We discovered that, with the exception of two DA domains in SIN3B and NPSR1 that overlapped between the *in vitro* and *in vivo* settings, the majority of chromatin accessibility changes following *in vitro* irradiation and post-irradiation therapy were reflected in changes across different chromatin domains that converged on the same genes.

o Fig. 2D: All text except for "Negative regulation of cell migration" is greyed and difficult to read, which should be changed for legibility.

R: We have changed Fig. 2D as suggested.

o Fig. 2I: The presented experimental schematic is confusing. The authors should only include bulk RNA-seq, as these data feed into 2J-K, and omit the other assays listed, which only appear in later figures (i.e. scRNA-seq, Visium, and FISH). At least a simplified version of the wounding schematic can be reproduced when presenting results for each assay.

R: We have changed Fig. 2I as suggested.

o Fig. 2K: The authors should adjust their scale, as several differential genes appear to have fold-changes of zero (white boxes).

R: We have changed Fig. 2K as suggested.

o Fig. 3K: The timing of TGF- β addition is unclear in the provided schematic. This should be updated to reflect TGF- β addition at 3h, D1, or D6 (as stated in the legend), rather than at all three of these timepoints.

R: We have changed Fig. 3K as suggested.

o Fig. 5A-D: The CD1 post-radiation model is still weak (reasons below), while the authors' new C57BL/6 model is much more compelling. The authors should move the C57 model to the main figures, and place the CD1 data in supplement.

The justification for using a 28-day resolution period post-irradiation of CD1 mice is unclear, especially since the authors' previous study shows signs of erythema at D28 post-radiation at all doses used in the present study (Diaz et al. Surgeries 2021, Fig. 2). Resolution of irradiation is much better characterized in the new C57 model, which shows clear resolution of macroscopic features of irradiation by the post-irradiation wounding timepoint (Fig. S7E,F).

o Fig. S7I-J (C57 model) clearly show a quantified, temporal wound-healing defect, as well as a failure to resolve the normally transient induction of Thbs1 in post-irradiated skin. These temporal characterizations are missing from the CD1 model.

o The authors should perform fibroblast-specific characterizations (i.e. at least PDGFR α /THBS1 co-stainings) in their C57 model to more conclusively refine conclusions to fibroblasts, and not just pan-dermal cells.

R: We appreciate the reviewer's recognition of our results from the C57BL/6 model, established during the limited timeframe of the second round of revision. In response to the reviewer's suggestion, we have relocated the C57 model data to the main figure (**new Fig. 5A-D**), and included the CD1 data in **Fig. S7**. Collectively, these changes highlight the long-term effects of radiation on wound healing and Thbs1 expression, as demonstrated by two distinct murine models with complementary results.

We concur with the reviewer that repeating the PDGFR α /THBS1 co-staining, already performed in the CD1 model (**Fig. S7E**), in the C57BL/6 model would enhance the rigor of our study. However, our most recent experiment with the C57BL/6 model exhausted our supply of skin and wound tissues for qRT-PCR and ATAC-seq analysis, leaving no tissue sections available for current histological analysis. To conduct the PDGFR α /THBS1 co-staining, we would need to replicate the C57BL/6 post-irradiation wound model, which would require approximately three additional months. We kindly invite the reviewer and editor to consider the necessity of this requested experiment for our conclusion. We believe it's important to share these intriguing findings with the research community promptly, and we value your guidance in this matter.

o Fig. S9G: "THBS1-antibody" should be explicitly stated instead of just "RT- antibody".

R: We have changed Fig. S9G as suggested.

Methods:

o Several package versions are missing, including for BEDTools, DESeq2, ChIPseeker, HOMER, deepTools, and Subread. Some specific functions and parameters are also missing throughout (e.g. for deepTools and HOMER), and should be explicitly listed (e.g. "the HOMER function findMotifsGenome was used with default parameters to identify enriched sequence motifs...").

R: We have provided this information in the revised manuscript.

References

1. Iannello, G. *et al.* Simple, Fast, and Efficient Method for Derivation of Dermal Fibroblasts From Skin Biopsies. *Curr Protoc* **3**, e714 (2023).
2. Nejaddehbashi, F. *et al.* Isolating human dermal fibroblasts using serial explant culture. *Stem Cell Investig* **6**, 23 (2019).
3. Rittie, L. & Fisher, G.J. Isolation and culture of skin fibroblasts. *Methods Mol Med* **117**, 83-98 (2005).

4. Łuszczynski, K. *et al.* Markers of Dermal Fibroblast Subpopulations for Viable Cell Isolation via Cell Sorting: A Comprehensive Review. *Cells* **13**, 1206 (2024).
5. Chang, Y., Li, H. & Guo, Z. Mesenchymal stem cell-like properties in fibroblasts. *Cell Physiol Biochem* **34**, 703-714 (2014).
6. Haniffa, M.A., Collin, M.P., Buckley, C.D. & Dazzi, F. Mesenchymal stem cells: the fibroblasts' new clothes? *Haematologica* **94**, 258-263 (2009).
7. Jiang, D. & Rinkevich, Y. Defining Skin Fibroblastic Cell Types Beyond CD90. *Front Cell Dev Biol* **6**, 133 (2018).
8. Saalbach, A. & Anderegg, U. Thy-1: more than a marker for mesenchymal stromal cells. *FASEB J* **33**, 6689-6696 (2019).
9. Korosec, A. *et al.* Lineage Identity and Location within the Dermis Determine the Function of Papillary and Reticular Fibroblasts in Human Skin. *J Invest Dermatol* **139**, 342-351 (2019).
10. Korosec, A., Frech, S. & Lichtenberger, B.M. Isolation of Papillary and Reticular Fibroblasts from Human Skin by Fluorescence-activated Cell Sorting. *J Vis Exp* (2019).